# Saddle-to-Saddle Dynamics Explains A Simplicity Bias Across Neural Network Architectures

**Yedi Zhang**[1,*]    **Andrew Saxe**[1,2]    **Peter E. Latham**[1]
[1]Gatsby Computational Neuroscience Unit, University College London
[2]Sainsbury Wellcome Centre, University College London
[*]Correspondence: yedi@gatsby.ucl.ac.uk

## Abstract

Neural networks trained with gradient descent often learn solutions of increasing complexity over time, a phenomenon known as simplicity bias. Despite being widely observed across architectures, existing theoretical treatments lack a unifying framework. We present a theoretical framework that explains a simplicity bias arising from saddle-to-saddle learning dynamics for a general class of neural networks, incorporating fully-connected, convolutional, and attention-based architectures. Here, *simple* means expressible with few hidden units, i.e., hidden neurons, convolutional kernels, or attention heads. Specifically, we show that linear networks learn solutions of increasing rank, ReLU networks learn solutions with an increasing number of kinks, convolutional networks learn solutions with an increasing number of convolutional kernels, and self-attention models learn solutions with an increasing number of attention heads. By analyzing fixed points, invariant manifolds, and dynamics of gradient descent learning, we show that saddle-to-saddle dynamics operates by iteratively evolving near an invariant manifold, approaching a saddle, and switching to another invariant manifold. Our analysis also disentangles data-induced and initialization-induced saddle-to-saddle dynamics. In particular, the former leads to low-rank weights while the latter to sparse weights. Equipped with the theory, we predict the effects of data distribution and weight initialization on the duration and number of plateaus in learning. Overall, our theory offers a framework for understanding when and why gradient descent progressively learns increasingly complex solutions.

## 1 Introduction

Deep neural networks trained with gradient descent often learn functions of increasing complexity over the course of training (Arpit et al., 2017; Kalimeris et al., 2019; Rahaman et al., 2019; Saxe et al., 2019; Refinetti et al., 2023; Bhattamishra et al., 2023; Abbe et al., 2023). This dynamical simplicity bias has been observed across architectures (Shah et al., 2020; Teney et al., 2022; Edelman et al., 2024), tasks (Karkada et al., 2025; Wurgaft et al., 2025; Wang & Pehlevan, 2025), and training paradigms ranging from supervised (Rahaman et al., 2019) to reinforcement (Schaul et al., 2019) and self-supervised learning (Simon et al., 2023). A particularly striking manifestation is stage-like dynamics: extended plateaus in loss alternating with bursts of rapid improvement as networks progress through increasingly complex input-output maps (Saxe et al., 2014; 2019). These dynamics, known as "saddle-to-saddle" dynamics because they can result from trajectories passing near a sequence of saddle points (Jacot et al., 2022; Berthier, 2023; Pesme & Flammarion, 2023), have been documented in deep linear networks (Saxe et al., 2014; 2019; Gissin et al., 2020; Jacot et al., 2022), two-layer and deep ReLU networks (Maennel et al., 2018; Boursier et al., 2022; Chistikov et al., 2023; Wang & Ma, 2023; Kumar & Haupt, 2024; Zhang et al., 2025a; Wu et al., 2025; Bantzis et al., 2026), and self-attention models (Boix-Adsera et al., 2023; Rende et al., 2024; Geshkovski et al., 2024; Zhang et al., 2025b; Varre et al., 2025), and have been hypothesized to be universal (Ziyin et al., 2025; Kunin et al., 2025). Yet the same architectures can also exhibit smooth, exponential training dynamics, simply by changing the initialization (Jacot et al., 2018; Tu et al.,

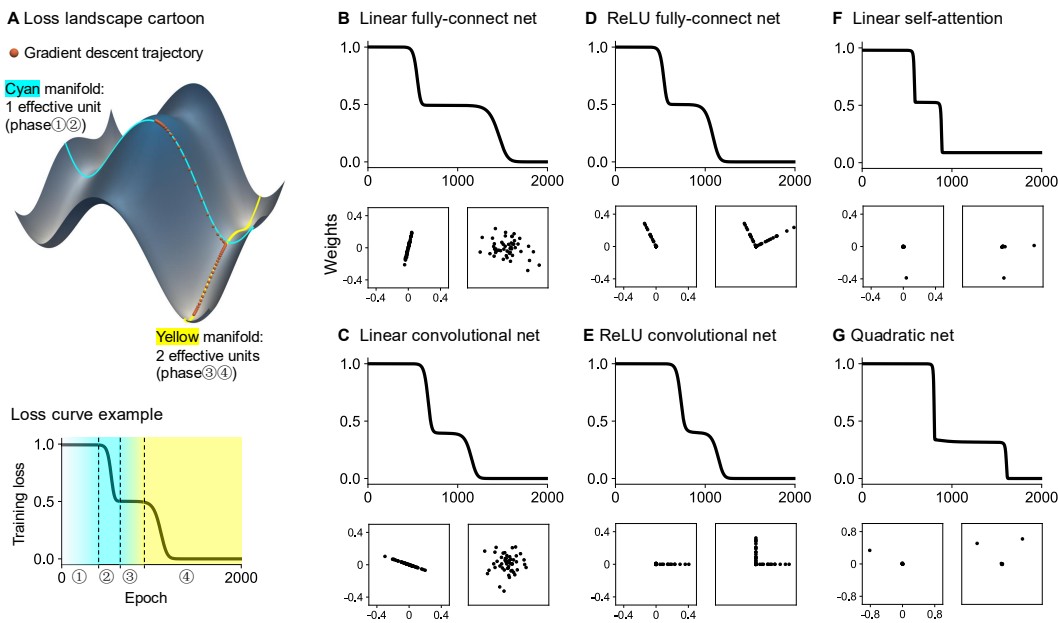

Figure 1: Saddle-to-saddle dynamics occurs in the gradient descent training of a wide range of architectures and leads to a dynamical simplicity bias. (A) Saddle-to-saddle dynamics on a cartoon loss landscape. The cyan and yellow curves represent invariant manifolds, on which the network implements input-output maps expressible by the architecture with one and two units, respectively. In general, saddle-to-saddle dynamics operates by repeating: i) during the plateau, escaping from a saddle associated with a width-$h$ network onto an invariant manifold with effective width $(h + 1)$; ii) during the rapid transition phase, approaching a fixed point on that manifold, which is a saddle associated with a width-$(h + 1)$ network. This figure shows two repeats of this process. (B-G) Loss and weight dynamics for various architectures. Each panel shows the loss during training (top), and the first-layer weights during the intermediate plateau (bottom left, phase 3 in panel A) and at the end of learning (bottom right). The first-layer weights to each hidden unit are two-dimensional and plotted as black dots. During the intermediate plateau, all networks visit a saddle, at which the input-output map of the network can be expressed by the architecture with only one unit. The network then converges to a stable fixed point, at which the input-output map is expressible with two units. The weight structures in BC, DE, and FG correspond to three categories of weight configurations of fixed points in Theorem 1; see Section 3 for details. A video version of this figure is provided at URL. Dynamics with other two-layer architectures and deep networks are provided in Figures 3 to 5. Experimental details are provided in Appendix I.

2024; Kunin et al., 2024); and more broadly, the emergence of stage-like dynamics can hinge on the data distribution (Yoshida & Okada, 2019; Goldt et al., 2020) and architectural choices (Orhan & Pitkow, 2018).

These diverse findings raise foundational questions about the nature of dynamical simplicity bias in deep neural networks. Is there a universal mechanism driving stage-like dynamics, or a collection of architecture-specific mechanisms? Is there a principled link between stages and simplicity, such that earlier stages in learning are simpler? And if simplicity does underlie these dynamics, what is the operative notion of simplicity, and how does it reflect an architecture's inductive bias?

Here we answer these questions. We show that for a range of architectures, including linear networks, ReLU networks, convolutional networks, quadratic networks, and linear self-attention (Figure 1B-G), there is a universal mechanism, saddle-to-saddle dynamics, driving stage-like learning. The operative notion of simplicity is the number of effective units in the architecture, i.e., hidden neurons, convolutional kernels, or attention heads. In particular, first we show that fixed points in the loss landscape are recursively embedded: fixed points of smaller networks are embedded in saddle points of larger networks, yielding a nested hierarchy of saddles. Second, we show that saddle points are connected by invariant manifolds along which a larger network behaves like a smaller one, preserving simplicity along the connecting trajectories. Third, the link between saddle-

to-saddle dynamics and simplicity arises from the interplay of the saddle hierarchy and timescale separation. Specifically, timescale separation steers dynamics toward invariant manifolds associated with simple input-output maps, thereby controlling the complexity increment at each stage. We also disentangle data-induced and initialization-induced timescale separation, showing that the former leads to low-rank weights (Figure 1B,C) while the latter leads to sparse weights (Figure 1F,G). Together, this theory paints a unified picture of embedded saddles, invariant manifolds, and dynamics which give rise to a simplicity bias across architectures, and predicts when instead non-stage-like behavior will arise.

**Related work**. We are inspired by a line of pioneering research that began with the seminal work of Fukumizu & Amari (2000) and continued in subsequent studies (Inoue et al., 2003; Amari et al., 2006; Wei et al., 2008; Amari et al., 2011; Fukumizu et al., 2019; Simsek et al., 2021; Zhang et al., 2021). In particular, Fukumizu & Amari (2000) first discovered a hierarchy of fixed points in two-layer fully-connected nonlinear neural networks. While their fixed points could, in principle, be extended to convolutional and attention-based architectures, they did not explore this, as convolutional architectures had not been popularized and attention-based architectures had not been invented. We study the fixed points across fully-connected, convolutional, and attention-based architectures. Further, we go beyond fixed points to study invariant manifolds and saddle-to-saddle dynamics, with implications for simplicity bias. A more detailed literature review is provided in Appendix A.

## 2 NETWORK SETUP

Let $f(\boldsymbol{x})$ represent a neural network with input $\boldsymbol{x} \in \mathbb{R}^D$. We focus on one layer in the network with $H$ units and trainable parameters $\boldsymbol{\theta}_{1:H}$,

$$f(\boldsymbol{x}; \boldsymbol{\theta}_{1:H}) = g_{\text{out}}\left(\sum_{i=1}^{H} \phi(g_{\text{in}}(\boldsymbol{x}); \boldsymbol{u}_i)\boldsymbol{v}_i\right), \quad \text{where } \boldsymbol{\theta}_i = \begin{bmatrix} \boldsymbol{v}_i \\ \boldsymbol{u}_i \end{bmatrix}. \quad (1)$$

Here $g_{\text{out}}(\cdot)$ and $g_{\text{in}}(\cdot)$ represent the processing after and before this layer, which are usually deeper and shallower layers of the network. The weights are $\boldsymbol{u}_i \in \mathbb{R}^{N_u}$, $\boldsymbol{v}_i \in \mathbb{R}^{N_v}$, and thus $\boldsymbol{\theta}_i \in \mathbb{R}^{N_u+N_v}$. We place the second-layer weight $\boldsymbol{v}_i$ on the right because $\phi(g_{\text{in}}(\boldsymbol{x}); \boldsymbol{u}_i)$ may be a scalar (as in a fully-connected layer) or matrix (as in a self-attention layer). The network output $f(\boldsymbol{x}; \boldsymbol{\theta}_{1:H})$ can be a scalar or vector. We will specify their dimensionality when we make them concrete.

The definition of a layer in Equation (1) incorporates major architectures. For a fully-connected layer, a unit is a hidden neuron: $\phi(\boldsymbol{z}; \boldsymbol{w}, b) = \sigma(\boldsymbol{w}^\top \boldsymbol{z} + b)$ where $\sigma(\cdot)$ is the activation function and $\boldsymbol{w}, b$ are the weight and bias. For a convolutional layer, a unit is a convolutional kernel: $\phi(\boldsymbol{z}; \boldsymbol{u}) = \sigma(\boldsymbol{u} * \boldsymbol{z})$ where $*$ denotes convolution. For a self-attention layer, a unit is an attention head: $\phi(\boldsymbol{Z}; \boldsymbol{K}, \boldsymbol{Q}) = \boldsymbol{I} \otimes \text{smax}(\boldsymbol{Z}\boldsymbol{Q}\boldsymbol{K}^\top \boldsymbol{Z}^\top)\boldsymbol{Z}$ where $\text{smax}(\cdot)$ denotes row-wise softmax and $\boldsymbol{K}, \boldsymbol{Q}$ are the key and query weights. A self-attention layer fits into our definition as follows,

$$\text{ATTN}(\boldsymbol{Z}) = \text{smax}(\boldsymbol{Z}\boldsymbol{Q}\boldsymbol{K}^\top \boldsymbol{Z}^\top)\boldsymbol{Z}\boldsymbol{V} = \boldsymbol{I} \otimes \text{smax}(\boldsymbol{Z}\boldsymbol{Q}\boldsymbol{K}^\top \boldsymbol{Z}^\top)\boldsymbol{Z}\text{vec}(\boldsymbol{V}) = \phi(\boldsymbol{Z}; \boldsymbol{K}, \boldsymbol{Q})\boldsymbol{v}. \quad (2)$$

We note that this is not a common notation for self-attention; we present it solely to show that Equation (1) incorporates self-attention. Hence, statements we will make about Equation (1) apply to fully-connected, convolutional, and self-attention architectures.

Let $\{\boldsymbol{x}_\mu, \boldsymbol{y}_\mu\}_{\mu=1}^P$ be a supervised learning training set. The training loss is averaged over the training set $\mathcal{L} = \frac{1}{P}\sum_{\mu=1}^P \ell(\boldsymbol{y}_\mu, f(\boldsymbol{x}_\mu))$, where the loss function $\ell$ is second order differentiable with respect to $f(\boldsymbol{x})$, including common choices like squared error loss. The parameters are trained with gradient flow on the training loss,

$$\dot{\boldsymbol{\theta}} = -\frac{\partial \mathcal{L}}{\partial \boldsymbol{\theta}} = -\frac{\partial \mathcal{L}}{\partial f(\boldsymbol{x})}\frac{\partial f(\boldsymbol{x})}{\partial \boldsymbol{\theta}}. \quad (3)$$

Gradient flow captures the behavior of gradient descent in the limit of a small learning rate.

**Definition 1.** A point $\boldsymbol{\theta}^*$ is a fixed point of the gradient flow dynamics in Equation (3) if $\frac{\partial \mathcal{L}}{\partial \boldsymbol{\theta}}\big|_{\boldsymbol{\theta}^*} = \boldsymbol{0}$.

## 3 LOSS LANDSCAPE: EMBEDDED FIXED POINTS

In this section, we establish that saddles generally exist in networks described by Equation (1). We show that a fixed point of a narrow network gives rise to a set of fixed points in a wider network.

These fixed points are constructed by embedding the narrow network into the wider network, as formalized in Theorem 1.

**Theorem 1** (Embedded fixed points). *If a network defined by Equation* (1) *with* $(H-1)$ *units has a fixed point* $\boldsymbol{\theta}^*_{1:(H-1)}$ *yielding an input-output map* $f^*(\boldsymbol{x})$, *then there exists* $\boldsymbol{\theta}_{1:H} \in \mathcal{S}$ *such that a network with* $H$ *units implements the same map* $f^*(\boldsymbol{x})$ *and* $\boldsymbol{\theta}_{1:H}$ *is a fixed point.*

*We construct* $\boldsymbol{\theta}_{1:H}$ *by setting the first* $(H-1)$ *units to* $\boldsymbol{\theta}^*_{1:(H-1)}$ *and modifying them as follows.*

*(i) For any* $\phi$, *the set* $\mathcal{S}$ *includes*

$$\boldsymbol{u}_H = \boldsymbol{u}^*_i, \; \boldsymbol{v}_H = \gamma_v \boldsymbol{v}^*_i, \; \boldsymbol{v}_i = (1-\gamma_v)\boldsymbol{v}^*_i, \quad \gamma_v \in \mathbb{R}, \, i \in \{1, \cdots, H-1\}. \tag{4}$$

*(ii) If* $\exists \, \boldsymbol{u}_{\mathrm{zero}}$ *such that* $\forall \boldsymbol{z}, \phi(\boldsymbol{z}; \boldsymbol{u}_{\mathrm{zero}}) = 0$, *the set* $\mathcal{S}$ *includes*

$$\boldsymbol{u}_H = \boldsymbol{u}_{\mathrm{zero}}, \boldsymbol{v}_H = \boldsymbol{0}. \tag{5}$$

*(iii) If* $\phi(\boldsymbol{z}; \boldsymbol{u})$ *is degree-1 homogeneous in* $\boldsymbol{u}$, *that is* $\forall \alpha \in \mathbb{F}, \phi(\boldsymbol{z}; \alpha\boldsymbol{u}) = \alpha\phi(\boldsymbol{z}; \boldsymbol{u})$, *where* $\mathbb{F} = \mathbb{R}$ *for general homogeneous functions, and* $\mathbb{F} = \mathbb{R}_{\geq 0}$ *for positively homogeneous functions, e.g., the ReLU activation function, the set* $\mathcal{S}$ *includes*

$$\boldsymbol{u}_H = \gamma_u \boldsymbol{u}^*_i, \; \boldsymbol{v}_H = \gamma_v \boldsymbol{v}^*_i, \; \boldsymbol{v}_i = (1-\gamma_u\gamma_v)\boldsymbol{v}^*_i, \quad \gamma_v \in \mathbb{R}, \gamma_u \in \mathbb{F}, i \in \{1, \cdots, H-1\}. \tag{6}$$

*(iv) If* $\phi(\boldsymbol{z}; \boldsymbol{u})$ *is linear in* $\boldsymbol{u}$, *that is degree-1 homogeneous,* $\forall \alpha \in \mathbb{R}, \phi(\boldsymbol{z}; \alpha\boldsymbol{u}) = \alpha\phi(\boldsymbol{z}; \boldsymbol{u})$, *and additive,* $\phi(\boldsymbol{z}; \boldsymbol{u}_i) + \phi(\boldsymbol{z}; \boldsymbol{u}_j) = \phi(\boldsymbol{z}; \boldsymbol{u}_i + \boldsymbol{u}_j)$, *the set* $\mathcal{S}$ *includes*

$$\boldsymbol{u}_H = \sum_{i=1}^{H-1} \gamma_{u_i} \boldsymbol{u}^*_i, \; \boldsymbol{v}_H = \sum_{i=1}^{H-1} \gamma_{v_i} \boldsymbol{v}^*_i, \quad \gamma_{v_i}, \gamma_{u_i} \in \mathbb{R},$$

$$\boldsymbol{v}_i = \boldsymbol{v}^*_i - \gamma_{u_i} \sum_{j=1}^{H-1} \gamma_{v_j} \boldsymbol{v}^*_j, \quad i = 1, \cdots, H-1. \tag{7}$$

The proof of Theorem 1, which is provided in the Appendix E, consists of two steps. First, verify that for the weight configurations given above, the width-$H$ network implements the same input-output map as the width-$(H-1)$ network. Second, show that gradients of the weights in the width-$H$ network are either equal or proportional to those in the width-$(H-1)$ network, which are zero.

*Remark* 1. Equation (4) is valid for any activation function $\phi$, while the rest are valid for $\phi$ with specific properties, implying that certain properties of $\phi$ give rise to a larger set of embedded fixed points in weight space. Equations (4) and (5) were first discovered by Fukumizu & Amari (2000). We extend these two constructions with Equations (6) and (7). This extension is crucial for studying learning dynamics, as the saddles visited during learning turn out to fall under Equations (5) to (7) but not Equation (4).

By induction, we obtain Corollary 2 by repeatedly applying Theorem 1 to embed multiple units in one layer and embed units in multiple layers of a deep network, with each layer defined by Equation (1).

**Corollary 2.** *If a depth-L network with* $h_l$ *units in layer* $l$ $(l = 1, \cdots, L)$ *has a fixed point yielding an input-output map* $f^*(\boldsymbol{x})$, *then for a depth-L network with* $H_l \geq h_l$ *units in each layer, there exist weight configurations such that the network implements the same map* $f^*(\boldsymbol{x})$ *and the weight configurations are fixed points.*

Theorem 1 and Corollary 2 indicate that the global minima of a narrow network, even if they incur nonzero training loss, remain fixed points of the gradient flow dynamics in any wider network with the same architecture. For example, the global minimum of a width-1 network typically lacks the expressivity to fit the training set and thus incurs nonzero loss. In a wide network capable of achieving zero loss, the fixed points corresponding to the width-1 network global minimum are either saddles or local minima. They are guaranteed to be saddles in deep linear networks with rank-$r$ ($r \geq 1$) target maps (Baldi & Hornik, 1989; Kawaguchi, 2016) and, under mild conditions, are saddles in general architectures (Fukumizu & Amari, 2000; Fukumizu et al., 2019).

In Figure 1, we show six cases where the network first visits a saddle, corresponding to a solution expressible by the architecture with a single unit. The network then converges to a stable fixed point,

corresponding to a solution expressible with two units. The fixed points visited during learning fit into three different categories in Theorem 1. In panels (B,C), the fixed points visited during learning are described by Equation (7), corresponding to rank-one and rank-two weights. In panels (D,E), the fixed points are described by Equation (6), corresponding to one and two rays of proportional weights. In panels (E,F), the fixed points are described by Equation (5), corresponding one or two units with large weights with the rest being near zero.

## 4    INVARIANT MANIFOLD: EFFECTIVELY NARROW NETWORKS

An invariant manifold of a dynamical system is a manifold such that any point starting on it remains on the manifold under the system's evolution. In Theorem 3, we show that for gradient flow dynamics of the class of neural networks we consider, invariant manifolds always exist. Further, these invariant manifolds correspond to weight configurations that make the network effectively narrower than its actual width.

**Theorem 3** (Invariant manifolds). *Let $T$ be any time such that one of the following conditions (i)-(iv) holds in a network defined by Equation (1). Then, in each case, the stated relationship between the weights is preserved for all $t \geq T$ under gradient flow dynamics:*

*(i) For any $\phi$, two units have equal weights: $\boldsymbol{\theta}_i = \boldsymbol{\theta}_j$.*

*(ii) If $\exists \, \boldsymbol{u}_{\mathrm{zero}}$ such that $\forall \boldsymbol{z}, \phi(\boldsymbol{z}; \boldsymbol{u}_{\mathrm{zero}}) = 0$, a unit has zero weights: $\boldsymbol{v}_i = \boldsymbol{0}, \boldsymbol{u}_i = \boldsymbol{u}_{\mathrm{zero}}$.*

*(iii) If $\phi(\boldsymbol{z}; \boldsymbol{u})$ is homogeneous in $\boldsymbol{u}$, two units have proportional weights: $\boldsymbol{\theta}_i = \gamma \boldsymbol{\theta}_j, \gamma \in \mathbb{F}$.*

*(iv) If $\phi(\boldsymbol{z}; \boldsymbol{u})$ is linear in $\boldsymbol{u}$, any number of units have linear dependence: $\boldsymbol{\theta}_i = \sum_{j \neq i} \gamma_j \boldsymbol{\theta}_j$.*

*The precise definitions of homogeneity and linearity are given in Theorem 1.*

The proof of Theorem 3 is provided in the Appendix F and is relatively straightforward. For example, when $\boldsymbol{\theta}_i = \boldsymbol{\theta}_j$, the gradients of $\boldsymbol{\theta}_i$ and $\boldsymbol{\theta}_j$ are equal and thus they stay equal for all future time. The invariant manifolds are larger in weight space when $\phi$ has zero, homogeneity or linearity properties, similar to the enlarged set of embedded fixed points in Theorem 1.

When the weights of a network lie on an invariant manifold, its input-output map is expressible with fewer units than its actual width: simply remove the $i$-th unit and appropriately modify the remaining weights (see Appendix F.3). Further, we can have more than one constraints; e.g., $\boldsymbol{\theta}_1 = \boldsymbol{\theta}_2$ and $\boldsymbol{\theta}_3 = \boldsymbol{\theta}_4$. Each added constraint reduces the effective width by 1. Hence, when weights evolve on an invariant manifold, the simplicity of the network's input-output map is constrained by the effective width associated with the invariant manifold, rather than the actual width.

The invariant manifolds indicate that there exist gradient flow paths connecting pairs of embedded fixed points defined in Theorem 1 (see Appendix F.4). Following such a path corresponds to an iteration of saddle-to-saddle dynamics. To see this, starting from an embedded fixed point with effective width $h$, we may apply a carefully chosen small perturbation that moves the weights onto the invariant manifold with effective width $(h + 1)$. This perturbation corresponds to breaking exactly one constraint. By Theorem 3, the dynamics then remains on the invariant manifold for all time, eventually converging to a fixed point on it, that is, an embedded fixed point with effective width $(h + 1)$. This process is one saddle-to-saddle transition: from the saddle with effective width $h$ to the saddle with $(h + 1)$. We illustrate this process in Figure 1A. In the next section, we develop heuristic arguments showing that the gradient flow dynamics can, in some cases, naturally evolve near such saddle-to-saddle paths on the invariant manifolds.

## 5    SADDLE-TO-SADDLE DYNAMICS

The embedded fixed points (Section 3) and invariant manifolds (Section 4) hold for general architectures defined by Equation (1). To analyze learning dynamics, however, we must work with concrete architectures. We focus on two-layer networks where $\phi(\boldsymbol{x}; \boldsymbol{u})$ is a homogeneous polynomial in the weights $\boldsymbol{u}$, studying the linear and quadratic cases in detail. The linear case includes fully-connected linear networks and convolutional linear networks. The quadratic case includes quadratic networks (defined by Equation (71)) and linear self-attention. Both types of architectures exhibit saddle-to-saddle dynamics, but their mechanisms differ. We show that the mechanism in the linear case is a

timescale separation between directions across all units due to the distribution of the data, while the the mechanism in the quadratic case is a timescale separation between units due to initialization.

## 5.1 LINEAR CASE: TIMESCALE SEPARATION BETWEEN DIRECTIONS

Consider a two-layer network in which $\phi(\boldsymbol{x}; \boldsymbol{u})$ is linear in the weights $\boldsymbol{u}$,

$$f(\boldsymbol{x}; \boldsymbol{\theta}_{1:H}) = \sum_{i=1}^{H} \boldsymbol{v}_i \boldsymbol{u}_i^\top \boldsymbol{z}(\boldsymbol{x}) \equiv \boldsymbol{W}\boldsymbol{z}, \quad \text{where } \boldsymbol{v} \in \mathbb{R}^{N_v}, \boldsymbol{u}, \boldsymbol{z} \in \mathbb{R}^{N_u}. \tag{8}$$

Here $\boldsymbol{z}(\boldsymbol{x})$ denotes any function of the input $\boldsymbol{x}$, as $\phi(\boldsymbol{x}; \boldsymbol{u})$ is linear in $\boldsymbol{u}$ but not necessarily linear in $\boldsymbol{x}$. The gradient flow dynamics of Equation (8) trained on squared loss is

$$\dot{\boldsymbol{v}}_i = (\boldsymbol{\Sigma}_{yz} - \boldsymbol{W}\boldsymbol{\Sigma}_{zz})\,\boldsymbol{u}_i, \quad \dot{\boldsymbol{u}}_i = (\boldsymbol{\Sigma}_{yz} - \boldsymbol{W}\boldsymbol{\Sigma}_{zz})^\top \boldsymbol{v}_i, \quad i = 1, \cdots, H, \tag{9}$$

where the data statistics are $\boldsymbol{\Sigma}_{yz} = \frac{1}{P} \sum_{\mu=1}^{P} \boldsymbol{y}_\mu \boldsymbol{z}_\mu^\top, \boldsymbol{\Sigma}_{zz} = \frac{1}{P} \sum_{\mu=1}^{P} \boldsymbol{z}_\mu \boldsymbol{z}_\mu^\top$. When the weights are initialized to be small, i.e., $\boldsymbol{v}_i(0) = O(\epsilon), \boldsymbol{u}_i(0) = O(\epsilon), i = 1, \cdots, H$, the first terms in Equation (9) dominate: $\boldsymbol{\Sigma}_{yz} - \boldsymbol{W}\boldsymbol{\Sigma}_{zz} = \boldsymbol{\Sigma}_{yz} + O(\epsilon^2)$. The weights thus approximately evolve as a linear dynamical system (Equation (10)), which we analyze in Theorem 4.

**Theorem 4** (Timescale separation between directions). *Consider the linear dynamical system*

$$\dot{\boldsymbol{v}}_i = \boldsymbol{\Sigma}_{yz}\boldsymbol{u}_i, \quad \dot{\boldsymbol{u}}_i = \boldsymbol{\Sigma}_{yz}^\top \boldsymbol{v}_i, \quad i = 1, \cdots, H. \tag{10}$$

*Let the singular value decomposition of $\boldsymbol{\Sigma}_{yz}$ be given by $\boldsymbol{\Sigma}_{yz} = \sum_{k=1}^{D} s_k \boldsymbol{q}_k \boldsymbol{r}_k^\top$, $D = \min(N_v, N_u)$ with singular values $s_1 \geq \cdots \geq s_D$, and let the largest singular value $s_1$ have multiplicity $r$ ($1 \leq r < D$). Let the initial weights be sampled independently from a Gaussian distribution $\mathcal{N}(0, \epsilon^2)$ with a small $\epsilon$. When the projection of the weights on the span of the top $r$ singular vectors reaches $O(1)$, that is*

$$\|\boldsymbol{P}\boldsymbol{\theta}_i\| = O(1), \quad \text{where } \boldsymbol{P} = \frac{1}{2} \sum_{k=1}^{r} \begin{bmatrix} \boldsymbol{q}_k \\ \boldsymbol{r}_k \end{bmatrix} \begin{bmatrix} \boldsymbol{q}_k^\top & \boldsymbol{r}_k^\top \end{bmatrix}, \boldsymbol{\theta}_i = \begin{bmatrix} \boldsymbol{v}_i \\ \boldsymbol{u}_i \end{bmatrix}, \tag{11}$$

*the projection on the remaining subspace is $\|(\boldsymbol{I} - \boldsymbol{P})\boldsymbol{\theta}_i\| = O(\epsilon^{1 - s_{r+1}/s_1})$ almost surely.*

We provide the proof in Appendix G.2 and the intuition here. The second and first-layer weights $\boldsymbol{v}_i, \boldsymbol{u}_i$ grow exponentially along the singular vectors $\boldsymbol{q}_k, \boldsymbol{r}_k$, respectively, at the rate $e^{s_k t}$. Relative to the dominant growth rate $e^{s_1 t}$ along the top singular vectors, the components along other singular vectors decay as $e^{(s_k - s_1)t}, k = r + 1, \cdots, D$. Consequently, during the early phase, the weights become increasingly aligned with the top singular vectors and thus approximately rank-$r$. Taking $r = 1$ as an example, the weights become approximately rank-one; specifically, $\boldsymbol{v}_i$ aligns with $\boldsymbol{q}_1$, and $\boldsymbol{u}_i$ aligns with $\boldsymbol{r}_1$ for every $i$.

Theorem 3 implies that rank-$r$ weights constrain a linear network to an invariant manifold corresponding to effective width $r$. Since the early phase dynamics drives the weights to be approximately rank-$r$, the network evolves near the invariant manifold and approaches a fixed point on it. This is the first iteration of saddle-to-saddle dynamics. In weight space, the weights move from the initial saddle at zero to the second saddle. In function space, the network learns a more complex solution, changing from a constant zero function to a rank-$r$ projection of the target linear map.

Subsequent iterations of saddle-to-saddle dynamics operate similarly. The dynamics near a rank-$r$ saddle, corresponding to a plateau in the loss, is again approximately a linear dynamical system

$$\dot{\boldsymbol{v}}_i = \widetilde{\boldsymbol{\Sigma}}_{yz}\boldsymbol{u}_i, \quad \dot{\boldsymbol{u}}_i = \widetilde{\boldsymbol{\Sigma}}_{yz}^\top \boldsymbol{v}_i, \quad i = 1, \cdots, H. \tag{12}$$

where $\widetilde{\boldsymbol{\Sigma}}_{yz}$ is $\boldsymbol{\Sigma}_{yz}$ projected onto a rank-$(D - r)$ subspace; see Appendix G.3. Via the same reasoning as Theorem 4, the weights grow the fastest along the top singular vectors of $\widetilde{\boldsymbol{\Sigma}}_{yz}$. Low-rank weight growth will again place a linear network near an invariant manifold with few more effective units, guiding the dynamics toward a fix point on that manifold.

To summarize, in the linear case, distinct singular values of the input-output correlation matrix induce a timescale separation between weight growth along different directions. If all singular values are distinct, the timescale separation leads to approximately rank-one weight growth during a loss plateau, causing the escape path from a saddle to closely follow an invariant manifold with one more effective unit.

## 5.2 QUADRATIC CASE: TIMESCALE SEPARATION BETWEEN UNITS

We now consider a two-layer network in which $\phi(\boldsymbol{x}; \boldsymbol{u})$ is quadratic in the weights $\boldsymbol{u}$,

$$f(\boldsymbol{x}; \boldsymbol{\theta}_{1:H}) = \sum_{i=1}^{H} v_i \boldsymbol{u}_i^\top \boldsymbol{Z}(\boldsymbol{x}) \boldsymbol{u}_i, \quad \text{where } v_i \in \mathbb{R}, \boldsymbol{u}_i \in \mathbb{R}^D, \boldsymbol{Z} \in \mathbb{R}^{D \times D}. \tag{13}$$

Here $\boldsymbol{Z}(\boldsymbol{x})$ denotes any function of the input $\boldsymbol{x}$. For example, linear self-attention fits into Equation (13) with $\boldsymbol{Z}(\boldsymbol{x})$ being a cubic function of the input $\boldsymbol{x}$, and $\phi(\boldsymbol{x}; \boldsymbol{u})$ a quadratic function of the key and query weights $\boldsymbol{u} = [\text{vec}(\boldsymbol{K}), \text{vec}(\boldsymbol{Q})]$. We consider the scalar output case because it already has saddle-to-saddle dynamics and involves non-closed-form solutions. The gradient flow dynamics of Equation (13) trained on squared loss is given by Equation (44). Near small initialization, the quadratic terms in Equation (44) dominate. In Proposition 5, we analyze the approximate dynamics and show that one unit with the largest initialization grows much faster than the rest.

**Proposition 5** (Timescale separation between units). *Consider the dynamical system*

$$\dot{v}_i = \boldsymbol{u}_i^\top \boldsymbol{\Sigma}_{yZ} \boldsymbol{u}_i, \quad \dot{\boldsymbol{u}}_i = 2 v_i \boldsymbol{\Sigma}_{yZ} \boldsymbol{u}_i, \quad i = 1, \cdots, H. \tag{14}$$

*Assume $\boldsymbol{\Sigma}_{yZ}$ is symmetric and has both positive and negative eigenvalues. Let the initial weights be sampled independently from a Gaussian distribution $\mathcal{N}(0, \epsilon^2)$ with a small $\epsilon$. When weights in one of the units reaches $O(1)$, the rest of the units is $O(\epsilon)$ almost surely.*

We provide derivations in Appendix H.2 and the intuition here. The intuition is that the quadratic dynamics in Equation (14) is a rich-get-richer process. We can get a flavor of such dynamics by considering the simplest quadratic dynamics, $\dot{v}_i = v_i^2$, which has the solution

$$v_i(t) = \left( \frac{1}{v_i(0)} - t \right)^{-1}, \quad i = 1, \cdots, H. \tag{15}$$

By solving for $t$ with $i$ and $j$, we can write $v_i(t)$ in terms of $v_j(t)$ as

$$v_i(t) = \left[ \frac{1}{v_j(0)} \left( \frac{v_j(0)}{v_i(0)} - 1 \right) + \frac{1}{v_j(t)} \right]^{-1}. \tag{16}$$

Assuming initial conditions of order $O(\epsilon)$, for example $v_i(0) \sim \mathcal{N}(0, \epsilon^2)$, and letting $v_j$ be the unit with the largest initial value, we see that when $v_j(t) \sim O(1)$, the other units are still small: $v_i(t) \sim O(\epsilon)$ for $i \neq j$. Thus, under quadratic dynamics $\dot{v}_i = v_i^2$, distinct initial conditions of the units induce a timescale separation in their growth. Although the general case, analyzed in Appendix H.2, is more complicated, the timescale separation between units essentially comes from the same mechanism.

In Theorem 3(*ii*), we showed that if $\phi(\boldsymbol{x}; \boldsymbol{0}) = 0 \, \forall \boldsymbol{x}$, then nonzero weights in one unit and zero weights in the rest of the units constrain a network to an invariant manifold with effective width one. Since the early dynamics drives one unit to grow much faster than the rest, the network evolves near the invariant manifold with effective width one and approaches a fixed point on it. This is the first iteration of saddle-to-saddle dynamics. Subsequent iterations operate similarly. Starting near the first saddle, one unit has nonzero weights and $(H - 1)$ units still have small weights. The dynamics near the first saddle drives one of the $(H - 1)$ units to grow much faster than the rest. Hence, the escape path from the first saddle again approximately follow the invariant manifold with two effective units, steering the dynamics toward a fixed point on that manifold. This process repeats.

For $\phi(\boldsymbol{x}; \boldsymbol{u})$ that is quadratic in $\boldsymbol{u}$ and has $\phi(\boldsymbol{x}; \boldsymbol{0}) = 0 \, \forall \boldsymbol{x}$, the distinct initial weights in each unit induce a timescale separation between the weight growth in different units. One unit grows much faster than the rest, causing the escape path from a saddle to closely follow an invariant manifold with one more effective unit.

**Higher-order polynomial activation**. If $\phi(\boldsymbol{x}; \boldsymbol{u})$ is a homogeneous polynomial of degree $p > 2$ in the weights $\boldsymbol{u}$, we conjecture that there is still a timescale separation between units, possibly even stronger than the quadratic case. Our intuition is that the dynamics near zero has a similar flavor to the scalar dynamics, $\dot{v}_i = v_i^p$. By similar reasoning to Proposition 5, the unit with the largest initialization grows much faster than the rest, causing a timescale separation between units. The dynamics in the cubic ($p = 3$) case is consistent with our intuition, as shown in Figure 4G.

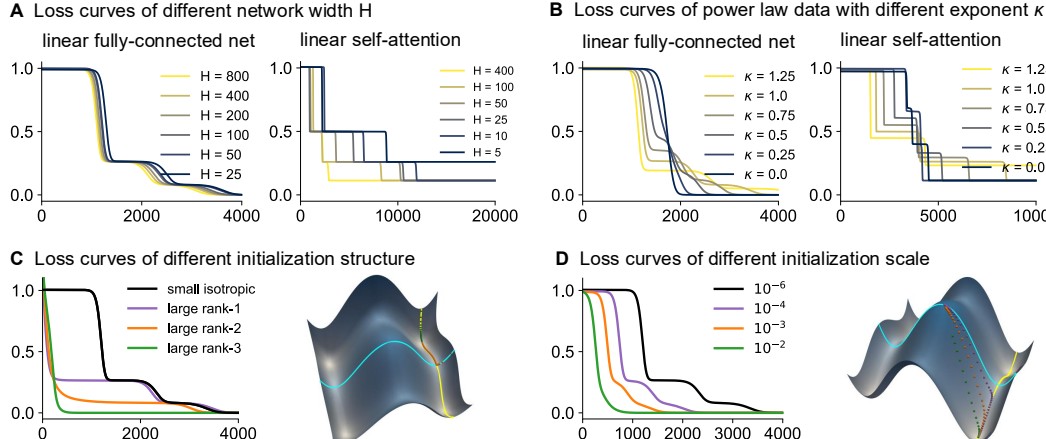

Figure 2: The effect of network width, data distribution, and initialization on learning dynamics. Singular values of $\mathbf{\Sigma}_{yz}$ (linear network) or positive singular values of $\mathbf{\Sigma}_{yZ}$ (linear self-attention) follow a power law, $s_n = n^{-\kappa}, n = 1, 2, 3$, and are normalized such that $\sum_{n=1}^{3} s_n = 1$. (A) Increasing the number of units $H$ has little effect on the loss curves of linear networks, but shortens the plateaus in linear self-attention. $\kappa = 1$ for both models. (B) Decreasing the power law exponent $\kappa$ shortens the plateaus in both linear networks and linear self-attention. Setting $\kappa = 0$ eliminates plateaus in linear networks but does not eliminate plateaus in linear self-attention. $H$ is 100 for linear networks and 25 for linear self-attention. (C) Linear networks with small isotropic initialization or large low-rank initialization exhibit saddle-to-saddle dynamics. The loss landscape cartoon illustrates large rank-$r$ weights places a linear network near an invariant manifold with $r$ effective units and thus approaches saddles during learning. (D) Increasing the scale of isotropic random initialization shortens the plateaus. $\kappa = 1$ for panels C,D.

**General nonlinear activation**. If $\phi(\boldsymbol{x}; \boldsymbol{u})$ is a general nonlinear activation function, we can Taylor expand $\phi(\boldsymbol{x}; \boldsymbol{u})$ around $\boldsymbol{u} = \boldsymbol{0}$. With small initialization, $\boldsymbol{u} \approx \boldsymbol{0}$, the early dynamics near initialization is dominated by the lowest-order non-vanishing term in the Taylor expansion, assuming the data statistic associated with that term is nonzero. For example, in a two-layer fully-connected tanh network, the lowest-order non-vanishing term is the linear term. The tanh network thus develops rank-one weights in the early phase near initialization, similar to Theorem 4. However, the subsequent dynamics is not necessarily saddle-to-saddle, since rank-one weights do not generally correspond to invariant manifolds for tanh networks; see Figure 4D. By comparison, in a two-layer fully-connected network with activation $\phi(\boldsymbol{x}; \boldsymbol{u}) = \boldsymbol{u}^{\top} \boldsymbol{x} \cdot \tanh(\boldsymbol{u}^{\top} \boldsymbol{x})$, the lowest-order non-vanishing term is quadratic. The network thus has a timescale separation between units similar to Proposition 5, and exhibits saddle-to-saddle dynamics as shown in Figure 4F.

## 6 IMPLICATIONS

We now validate our theory and demonstrate its predictive power by examining how the network width, data distribution, and initialization affect learning dynamics.

**Effect of network width**. Our analysis in Section 5.1 shows that in linear networks, the timescale separation occurs between directions across all units. Consequently, increasing the number of units in linear networks has little effect on the dynamics, provided there are enough units to learn all directions. In contrast, the analysis in Section 5.2 implies that increasing the number of units in networks where $\phi(\boldsymbol{x}; \boldsymbol{u})$ that is quadratic in $\boldsymbol{u}$ can shorten the plateaus. That is because the timescale separation in the quadratic case occurs between learning different units due to their distinct initial values. When sampling initial weights from a fixed distribution, increasing the number of weights reduces the gaps between adjacent samples, thereby shortening the plateaus. Simulations in Figure 2A confirm our theoretical prediction. In this case, increasing the number of heads of linear self-attention, for which $\phi(\boldsymbol{x}; \boldsymbol{u})$ is quadratic in $\boldsymbol{u}$, speeds up learning, while increasing the width of fully-connected linear networks does not. This demonstrates an interesting, theoretically grounded advantage of scaling up linear self-attention over scaling up fully-connected linear networks.

**Effect of data distribution**. In linear networks, the timescale separation in learning different directions arises from the distinct singular values of $\mathbf{\Sigma}_{yz}$. In Figure 2B, we let the singular values of $\mathbf{\Sigma}_{yz}$ follow a power law. As expected, decreasing the power law exponent narrows the gaps between singular values, thereby shortening the plateaus. When the exponent is 0, all the singular values are equal, eliminating the plateaus except the initial one corresponding to the escape from the saddle at zero. In this case, the largest singular value has multiplicity $r = D$ in Theorem 4, causing the solution to jump directly from effective width 0 to $D$, skipping the stages in between. By contrast, in networks for which $\phi(\boldsymbol{x}; \boldsymbol{u})$ is quadratic in $\boldsymbol{u}$, the timescale separation is due to the distinct initial values in the units. Therefore, setting the positive singular values of $\mathbf{\Sigma}_{yZ}$ to be equal shortens but does not eliminate plateaus. Simulations with linear self-attention in Figure 2B confirm our prediction.

**Effect of initialization structure**. According to our theory, to have saddle-to-saddle dynamics the initialization must be near an invariant manifold, and the escape path from saddles must follow an invariant manifold. Perhaps surprisingly, however, initializing near a saddle is not a necessary condition. In Figure 2C we initialize the weights near an invariant manifold but away from saddles; for linear networks, this corresponds to large low-rank weights with a small perturbation. As predicted, learning undergoes saddle-to-saddle dynamics. Because the initialization is away from saddles, there is not a plateau at the start; the loss first drops exponentially and then exhibits plateaus followed by sigmoid-shaped drops. To our knowledge, this regime has not previously been observed. If we initialize near the invariant manifold associated with exactly the required number of effective units, loss undergoes a rapid exponential drop, even though the network learns a solution with low-rank weights, which is the feature learning solution in linear networks (Dominé et al., 2025). This result adds nuance to the common view that exponential loss curves are often a hallmark of lazy learning (Jacot et al., 2018; Chizat et al., 2019).

**Effect of initialization scale**. We examine the effect of initialization scale when using an isotropic Gaussian distribution, a common choice in practice. As shown in Figure 2D, increasing the initialization scale gradually shortens the plateaus. Saddle-to-saddle dynamics becomes weaker in the sense that the learning trajectory does not approach the saddles as closely as it does with small initialization. For intermediate initialization scales, plateaus are less pronounced, yet the network still approximately learns solution of increasing complexity, similar to the case with small initialization. In architectures that have saddle-to-saddle dynamics, we conjecture that the distance from the initial weights to invariant manifolds associated with low effective width determines the strength of feature learning. This criterion can be viewed as an extension of prior beliefs, in which the relative scale of initial weights across layers (Dominé et al., 2025) or the rank of initial weights (Liu et al., 2024) were thought to determine the strength of feature learning.

## 7 DISCUSSION

We studied the gradient flow dynamics of a broad class of architectures, analyzing fixed points, invariant manifolds, and dynamics near fixed points. Our theoretical framework reveals a general mechanism for saddle-to-saddle dynamics and provides a definition of simplicity that reflects the inductive biases of different architectures. When a network exhibits saddle-to-saddle dynamics, it recruits one or a few new effective units during each transition and learns solutions of increasingly complexity, where complexity is measured by the minimal number of units required for the architecture to express the solution. On a high level, we identify a mechanism behind the intuition that a neural network can decompose a task into smaller pieces and learn piece by piece over time. The learning process sometimes reconstructs the network's own architecture, one unit at a time.

**Condition for saddle-to-saddle dynamics**. Saddle-to-saddle dynamics depends on two conditions: (i) the escape path from saddles closely follows invariant manifolds with few additional effective units; and (ii) the initialization is close to an invariant manifold with fewer effective units than needed to attain zero loss. As an example violating the first condition, two-layer tanh networks with small initialization develop rank-one weights during the early phase. This is because the tanh function is approximately linear near zero, and thus the early dynamics is approximately a linear dynamical system, similar to Theorem 4. However, since tanh is not homogeneous, rank-one weights do not correspond to an invariant manifold with effective width one. Consequently, tanh networks are not guided to approach the saddle with one effective unit, and probably do not have saddle-to-

saddle dynamics in general. As an example violating the second condition, large isotropic random initialization is almost surely away from invariant manifolds. Thus, neural networks with large random initialization generally do not exhibit saddle-to-saddle dynamics. A special case violating the second condition is when an architecture has full expressivity with a single unit, such as linear networks with scalar input or scalar output (Shamir, 2019), and linear self-attention with merged key and query weights (Zhang et al., 2025b).

**Deep networks**. The fixed points and invariant manifolds in Sections 3 and 4 apply to general deep networks defined by Equation (1), whereas the analysis of dynamics in Section 5 only applies to two-layer networks. Nonetheless, many deep newtorks still exhibit saddle-to-saddle dynamics, with some showing a timescale separation between directions and some between units, as shown in Figure 5. Although a general treatment of deep network dynamics is beyond the scope of this paper, we propose a conjecture for predicting which type of timescale separation (between directions or units) arises within a layer of a deep network. We conjecture that the order of the activation function $\phi(g_{\text{in}}(\boldsymbol{x}); \boldsymbol{u}_i)$, whether it is linear or quadratic in $\boldsymbol{u}_i$, continues to predict learning behaviors, including the type of the timescale separation and the effects of width and data distribution. In deep networks, $g_{\text{in}}(\boldsymbol{x})$ in Equation (1) may involve weights that are not specific to any individual unit of the layer under consideration, i.e., weights in shallower layers not indexed by $i$. For example, let us consider the second hidden layer of a depth-3 linear fully-connected network:

$$f(\boldsymbol{x}) = \sum_{i=1}^{H} \boldsymbol{v}_i \phi(g_{\text{in}}(\boldsymbol{x}); \boldsymbol{u}_i) = \sum_{i=1}^{H} \boldsymbol{v}_i \boldsymbol{u}_i^\top g_{\text{in}}(\boldsymbol{x}), \quad \text{where } g_{\text{in}}(\boldsymbol{x}) = \boldsymbol{W}\boldsymbol{x}, \tag{17}$$

where $\boldsymbol{W}$ is the first-layer weight matrix.[1] Since $\phi(g_{\text{in}}(\boldsymbol{x}); \boldsymbol{u}_i) = \boldsymbol{u}_i^\top g_{\text{in}}(\boldsymbol{x})$ is linear in $\boldsymbol{u}_i$, we predict a timescale separation between directions similar to Section 5.1, and that the weights acquire an additional rank during each saddle-to-saddle transition. This is consistent with the existing literature (Gidel et al., 2019; Gissin et al., 2020) and our simulations in Figure 5.

We further note that deep networks introduce several new questions that do not arise in the two-layer setting. If deep networks visit a sequence of embedded fixed points and learn increasingly complex solutions by recruiting additional effective units, which layers recruit additional units at each increase in complexity? This question is particularly interesting for transformers, which have self-attention, fully-connected layers, and skip connections. With skip connections, a deep network may also learn increasingly complex solutions by recruiting additional layers. This possibility seems consistent with the literature on layer pruning showing that large-scale transformers maintain their performance when removing up to half of the deeper layers and performing a small amount of finetuning (Gromov et al., 2025). Another work modeled the increasingly complex solutions of a transformer by increasing the width of its fully-connected layers (Wurgaft et al., 2025).

**Exhaustiveness of fixed points and invariant manifolds**. Although we have not identified any fixed points or invariant manifolds beyond Proposition 5 and Theorem 3, it remains an open question whether these are exhaustive. If not, under what conditions do they become so? If the fixed points are exhaustive under reasonable assumptions, they would provide a useful diagnostic: each plateau during training would indicate that the network is implementing a solution expressible by a narrower sub-network. Moreover, the fixed points and invariant manifolds we describe arise solely from the network architecture and thus hold for any training data set. A further question is whether particular data sets can induce more fixed points or invariant manifolds than the data-agnostic ones (Zhao et al., 2023; Misof et al., 2025).

**Other architectures and learning rules**. At its core, our theory exploits the permutation symmetry of units in feed-forward neural networks defined by Equation (1). Permutation symmetry exists beyond feed-forward architectures and supervised learning rules. Indeed, stage-like learning curves have been observed in recurrent neural networks (Proca et al., 2025; Ger & Barak, 2025), and other learning rules, such as reinforcement learning (Schaul et al., 2019), self-supervised learning (Simon et al., 2023), and predictive coding (Innocenti et al., 2024). This suggests the possibility of an even broader theory that incorporates these architectures and learning rules, with progressive permutation symmetry breaking as a unifying explanation for progressive learning behaviors.

---

[1]A depth-3 linear network differs from linear self-attention, $\sum_{i=1}^{H} \boldsymbol{X}\boldsymbol{Q}_i\boldsymbol{K}_i^\top \boldsymbol{X}^\top \boldsymbol{X}\boldsymbol{V}_i$, because in linear self-attention all weights are indexed by $i$, and thus cannot be absorbed into $g_{\text{in}}(\boldsymbol{x})$.

ACKNOWLEDGMENTS

We thank Samuel Liebana, Loek van Rossem, Erin Grant, Stefano Sarao Mannelli, Máté Lengyel, Valentina Njaradi, Aaditya K. Singh, Andrew Lampinen, and Jin Hwa Lee for helpful conversations, and anonymous reviewers for their constructive feedback.

We thank the following funding sources: Gatsby Charitable Foundation (GAT3850 and GAT4058) to YZ, AS, and PEL; Sainsbury Wellcome Centre Core Grant from Wellcome (219627/Z/19/Z) to AS; Schmidt Science Polymath Award to AS. AS is a CIFAR Azrieli Global Scholar in the Learning in Machines & Brains program.

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

# Table of Contents

# Appendix

## A  ADDITIONAL RELATED WORK

### A.1  SADDLE-TO-SADDLE DYNAMICS

Though saddle-to-saddle dynamics is a recurring phenomenon in the theory literature on learning dynamics, many questions remain open. The only case in which saddle-to-saddle dynamics has been proven for the full trajectory is the diagonal linear network in the limit of small initialization (Berthier, 2023; 2026; Pesme & Flammarion, 2023). For fully-connected linear networks, saddle-to-saddle dynamics has been shown under white input covariance and small spectral initialization (Saxe et al., 2014; 2019; Gidel et al., 2019; Gissin et al., 2020; Li et al., 2021). Outside this setting, the phase of visiting the first saddle from small initialization is well understood in linear networks (Jacot et al., 2022), while visiting subsequent saddles is not. For linear self-attention, Zhang et al. (2025b) showed saddle-to-saddle dynamics, but several phenomena remain unexplained, such as why it occurs even when the eigenvalues thought to govern the duration of plateaus are equal, and how increasing the number of heads affects the dynamics. While our work does not provide rigorous proofs for the technical open questions, we offer explanatory insights into these phenomena, and generate novel predictions on how network width, data statistics, and initialization affect saddle-to-saddle dynamics (Section 6).

There is also some work that did not focus on saddle-to-saddle dynamics but included relevant analysis. Zhang et al. (2024) showed that multimodal deep linear networks exhibit saddle-to-saddle dynamics. In multimodal deep linear networks with two input modalities, the saddle corresponds to a unimodal solution that is learning to fit the output only using one of the faster-to-learn modality. Rubruck et al. (2025) showed that two-layer linear networks with bias terms in the first layer exhibit saddle-to-saddle dynamics. Under some conditions, the first saddle corresponds to learning an optimal constant solution (Kang et al., 2024) that is the mean of the target output regardless of input.

Phenomena related to the timescale separation between directions in Theorem 4 have been examined in prior studies and are often discussed as weight alignment (Ji & Telgarsky, 2019; 2020; Atanasov et al., 2022). In a seminal work on the learning dynamics of deep linear networks (Saxe et al., 2014), aligned weights were introduced as an ansatz, with the analysis assuming aligned initial weights rather than deriving alignment from small isotropic initialization. This ansatz is often referred to in the linear network literature as the "spectral initialization" assumption; see Table 1 of Tarmoun et al. (2021) for a list of common initialization assumptions for linear networks. Later, Atanasov et al. (2022) analyzed the early phase dynamics of two-layer linear networks with scalar output from small isotropic initialization. They showed that the weights become increasingly aligned with a rank-one direction in the early phase, coined as "silent alignment". Our Theorem 4 extends the "silent alignment" to the vector output case, and recovers it when the output is scalar. A useful note is that fully-connected linear networks with scalar output or scalar input do not have nonzero saddles, since a width-one network already has full expressivity. Consequently, they do not exhibit saddle-to-saddle dynamics except for escaping the saddle at zero.

Hu et al. (2020) showed that two-layer nonlinear networks from a particular symmetric initialization have similar dynamics to a linear model on the input in the early phase of training. Because the network outputs zero for any input at their symmetric initialization, their analysis is related to our analysis of linear network near small initialization (Section 5.1). Our Theorem 4 also leverages the fact that the network output is close to zero and the dynamics is approximately linear in the weights.

Kunin et al. (2025) proposed an algorithmic framework to capture staircase learning curves in two-layer networks. We go beyond the algorithmic level by presenting a theoretical framework that analyzes embedded fixed points, invariant manifolds, and two different timescale separation mechanisms. We complement their work by addressing the question of why gradient descent dynamics behaves similarly to their algorithm. Further, our results on fixed points and invariant manifolds are not limited to two-layer networks; they apply to deep networks defined by Equation (1). We also disentangle data-induced and initialization-induced saddle-to-saddle dynamics, and show that the former leads to low-rank weights while the latter leads to sparse weights. These two mechanisms were not distinguished in prior literature.

Ziyin et al. (2025) proposed a hypothesis that learning dynamics is generally symmetry-to-symmetry. The saddle-to-saddle dynamics in our work is related to permutation symmetry between the units. Our work makes a theoretical case for their hypothesis and identifies the conditions under which saddle-to-saddle dynamics occurs.

## A.2   INCREMENTAL LEARNING IN OTHER SETTINGS

Incremental learning, characterized by earlier phases corresponding to simpler solutions, has been examined in several other theoretical settings. Cao et al. (2021); Ghosh et al. (2022) studied the spectral bias in the neural tangent kernel regime. In the kernel regime, eigenfunctions with larger eigenvalues are learned faster. This behavior differs from saddle-to-saddle dynamics, as the network in the kernel regime neither visits saddles nor exhibits plateaus during learning. Instead, the training loss decreases throughout learning, with faster decay early in learning and slower decay later. Outside the kernel regime, Abbe et al. (2023) studied a layer-wise training setup, in which the first and second layers are trained separately. Marion & Berthier (2023); Berthier et al. (2024) studied a two-timescale regime, in which the second-layer weights are trained with a much larger learning rate than the first-layer weights. In comparison, our analysis focuses on the learning dynamics of standard gradient descent outside the neural tangent kernel regime.

## A.3   SIMPLICITY BIAS

In this paper, we focus on the dynamical simplicity bias; that is, learning increasingly complex solutions over the course of training. A broader, longstanding body of theoretical and experimental research has explored the "stationary" simplicity bias, independent of training dynamics. Early studies (Hinton & van Camp, 1993; Hochreiter & Schmidhuber, 1997) connected generalization to the minimum description length of the weights, suggesting that flat minima correspond to simple solutions that potentially generalize well. A line of work based on algorithmic information theory (Schmidhuber, 1997; Valle-Perez et al., 2019; Dingle et al., 2018; Mingard et al., 2021; 2025; Goldblum et al., 2024; Teney et al., 2024) showed that standard architectures with randomly sampled weights are biased toward input-output maps with low Kolmogorov complexity. Empirical work (Huh et al., 2023) documented that both randomly initialized networks and trained networks exhibit a bias toward input-output maps with low effective rank embeddings. These findings motivated a volume hypothesis: neural networks have a simplicity bias arising from the loss landscape; specifically, the simple solutions occupy a larger volume in the weight space than the complex ones (Huh et al., 2023; Chiang et al., 2023). Our work on dynamical simplicity bias complements the stationary simplicity bias literature by examining how gradient descent dynamics drives the progression of solution complexity over time.

According to the no free lunch theorem, no single inductive bias is universally beneficial. Thus, the simplicity bias can be either advantageous or detrimental depending on the task. Several studies (Hermann & Lampinen, 2020; Shah et al., 2020; Petrini et al., 2022; Yang et al., 2024) have shown that favoring simple solutions may harm generalization when the simple solution relies on simple but spurious features, whereas more complex but robust features yield better generalization. For example, convolutional neural networks often prefer to classify objects by texture rather than shape, even though the classifier relying on shape can generalize better (Geirhos et al., 2019). Moreover, studies on two-layer ReLU networks (Holzmüller & Steinwart, 2022; Tsoy & Konstantinov, 2024; Boursier & Flammarion, 2025a;b) demonstrated that simplicity bias can sometimes cause optimization difficulties, where the first-layer weights align with a limited set of spurious directions when omnidirectional weights are required to reach global minima.

# B  ADDITIONAL FIGURES

## B.1  LEARNING DYNAMICS IN TWO-LAYER NETWORKS

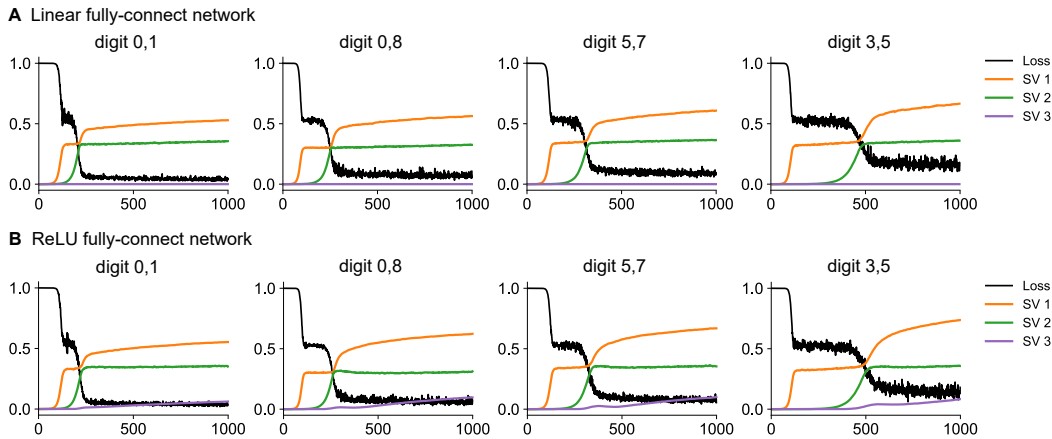

Figure 3: Saddle-to-saddle dynamics in two-layer fully-connected linear and ReLU networks trained for binary classification of MNIST digits. The input dimension is $28 \times 28 = 784$, the hidden layer width is 1000, and the target outputs are two-dimensional one-hot vectors. The intermediate plateau is longer when the two digits are harder to distinguish. For example, digits 3/5 are harder to distinguish than digits 0/1. The colored curves represent the top three singular values of the first-layer weight matrix, $\boldsymbol{U} \in \mathbb{R}^{1000 \times 784}$. Consistent with our theory, the growth of the first and second singular values coincides with the first and second abrupt drops in the training loss, respectively, corresponding to the increase in the effective width. The third largest singular value is close to zero, meaning the rank of the first-layer weight matrix is at most two, approximately. Details: The batch size is 64. The learning rate is 0.01. The initial weights are sampled independently from $\mathcal{N}(0, 10^{-12})$.

Table 1: Singular values of MNIST binary classification data.

| Digit pair | 0,1 | 0,8 | 5,7 | 3,5 |
|---|---|---|---|---|
| Singular value governing the first plateau | 4.20 | 5.21 | 4.13 | 4.62 |
| Singular value governing the second plateau | 3.90 | 5.21 | 3.26 | 1.38 |

In Figure 3, we show the learning dynamics of two-layer linear and ReLU networks trained for binary classification of MNIST digits. Despite being noisier than the learning dynamics on synthetic datasets, the plateaus and abrupt drops in the training loss are still pronounced. The abrupt drop in loss also coincides with the growth of a singular value of the first-layer weight matrix, implying that an increase in the effective width can explain the observed dynamics. We show the singular values governing the duration of the first and second plateaus of the linear network dynamics in Table 1, which are the first singular values of $\boldsymbol{\Sigma}_{yx}$ and $(\boldsymbol{I} - \boldsymbol{e}_1 \boldsymbol{e}_1^\top)\boldsymbol{\Sigma}_{yx}$, respectively. Here $\boldsymbol{e}_1$ is the first eigenvector of the matrix $\boldsymbol{\Sigma}_{yx}\boldsymbol{\Sigma}_{xx}^{-1}\boldsymbol{\Sigma}_{yx}^\top$; see Lemma 6. The sizes of the singular values approximately match the duration of the plateaus in Figure 3.

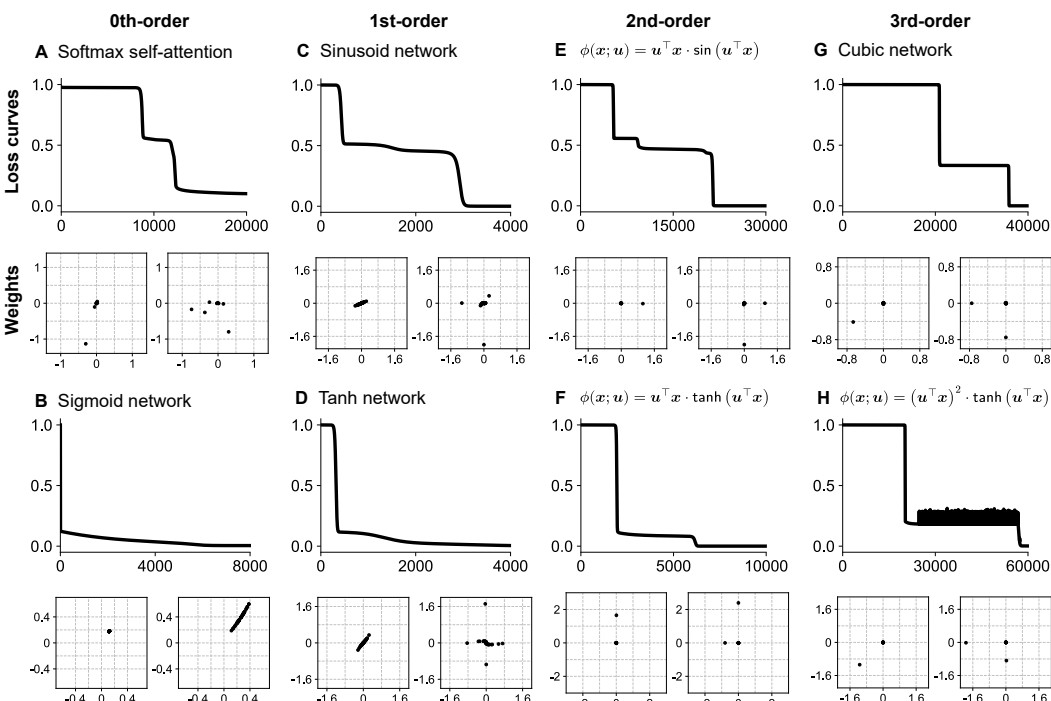

Figure 4: Learning dynamics in two-layer networks with other activation functions. Each panel shows the loss during training (top), and the first-layer weights right after the first abrupt loss drop (bottom left) and at the end of learning (bottom right). The first-layer weights to each hidden unit are two-dimensional and plotted as black dots. (A) The softmax self-attention model is the same as linear self-attention in Figure 1F, except for adding the softmax activation function. The training data set is the same as that of Figure 1F. (B) Two-layer fully-connected sigmoid network, i.e., $\phi(\boldsymbol{x}; \boldsymbol{u}) = \mathsf{sigmoid}\left(\boldsymbol{u}^{\top}\boldsymbol{x}\right)$. (C) Two-layer fully-connected sinusoid network, i.e., $\phi(\boldsymbol{x}; \boldsymbol{u}) = \sin\left(\boldsymbol{u}^{\top}\boldsymbol{x}\right)$. (D) Two-layer fully-connected tanh network, i.e., $\phi(\boldsymbol{x}; \boldsymbol{u}) = \tanh\left(\boldsymbol{u}^{\top}\boldsymbol{x}\right)$. (E,F,H) Two-layer fully-connected networks with the given activation functions. (G) Two-layer fully-connected cubic network, i.e., $\phi(\boldsymbol{x}; \boldsymbol{u}) = \left(\boldsymbol{u}^{\top}\boldsymbol{x}\right)^{3}$. Details: Except for panel A, the training set is generated by a width-2 teacher network with the same activation function, $y = \phi(\boldsymbol{x}; \boldsymbol{u}_1^*) + \phi(\boldsymbol{x}; \boldsymbol{u}_2^*), \boldsymbol{x} \in \mathbb{R}^2, y \in \mathbb{R}$. The input is sampled from $\mathcal{N}(\boldsymbol{0}, \boldsymbol{I})$. For panels B-F, the teacher network has $\boldsymbol{u}_1^* = [1, 0]^{\top}, \boldsymbol{u}_2^* = [0, 2]^{\top}$. For panels G,H, the teacher network has $\boldsymbol{u}_1^* = [1, 0]^{\top}, \boldsymbol{u}_2^* = [0, 1]^{\top}$. The number of training samples is 8192. The learning rate is 0.02. The initial weights are sampled independently from $\mathcal{N}(0, 10^{-12})$ for panels A-D, and from $\mathcal{N}(0, 0.005^2)$ for panels E-H. The width is $H = 50$ for panels A-D, and $H = 10$ for panels E-H.

In Figure 4, we plot the loss and weights dynamics for two-layer networks with other activation functions. If we Taylor expand $\phi(\boldsymbol{x}; \boldsymbol{u})$ around $\boldsymbol{u} = \boldsymbol{0}$, the lowest-order non-vanishing terms are the 0th-order, 1st-order, 2nd-order, and 3rd-order terms for panels (A,B), (C,D), (E,F), and (G,H), respectively. In panels (C,D), the networks develop rank-one weights in the early dynamics, since the sinusoid and tanh activation functions are approximately linear around zero. After the first abrupt drop in loss, our theory cannot predict the dynamics anymore because rank-one weights do not generally correspond to embedded fixed points or invariant manifolds for sinusoid and tanh networks. In panels (E,F,G,H), the networks undergo saddle-to-saddle dynamics, similar to the quadratic networks studied in Section 5.2. This is because dynamics with the 2nd-order and 3rd-order terms exhibit a timescale separation between units, as discussed at the end of Section 5. Indeed, the weights dynamics in panels (E,F,G,H) show that there are one and two units with large weights with the rest being near zero during the intermediate plateau and at convergence, respectively.

## B.2 LEARNING DYNAMICS IN DEEP NETWORKS

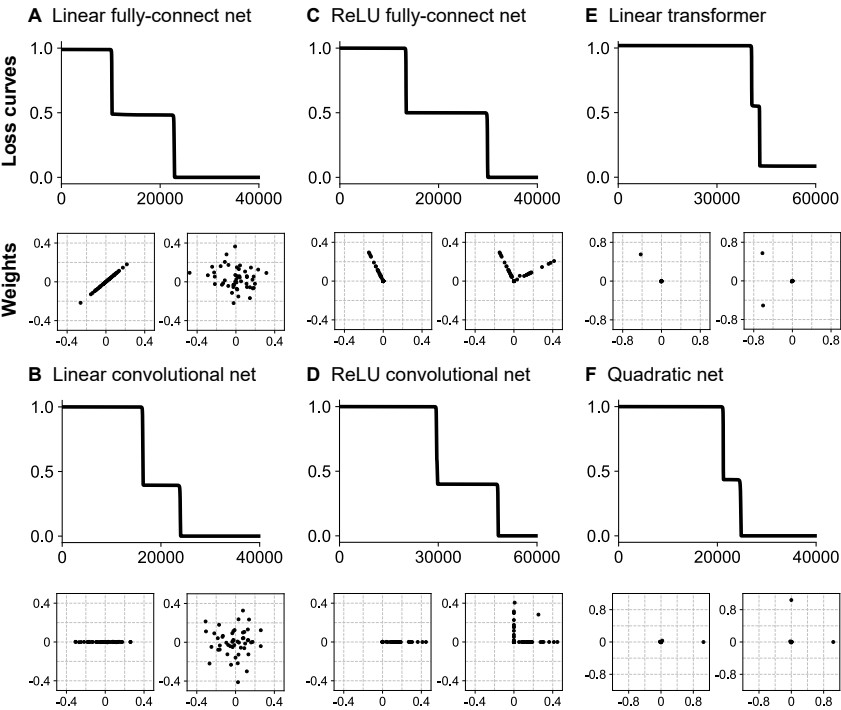

Figure 5: Learning dynamics in deep networks. Each panel shows the loss over training time (top), and the first-layer weights right after the first abrupt loss drop (bottom left) and at the end of learning (bottom right). The first-layer weights to each hidden unit are two-dimensional and plotted as black dots. The training sets in panels A,C,E are the same as those in Figure 1(B,D,F). The training sets in panels B,D,F split the scalar output in Figure 1(C,E,G) into a two-dimensional vector output. (A) Three-layer linear fully-connected network. (B) The network has a convolutional linear layer as the first hidden layer and a fully-connected linear layer as the second hidden layer. (C) Three-layer ReLU fully-connected network. (D) The network has a convolutional ReLU layer as the first hidden layer and a fully-connected ReLU layer as the second hidden layer. (E) One-layer linear transformer, consisting of one linear self-attention layer and two fully-connected linear layers. (F) The network has a fully-connected layer with quadratic activation as the first hidden layer and a fully-connected linear layer as the second hidden layer. Details: The number of training samples is 8192. The learning rate is 0.02. The initial weights are sampled independently from $\mathcal{N}(0, 0.005^2)$ for panels A-E, and from $\mathcal{N}(0, 0.05^2)$ for panel F. The width is $H = 50$ for panels A-D, and $H = 10$ for panels E,F.

In Figure 5, we present the learning dynamics of deep networks with various architectures, in comparison with the two-layer architectures in Figure 1. Similar to two-layer networks, the deep networks also exhibit saddle-to-saddle dynamics. The weight structures indicate that the visited saddles in panels (A,B) correspond to the embedded fixed points in Equation (7), those in panels (C,D) correspond to Equation (6), and those in panels (E,F) correspond to Equation (5). The effective width of the first layer during the intermediate plateau and at convergence is one and two, respectively.

We let the output of the networks in Figure 5(B,D,F) be two-dimensional, rather than one-dimensional as in Figure 1(C,E,G), due to considerations of expressivity. If the output is one-dimensional, the second fully-connected linear layer can achieve full expressivity with an effective width of one. This would make the second saddle-to-saddle transition (if there is one) different from the first: in the first transition, the effective width of both layers increases by one, whereas in the second transition only the effective width of the first layer increases. We leave this interesting problem to future research.

## B.3 THE EFFECT OF SKIP CONNECTION

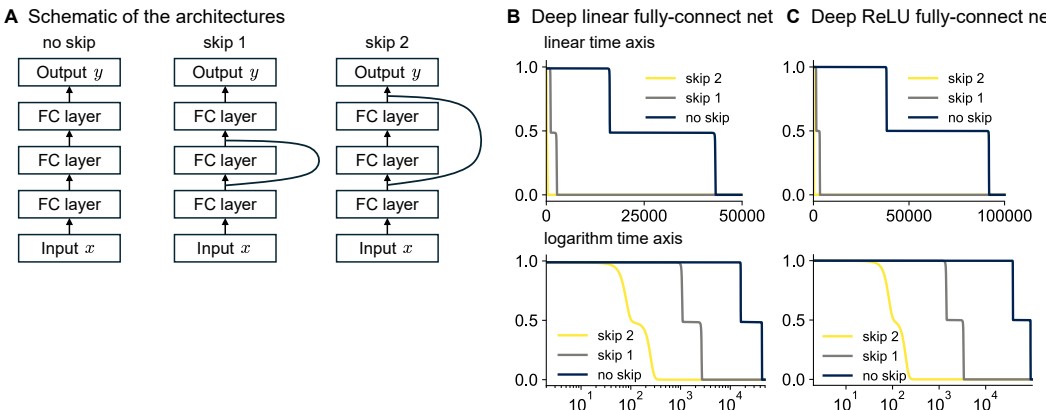

Figure 6: Saddle-to-saddle dynamics in deep fully-connected networks with skip connections. (A) Schematic of three four-layer fully-connected networks: one with no skip connection, one with a skip connection that skips one layer, and one with a skip connection that skips two layers. The three linear networks are defined in Equation (18). (B,C) Loss curves of linear and ReLU networks with skip connections, plotted using linear time (top row) and logarithmic time (bottom row) axes. All networks exhibit saddle-to-saddle dynamics, with the network that skips more layers learning faster. With small initialization, shallower linear networks learn faster (Saxe et al., 2019). In the network without a skip connection, all four layers must escape the zero fixed point and learn. In the network that skips one layer, the second-layer weights can remain near zero while the other three layers escape the zero fixed point and learn, yielding dynamics similar to a three-layer network. In the network that skips two layers, only the first and last layers need to learn, yielding dynamics similar to a two-layer network.

In Figure 6, the networks are four-layer linear networks with skip connections, defined as

$$\text{no skip: } f(\boldsymbol{x}) = \boldsymbol{W}_4\boldsymbol{W}_3\boldsymbol{W}_2\boldsymbol{W}_1\boldsymbol{x} \tag{18a}$$

$$\text{skip 1: } f(\boldsymbol{x}) = \boldsymbol{W}_4\boldsymbol{W}_3(\boldsymbol{W}_2\boldsymbol{W}_1\boldsymbol{x} + \boldsymbol{W}_1\boldsymbol{x}) \tag{18b}$$

$$\text{skip 2: } f(\boldsymbol{x}) = \boldsymbol{W}_4(\boldsymbol{W}_3\boldsymbol{W}_2\boldsymbol{W}_1\boldsymbol{x} + \boldsymbol{W}_1\boldsymbol{x}) \tag{18c}$$

where the input $\boldsymbol{x} \in \mathbb{R}^2$ and the weights $\boldsymbol{W}_4 \in \mathbb{R}^{2\times50}, \boldsymbol{W}_3, \boldsymbol{W}_2 \in \mathbb{R}^{50\times50}, \boldsymbol{W}_1 \in \mathbb{R}^{50\times2}$.

All three networks defined in Equation (18) exhibit saddle-to-saddle dynamics when trained from small initialization, with the network that skips more layers learning faster. This is because weights in the skipped layers can remain near zero while weights in other layers escape the zero fixed point and learn. When the skipped layers are effectively unused, the network behaves like a shallower network consisting only of the unskipped layers and exhibits saddle-to-saddle dynamics. Furthermore, since shallower linear networks learn faster (Saxe et al., 2019), networks that skip more layer also learn faster. The dynamics in ReLU networks with skip connections is similar.

## C  ADDITIONAL DISCUSSION

**ReLU activation function**. Because the ReLU activation function is piece-wise linear, we conjecture that ReLU networks trained from small initialization have a timescale separation between different directions, similar to the mechanism in linear networks. Indeed, prior studies have found that the direction that maximizes the correlation with the output grows the fastest, that is the direction $\arg\max_{\|\boldsymbol{u}\|=1}\langle\text{ReLU}(\boldsymbol{u}^\top\boldsymbol{x})y\rangle$. This was known as quantizing (Maennel et al., 2018), condensing (Luo et al., 2021), random feature amplification (Frei et al., 2023a), and studied in many other theoretical works on the learning dynamics of ReLU networks (Le & Jegelka, 2022; Petrini et al., 2022; Timor et al., 2023; Kou et al., 2023; Glasgow, 2024; Min et al., 2024).

**Reduction of high-dimensional learning dynamics**. The invariant manifolds in Theorem 3 can also be useful for reducing high-dimensional learning dynamics. Whenever a network evolves near an invariant manifold during a learning phase, Theorem 3 suggests that the full learning dynamics may be well approximated by the dynamics of a narrower network. For example, if a width-2 homogeneous network has nearly proportional weights, $\boldsymbol{\theta}_1 \approx \gamma\boldsymbol{\theta}_2$, dynamics in $\boldsymbol{\theta}_1$ and $\boldsymbol{\theta}_2$ may be well approximated by lower-dimensional dynamics only in $\boldsymbol{\theta}_1$. Many prior analyses have successfully reduced high-dimensional learning dynamics for individual architectures using a similar idea, including deep linear networks (Saxe et al., 2014; 2019; Advani et al., 2020), ReLU networks (Phuong & Lampert, 2021; Sarussi et al., 2021; Lyu et al., 2021; Frei et al., 2023b; Tsoy & Konstantinov, 2024; Zhang et al., 2025a), and self-attention (Zhang et al., 2025b; Yüksel et al., 2025).

**Distributed or localized features**. Identifying which type of fixed points a network visits among Equations (4) to (7) may be of interest to representation learning (Hinton et al., 1986; Elhage et al., 2022) and pruning problems (LeCun et al., 1989; Frankle & Carbin, 2019; Gromov et al., 2025). The fixed points described by Equations (4), (6) and (7) correspond to networks with distributed, nonlocal, or polysemantic features, while fixed points described by Equation (5) correspond to networks with localized or monosemantic features. Networks with localized features and simplicity bias can be more easily pruned. In Section 5, we showed that the timescale separation between directions due to data distribution gives rise to distributed features, while the timescale separation between units gives rise to localized features.

**Technical future directions**. The goal of this paper is to establish a theoretical framework and validate its predictive power. To serve this goal, we have prioritized the completeness of the framework, at times relying on heuristics and empirical observations. Several interesting technical questions remain open. First, how close must a point in weight space be to an invariant manifold in order to approach a fixed point on that manifold before leaving the manifold? Quantifying this could enable rigorous proofs of saddle-to-saddle dynamics in a wider range of architectures, extending beyond diagonal linear networks (Berthier, 2023; 2026; Pesme & Flammarion, 2023). Second, is the sequence of saddles visited during training Markovian? That is, can the next saddle be inferred solely from the current one, independent of earlier ones? In dynamical systems literature, it has been shown that certain saddle-to-saddle transitions are non-Markovian (Bakhtin, 2011).

## D  GRADIENT CALCULATIONS

We write down the gradients of the weights $\boldsymbol{\theta}_{1:H}$ for the network defined in Equation (1). We denote

$$f(\boldsymbol{x}) = g_{\text{out}}(\boldsymbol{\zeta}), \quad \text{where } \boldsymbol{\zeta} = \sum_{i=1}^{H} \phi(g_{\text{in}}(\boldsymbol{x}); \boldsymbol{u}_i)\boldsymbol{v}_i. \tag{19}$$

The variables have dimensionality

$$\boldsymbol{u} \in \mathbb{R}^{N_u}, \boldsymbol{v} \in \mathbb{R}^{N_v}, \phi(g_{\text{in}}(\boldsymbol{x}); \boldsymbol{u}) \in \mathbb{R}^{N_\phi \times N_v}, \boldsymbol{\zeta} \in \mathbb{R}^{N_\phi}.$$

Recall that the training loss is defined as

$$\mathcal{L} = \frac{1}{P} \sum_{\mu=1}^{P} \ell_\mu = \frac{1}{P} \sum_{\mu=1}^{P} \ell(\boldsymbol{y}_\mu, f(\boldsymbol{x}_\mu)). \tag{20}$$

Using the chain rule, the gradient flow dynamics of $\boldsymbol{v}_i$ can be written

$$\dot{\boldsymbol{v}}_i = -\frac{\partial \mathcal{L}}{\partial \boldsymbol{v}_i} = -\frac{1}{P} \sum_{\mu=1}^{P} \left( \frac{\partial \boldsymbol{\zeta}}{\partial \boldsymbol{v}_i} \right)^\top \frac{\partial \ell_\mu}{\partial \boldsymbol{\zeta}} = -\frac{1}{P} \sum_{\mu=1}^{P} \phi(g_{\text{in}}(\boldsymbol{x}_\mu); \boldsymbol{u}_i)^\top \frac{\partial \ell_\mu}{\partial \boldsymbol{\zeta}}. \tag{21}$$

The gradient flow dynamics of $\boldsymbol{u}_i$ involves matrix-by-vector derivatives, which generally require tensor notations. To avoid introducing tensor notations, we instead write the gradient entrywise, for $n = 1, \cdots, N_u$,

$$\dot{u}_{i,n} = -\frac{\partial \mathcal{L}}{\partial u_{i,n}} = -\frac{1}{P} \sum_{\mu=1}^{P} \left( \frac{\partial \boldsymbol{\zeta}}{\partial u_{i,n}} \right)^\top \frac{\partial \ell_\mu}{\partial \boldsymbol{\zeta}} = -\frac{1}{P} \sum_{\mu=1}^{P} \boldsymbol{v}_i^\top \frac{\partial \phi(g_{\text{in}}(\boldsymbol{x}_\mu); \boldsymbol{u}_i)^\top}{\partial u_{i,n}} \frac{\partial \ell_\mu}{\partial \boldsymbol{\zeta}}. \tag{22}$$

# E EMBEDDED FIXED POINTS

Here we prove Theorem 1.

*Proof.* We have that $\boldsymbol{\theta}^*_{1:(H-1)}$ is a fixed point of the gradient flow dynamics of the width-$(H-1)$ network, that is

$$\dot{\boldsymbol{\theta}}_i = -\frac{\partial \mathcal{L}}{\partial \boldsymbol{\theta}_i}\Big|_{\boldsymbol{\theta}_i=\boldsymbol{\theta}_i^*} = -\frac{\partial \mathcal{L}}{\partial f^*(\boldsymbol{x})}\frac{\partial f^*(\boldsymbol{x})}{\partial \boldsymbol{\theta}_i^*} = \boldsymbol{0}, \quad i=1,\cdots,H-1. \tag{23}$$

We denote

$$\boldsymbol{\zeta}^* = \sum_{j=1}^{H-1} \phi(g_{\text{in}}(\boldsymbol{x}); \boldsymbol{u}_j^*)\boldsymbol{v}_j^*. \tag{24}$$

The input-output map the width-$(H-1)$ network is $f^*(\boldsymbol{x}) = g_{\text{out}}(\boldsymbol{\zeta}^*)$.

We prove the four statements one by one.

(i) For any $\phi$, we analyze Equation (4).

First, the width-$H$ network implements the same input-output map as the width-$(H-1)$ network, that is $f(\boldsymbol{x}) = g_{\text{out}}(\boldsymbol{\zeta})$ where

$$\begin{aligned}
\boldsymbol{\zeta} &= \sum_{j=1}^{H} \phi(g_{\text{in}}(\boldsymbol{x}); \boldsymbol{u}_j)\boldsymbol{v}_j \\
&= \phi(g_{\text{in}}(\boldsymbol{x}); \boldsymbol{u}_H)\boldsymbol{v}_H + \phi(g_{\text{in}}(\boldsymbol{x}); \boldsymbol{u}_i)\boldsymbol{v}_i + \sum_{j=1,j\neq i}^{H-1} \phi(g_{\text{in}}(\boldsymbol{x}); \boldsymbol{u}_j)\boldsymbol{v}_j \\
&= \phi(g_{\text{in}}(\boldsymbol{x}); \boldsymbol{u}_i^*)\gamma_v\boldsymbol{v}_i^* + \phi(g_{\text{in}}(\boldsymbol{x}); \boldsymbol{u}_i^*)(1-\gamma_v)\boldsymbol{v}_i^* + \sum_{j=1,j\neq i}^{H-1} \phi(g_{\text{in}}(\boldsymbol{x}); \boldsymbol{u}_j^*)\boldsymbol{v}_j^* \\
&= \sum_{j=1}^{H-1} \phi(g_{\text{in}}(\boldsymbol{x}); \boldsymbol{u}_j^*)\boldsymbol{v}_j^* \\
&= \boldsymbol{\zeta}^*.
\end{aligned}$$

Second, we calculate the gradients of the weights in the width-$H$ network. For the units with unmodified weights, the gradients are the same as the width-$(H-1)$ network

$$\dot{\boldsymbol{\theta}}_j = -\frac{\partial \mathcal{L}}{\partial f(\boldsymbol{x})}\frac{\partial f(\boldsymbol{x})}{\partial \boldsymbol{\theta}_j} = -\frac{\partial \mathcal{L}}{\partial f^*(\boldsymbol{x})}\frac{\partial f^*(\boldsymbol{x})}{\partial \boldsymbol{\theta}_j^*} = \boldsymbol{0}, \quad j=1,\cdots,H-1,\, j\neq i.$$

For the $i$-th unit, the gradients can be expressed using Equations (21) and (22) as

$$\dot{\boldsymbol{v}}_i = -\frac{1}{P}\sum_{\mu=1}^{P} \phi(g_{\text{in}}(\boldsymbol{x}_\mu); \boldsymbol{u}_i)^\top \frac{\partial \ell_\mu}{\partial \boldsymbol{\zeta}} = -\frac{1}{P}\sum_{\mu=1}^{P} \phi(g_{\text{in}}(\boldsymbol{x}_\mu); \boldsymbol{u}_i^*)^\top \frac{\partial \ell_\mu}{\partial \boldsymbol{\zeta}^*} = \dot{\boldsymbol{v}}_i^* = \boldsymbol{0},$$

$$\begin{aligned}
\dot{u}_{i,n} &= -\frac{1}{P}\sum_{\mu=1}^{P} \boldsymbol{v}_i^\top \frac{\partial \phi(g_{\text{in}}(\boldsymbol{x}_\mu); \boldsymbol{u}_i)^\top}{\partial u_{i,n}} \frac{\partial \ell_\mu}{\partial \boldsymbol{\zeta}} \\
&= -(1-\gamma_v)\frac{1}{P}\sum_{\mu=1}^{P} \boldsymbol{v}_i^{*\top} \frac{\partial \phi(g_{\text{in}}(\boldsymbol{x}_\mu); \boldsymbol{u}_i^*)^\top}{\partial u_{i,n}} \frac{\partial \ell_\mu}{\partial \boldsymbol{\zeta}^*} \\
&= (1-\gamma_v)\dot{u}_{i,n}^* \\
&= 0, \quad n=1,\cdots,N_u.
\end{aligned}$$

Similarly, for the new $H$-th unit, the gradients are

$$\dot{v}_H = -\frac{1}{P}\sum_{\mu=1}^{P}\phi(g_{\text{in}}(x_\mu); u_i^*)^\top \frac{\partial \ell_\mu}{\partial \zeta^*} = \dot{v}_i^* = \mathbf{0},$$

$$\dot{u}_{H,n} = -\gamma_v \frac{1}{P}\sum_{\mu=1}^{P} v_i^{*\top} \frac{\partial \phi(g_{\text{in}}(x_\mu); u_i^*)^\top}{\partial u_{i,n}} \frac{\partial \ell_\mu}{\partial \zeta^*} = \gamma_v \dot{u}_{i,n}^* = 0, \quad n = 1, \cdots, N_u.$$

(ii) For $\phi$ such that $\forall z, \phi(z; u_{\text{zero}}) = 0$, we analyze Equation (5).

The width-$H$ network implements the same input-output map as the width-$(H-1)$ network, that is $f(x) = g_{\text{out}}(\zeta)$ where

$$\zeta = \sum_{i=1}^{H}\phi(g_{\text{in}}(x); u_i)v_i = \phi(g_{\text{in}}(x); u_{\text{zero}})\mathbf{0} + \sum_{j=1}^{H-1}\phi(g_{\text{in}}(x); u_j^*)v_j^* = \zeta^*.$$

Because the first $(H-1)$ units have unmodified weights, their gradients are the same as those in the width-$(H-1)$ network, which are zero. For the new $H$-th unit, the gradients can be expressed using Equations (21) and (22) as

$$\dot{v}_H = -\frac{1}{P}\sum_{\mu=1}^{P}\phi(g_{\text{in}}(x_\mu); u_{\text{zero}})^\top \frac{\partial \ell_\mu}{\partial \zeta} = \mathbf{0},$$

$$\dot{u}_{H,n} = -\frac{1}{P}\sum_{\mu=1}^{P}\mathbf{0}^\top \frac{\partial \phi(g_{\text{in}}(x_\mu); u_H)^\top}{\partial u_{H,n}} \frac{\partial \ell_\mu}{\partial \zeta} = 0, \quad n = 1, \cdots, N_u.$$

(iii) If $\phi(z; u)$ is degree-1 homogeneous in $u$, we analyze Equation (6).

The width-$H$ network implements the same input-output map as the width-$(H-1)$ network, that is $f(x) = g_{\text{out}}(\zeta)$ where

$$\zeta = \sum_{j=1}^{H}\phi(g_{\text{in}}(x); u_j)v_j$$

$$= \phi(g_{\text{in}}(x); \gamma_u u_i^*)\gamma_v v_i^* + \phi(g_{\text{in}}(x); u_i^*)(1 - \gamma_u \gamma_v)v_i^* + \sum_{j=1,j\neq i}^{H-1}\phi(g_{\text{in}}(x); u_j^*)v_j^*$$

$$= \phi(g_{\text{in}}(x); u_i^*)\gamma_u \gamma_v v_i^* + \phi(g_{\text{in}}(x); u_i^*)(1 - \gamma_u \gamma_v)v_i^* + \sum_{j=1,j\neq i}^{H-1}\phi(g_{\text{in}}(x); u_j^*)v_j^*$$

$$= \sum_{j=1}^{H-1}\phi(g_{\text{in}}(x); u_j^*)v_j^*$$

$$= \zeta^*.$$

Because the units with indices $j = 1, \cdots, H-1, j \neq i$ have unmodified weights, their gradients are the same as those in the width-$(H-1)$ network, which are zero. For the $i$-th

unit, the gradients can be expressed using Equations (21) and (22) as

$$\dot{\boldsymbol{v}}_i = -\frac{1}{P}\sum_{\mu=1}^{P}\phi(g_{\text{in}}(\boldsymbol{x}_\mu);\boldsymbol{u}_i^*)^\top\frac{\partial\ell_\mu}{\partial\boldsymbol{\zeta}^*} = \dot{\boldsymbol{v}}_i^* = \boldsymbol{0},$$

$$\dot{u}_{i,n} = -\frac{1}{P}\sum_{\mu=1}^{P}\boldsymbol{v}_i^\top\frac{\partial\phi(g_{\text{in}}(\boldsymbol{x}_\mu);\boldsymbol{u}_i)^\top}{\partial u_{i,n}}\frac{\partial\ell_\mu}{\partial\boldsymbol{\zeta}}$$

$$= -\frac{1}{P}\sum_{\mu=1}^{P}(1-\gamma_u\gamma_v)\boldsymbol{v}_i^{*\top}\frac{\partial\phi(g_{\text{in}}(\boldsymbol{x}_\mu);\boldsymbol{u}_i^*)^\top}{\partial u_{i,n}}\frac{\partial\ell_\mu}{\partial\boldsymbol{\zeta}^*}$$

$$= (1-\gamma_u\gamma_v)\dot{u}_{i,n}^*$$

$$= 0, \quad n = 1, \cdots, N_u.$$

By Euler's homogeneous function theorem, the partial derivative of the homogeneous function $\phi(\boldsymbol{z};\boldsymbol{u})$ has the following property

$$\frac{\partial\phi(\boldsymbol{z};\boldsymbol{u})}{\partial u_n}\bigg|_{\boldsymbol{u}=\gamma\boldsymbol{u}^*} = \frac{\partial\phi(\boldsymbol{z};\boldsymbol{u})}{\partial u_n}\bigg|_{\boldsymbol{u}=\boldsymbol{u}^*}, \quad n = 1, \cdots, N_u. \tag{25}$$

Thus, for the new $H$-th unit, the gradients are

$$\dot{\boldsymbol{v}}_H = -\frac{1}{P}\sum_{\mu=1}^{P}\phi(g_{\text{in}}(\boldsymbol{x}_\mu);\boldsymbol{u}_H)^\top\frac{\partial\ell_\mu}{\partial\boldsymbol{\zeta}} = -\frac{1}{P}\sum_{\mu=1}^{P}\phi(g_{\text{in}}(\boldsymbol{x}_\mu);\gamma_u\boldsymbol{u}_i^*)^\top\frac{\partial\ell_\mu}{\partial\boldsymbol{\zeta}^*} = \gamma_u\dot{\boldsymbol{v}}_i^* = \boldsymbol{0},$$

$$\dot{u}_{H,n} = -\frac{1}{P}\sum_{\mu=1}^{P}\boldsymbol{v}_H^\top\frac{\partial\phi(g_{\text{in}}(\boldsymbol{x}_\mu);\boldsymbol{u}_H)^\top}{\partial u_{H,n}}\frac{\partial\ell_\mu}{\partial\boldsymbol{\zeta}^*}$$

$$= -\frac{1}{P}\sum_{\mu=1}^{P}\gamma_v\boldsymbol{v}_i^{*\top}\frac{\partial\phi(g_{\text{in}}(\boldsymbol{x}_\mu);\gamma_u\boldsymbol{u}_i^*)^\top}{\partial u_{i,n}}\frac{\partial\ell_\mu}{\partial\boldsymbol{\zeta}^*}$$

$$= \gamma_v\dot{u}_{i,n}^* = 0, \quad n = 1, \cdots, N_u.$$

(iv) If $\phi(\boldsymbol{z};\boldsymbol{u})$ is linear in $\boldsymbol{u}$, we analyze Equation (7).

The width-$H$ network implements the same input-output map as the width-$(H-1)$ network, that is $f(\boldsymbol{x}) = g_{\text{out}}(\boldsymbol{\zeta})$ where

$$\boldsymbol{\zeta} = \sum_{j=1}^{H}\phi(g_{\text{in}}(\boldsymbol{x});\boldsymbol{u}_j)\boldsymbol{v}_j$$

$$= \phi(g_{\text{in}}(\boldsymbol{x});\boldsymbol{u}_H)\boldsymbol{v}_H + \sum_{j=1}^{H-1}\phi(g_{\text{in}}(\boldsymbol{x});\boldsymbol{u}_j)\boldsymbol{v}_j$$

$$= \phi\left(g_{\text{in}}(\boldsymbol{x});\sum_{i=1}^{H-1}\gamma_{u_i}\boldsymbol{u}_i^*\right)\left(\sum_{i=1}^{H-1}\gamma_{v_i}\boldsymbol{v}_i^*\right) + \sum_{j=1}^{H-1}\phi(g_{\text{in}}(\boldsymbol{x});\boldsymbol{u}_j^*)\left(\boldsymbol{v}_j^* - \gamma_{u_j}\sum_{j'=1}^{H-1}\gamma_{v_{j'}}\boldsymbol{v}_{j'}^*\right)$$

$$= \sum_{i,i'=1}^{H-1}\gamma_{u_i}\gamma_{v_{i'}}\phi(g_{\text{in}}(\boldsymbol{x});\boldsymbol{u}_i^*)\boldsymbol{v}_{i'}^* + \sum_{j=1}^{H-1}\phi(g_{\text{in}}(\boldsymbol{x});\boldsymbol{u}_j^*)\boldsymbol{v}_j^* - \sum_{j,j'=1}^{H-1}\gamma_{u_j}\gamma_{v_{j'}}\phi(g_{\text{in}}(\boldsymbol{x});\boldsymbol{u}_j^*)\boldsymbol{v}_{j'}^*$$

$$= \sum_{j=1}^{H-1}\phi(g_{\text{in}}(\boldsymbol{x});\boldsymbol{u}_j^*)\boldsymbol{v}_j^*$$

$$= \boldsymbol{\zeta}^*.$$

For $i = 1, \cdots, H - 1$, the gradients can be expressed using Equations (21) and (22) as

$$\dot{\boldsymbol{v}}_i = -\frac{1}{P} \sum_{\mu=1}^{P} \phi(g_{\text{in}}(\boldsymbol{x}_\mu); \boldsymbol{u}_i^*)^\top \frac{\partial \ell_\mu}{\partial \boldsymbol{\zeta}^*} = \dot{\boldsymbol{v}}_i^* = \boldsymbol{0},$$

$$\dot{u}_{i,n} = -\frac{1}{P} \sum_{\mu=1}^{P} \left( \boldsymbol{v}_i^* - \gamma_{u_i} \sum_{j=1}^{H-1} \gamma_{v_j} \boldsymbol{v}_j^* \right)^\top \frac{\partial \phi(g_{\text{in}}(\boldsymbol{x}_\mu); \boldsymbol{u}_i)^\top}{\partial u_{i,n}} \frac{\partial \ell_\mu}{\partial \boldsymbol{\zeta}}$$

$$= -\frac{1}{P} \sum_{\mu=1}^{P} \boldsymbol{v}_i^{*\top} \frac{\partial \phi(g_{\text{in}}(\boldsymbol{x}_\mu); \boldsymbol{u}_i)^\top}{\partial u_{i,n}} \frac{\partial \ell_\mu}{\partial \boldsymbol{\zeta}} - \gamma_{u_i} \sum_{j=1}^{H-1} \gamma_{v_j} \frac{1}{P} \sum_{\mu=1}^{P} \boldsymbol{v}_j^{*\top} \frac{\partial \phi(g_{\text{in}}(\boldsymbol{x}_\mu); \boldsymbol{u}_j)^\top}{\partial u_{j,n}} \frac{\partial \ell_\mu}{\partial \boldsymbol{\zeta}}$$

$$= \dot{u}_{i,n}^* + \gamma_{u_i} \sum_{j=1}^{H-1} \gamma_{v_j} \dot{u}_{j,n}^*$$

$$= 0, \quad n = 1, \cdots, N_u.$$

We leverage the degree-1 homogeneous and additive properties of linearity, which yields

$$\phi\left( \boldsymbol{z}; \sum_{i=1}^{H-1} \gamma_{u_i} \boldsymbol{u}_i^* \right) = \sum_{i=1}^{H-1} \gamma_{u_i} \phi(\boldsymbol{z}; \boldsymbol{u}_i^*) \tag{26}$$

Thus, for the new $H$-th unit, the gradient for $\boldsymbol{v}_H$ is

$$\dot{\boldsymbol{v}}_H = -\frac{1}{P} \sum_{\mu=1}^{P} \phi(g_{\text{in}}(\boldsymbol{x}_\mu); \boldsymbol{u}_H)^\top \frac{\partial \ell_\mu}{\partial \boldsymbol{\zeta}}$$

$$= -\frac{1}{P} \sum_{\mu=1}^{P} \phi\left( g_{\text{in}}(\boldsymbol{x}_\mu); \sum_{i=1}^{H-1} \gamma_{u_i} \boldsymbol{u}_i^* \right)^\top \frac{\partial \ell_\mu}{\partial \boldsymbol{\zeta}^*}$$

$$= \sum_{i=1}^{H-1} \gamma_{u_i} \left( -\frac{1}{P} \sum_{\mu=1}^{P} \phi(g_{\text{in}}(\boldsymbol{x}_\mu); \boldsymbol{u}_i^*)^\top \frac{\partial \ell_\mu}{\partial \boldsymbol{\zeta}^*} \right)$$

$$= \sum_{i=1}^{H-1} \gamma_{u_i} \dot{\boldsymbol{v}}_i^*$$

$$= \boldsymbol{0}.$$

For $\phi(\boldsymbol{z}; \boldsymbol{u})$ that is linear in $\boldsymbol{u}$, the partial derivative, $\frac{\partial \phi(\boldsymbol{z}; \boldsymbol{u})}{\partial u_n}$, is a function that does not involve $\boldsymbol{u}$. Thus, the gradient for $\boldsymbol{u}_H$ is

$$\dot{u}_{H,n} = -\frac{1}{P} \sum_{\mu=1}^{P} \boldsymbol{v}_H^\top \frac{\partial \phi(g_{\text{in}}(\boldsymbol{x}_\mu); \boldsymbol{u}_H)^\top}{\partial u_{H,n}} \frac{\partial \ell_\mu}{\partial \boldsymbol{\zeta}}$$

$$= \sum_{i=1}^{H-1} \gamma_{v_i} \left( -\frac{1}{P} \sum_{\mu=1}^{P} \boldsymbol{v}_i^{*\top} \frac{\partial \phi(g_{\text{in}}(\boldsymbol{x}_\mu); \boldsymbol{u}_H)^\top}{\partial u_{H,n}} \frac{\partial \ell_\mu}{\partial \boldsymbol{\zeta}} \right)$$

$$= \sum_{i=1}^{H-1} \gamma_{v_i} \dot{u}_{i,n}^*$$

$$= 0, \quad n = 1, \cdots, N_u.$$

Hence, in all the four cases, the weights of the width-$H$ network given by Theorem 1 satisfy that the input-output map is the same as the width-$(H-1)$ network and the gradients are zero. ∎

# F INVARIANT MANIFOLDS

## F.1 DEFINITION OF INVARIANT MANIFOLDS

We provide the definition of invariant manifolds used in this work.

**Definition 2** (Invariant manifold). A set $\mathcal{M} \subset \mathbb{R}^n$ is an invariant set under the dynamical system $\dot{\boldsymbol{\theta}} = h(\boldsymbol{\theta})$ if for any $\boldsymbol{\theta}(0) = \boldsymbol{\theta}_0 \in \mathcal{M}$ we have $\boldsymbol{\theta}(t) \in \mathcal{M}$ for all $t \in \mathbb{R}$. An invariant set $\mathcal{M} \subset \mathbb{R}^n$ is an invariant manifold if $\mathcal{M}$ has the structure of a differentiable manifold.

In this work, we focus on a particular class of invariant manifolds defined via the constraint $\Phi(\boldsymbol{\theta}) = 0$; see Theorem 3. For defining invariant manifolds in more generality, we refer the readers to Wiggins (2003).

If a trajectory on an invariant manifold connects two distinct fixed points as $t \to \pm\infty$, it is called a heteroclinic orbit in the dynamical systems literature (Bakhtin, 2011). The saddle-to-saddle transitions in our setup can be viewed as heteroclinic orbits.

## F.2 PROOF OF INVARIANT MANIFOLDS

We here prove Theorem 3.

*Proof.* We prove the four statements one by one. Recall that $\boldsymbol{\theta}_i$ is defined in Equation (1), which is the stacked second-layer and first-layer weights in the $i$-th unit

$$\boldsymbol{\theta}_i = \begin{bmatrix} \boldsymbol{v}_i \\ \boldsymbol{u}_i \end{bmatrix}.$$

(i) For any $\phi$, two units have equal weights: $\boldsymbol{\theta}_i = \boldsymbol{\theta}_j$.

We write down the dynamics of the difference $(\boldsymbol{\theta}_i - \boldsymbol{\theta}_j)$ and substitute in $\boldsymbol{\theta}_i = \boldsymbol{\theta}_j$. Using Equation (21), the dynamics of $(\boldsymbol{v}_i - \boldsymbol{v}_j)$ is given by

$$\frac{\mathrm{d}}{\mathrm{d}t}(\boldsymbol{v}_i - \boldsymbol{v}_j) = -\frac{1}{P} \sum_{\mu=1}^{P} \big(\phi(g_{\mathrm{in}}(\boldsymbol{x}_\mu); \boldsymbol{u}_i) - \phi(g_{\mathrm{in}}(\boldsymbol{x}_\mu); \boldsymbol{u}_j)\big)^\top \frac{\partial \ell_\mu}{\partial \boldsymbol{\zeta}} = \boldsymbol{0}.$$

Using Equation (22), the dynamics of $(u_{i,n} - u_{j,n})$ for $n = 1, \cdots, N_u$ is given by

$$\frac{\mathrm{d}}{\mathrm{d}t}(u_{i,n} - u_{j,n}) = -\frac{1}{P} \sum_{\mu=1}^{P} \left( \boldsymbol{v}_i^\top \frac{\partial \phi(g_{\mathrm{in}}(\boldsymbol{x}_\mu); \boldsymbol{u}_i)^\top}{\partial u_{i,n}} - \boldsymbol{v}_j^\top \frac{\partial \phi(g_{\mathrm{in}}(\boldsymbol{x}_\mu); \boldsymbol{u}_j)^\top}{\partial u_{j,n}} \right) \frac{\partial \ell_\mu}{\partial \boldsymbol{\zeta}} = 0.$$

(ii) If $\exists \boldsymbol{u}_{\mathrm{zero}}$ such that $\forall \boldsymbol{z}, \phi(\boldsymbol{z}; \boldsymbol{u}_{\mathrm{zero}}) = 0$, a unit has zero weights: $\boldsymbol{v}_i = \boldsymbol{0}, \boldsymbol{u}_i = \boldsymbol{u}_{\mathrm{zero}}$.

Substituting $\boldsymbol{v}_i = \boldsymbol{0}, \boldsymbol{u}_i = \boldsymbol{u}_{\mathrm{zero}}$ into Equations (21) and (22), we obtain

$$\dot{\boldsymbol{v}}_i = -\frac{1}{P} \sum_{\mu=1}^{P} \phi(g_{\mathrm{in}}(\boldsymbol{x}_\mu); \boldsymbol{u}_{\mathrm{zero}})^\top \frac{\partial \ell_\mu}{\partial \boldsymbol{\zeta}} = \boldsymbol{0},$$

$$\dot{u}_{i,n} = -\frac{1}{P} \sum_{\mu=1}^{P} \boldsymbol{0}^\top \frac{\partial \phi(g_{\mathrm{in}}(\boldsymbol{x}_\mu); \boldsymbol{u}_i)^\top}{\partial u_{i,n}} \frac{\partial \ell_\mu}{\partial \boldsymbol{\zeta}} = 0, \quad n = 1, \cdots, N_u.$$

(iii) If $\phi(\boldsymbol{z}; \boldsymbol{u})$ is homogeneous in $\boldsymbol{u}$, two units have proportional weights: $\boldsymbol{\theta}_i = \gamma \boldsymbol{\theta}_j, \gamma \in \mathbb{F}$.

Using Equation (21) and the degree-1 homogeneous property, $\forall \gamma \in \mathbb{F}$, $\phi(\boldsymbol{z}; \gamma \boldsymbol{u}) = \gamma \phi(\boldsymbol{z}; \boldsymbol{u})$, the dynamics of $(\boldsymbol{v}_i - \gamma \boldsymbol{v}_j)$ is given by

$$\frac{\mathrm{d}}{\mathrm{d}t}(\boldsymbol{v}_i - \gamma \boldsymbol{v}_j) = -\frac{1}{P} \sum_{\mu=1}^{P} \big(\phi(g_{\text{in}}(\boldsymbol{x}_\mu); \boldsymbol{u}_i) - \gamma \phi(g_{\text{in}}(\boldsymbol{x}_\mu); \boldsymbol{u}_j)\big)^\top \frac{\partial \ell_\mu}{\partial \boldsymbol{\zeta}}$$

$$= -\frac{1}{P} \sum_{\mu=1}^{P} \big(\phi(g_{\text{in}}(\boldsymbol{x}_\mu); \boldsymbol{u}_i) - \phi(g_{\text{in}}(\boldsymbol{x}_\mu); \gamma \boldsymbol{u}_j)\big)^\top \frac{\partial \ell_\mu}{\partial \boldsymbol{\zeta}}$$

$$= \boldsymbol{0}.$$

Using Equations (22) and (25), the dynamics of $(u_{i,n} - \gamma u_{j,n})$ for $n = 1, \cdots, N_u$ is given by

$$\frac{\mathrm{d}}{\mathrm{d}t}(u_{i,n} - \gamma u_{j,n}) = -\frac{1}{P} \sum_{\mu=1}^{P} \left(\boldsymbol{v}_i^\top \frac{\partial \phi(g_{\text{in}}(\boldsymbol{x}_\mu); \boldsymbol{u}_i)^\top}{\partial u_{i,n}} - \gamma \boldsymbol{v}_j^\top \frac{\partial \phi(g_{\text{in}}(\boldsymbol{x}_\mu); \boldsymbol{u}_j)^\top}{\partial u_{j,n}}\right) \frac{\partial \ell_\mu}{\partial \boldsymbol{\zeta}}$$

$$= -\frac{1}{P} \sum_{\mu=1}^{P} (\boldsymbol{v}_i - \gamma \boldsymbol{v}_j)^\top \frac{\partial \phi(g_{\text{in}}(\boldsymbol{x}_\mu); \boldsymbol{u}_i)^\top}{\partial u_{i,n}} \frac{\partial \ell_\mu}{\partial \boldsymbol{\zeta}}$$

$$= 0.$$

(iv) If $\phi(\boldsymbol{z}; \boldsymbol{u})$ is linear in $\boldsymbol{u}$, any number of units have linear dependence: $\boldsymbol{\theta}_i = \sum_{j \neq i} \gamma_j \boldsymbol{\theta}_j$.

We leverage the degree-1 homogeneous and additive properties of linearity, which yields

$$\phi(\boldsymbol{z}; \boldsymbol{u}_i) - \sum_{j \neq i} \gamma_j \phi(\boldsymbol{z}; \boldsymbol{u}_j) = \phi\left(\boldsymbol{z}; \boldsymbol{u}_i - \sum_{j \neq i} \gamma_j \boldsymbol{u}_j\right) \tag{27}$$

Using Equations (21) and (27), we obtain the dynamics of $\left(\boldsymbol{v}_i - \sum_{j \neq i} \gamma_j \boldsymbol{v}_j\right)$

$$\frac{\mathrm{d}}{\mathrm{d}t}\left(\boldsymbol{v}_i - \sum_{j \neq i} \gamma_j \boldsymbol{v}_j\right) = -\frac{1}{P} \sum_{\mu=1}^{P} \left(\phi(g_{\text{in}}(\boldsymbol{x}_\mu); \boldsymbol{u}_i) - \sum_{j \neq i} \gamma_j \phi(g_{\text{in}}(\boldsymbol{x}_\mu); \boldsymbol{u}_j)\right)^\top \frac{\partial \ell_\mu}{\partial \boldsymbol{\zeta}}$$

$$= -\frac{1}{P} \sum_{\mu=1}^{P} \phi\left(g_{\text{in}}(\boldsymbol{x}_\mu); \boldsymbol{u}_i - \sum_{j \neq i} \gamma_j \boldsymbol{u}_j\right)^\top \frac{\partial \ell_\mu}{\partial \boldsymbol{\zeta}}$$

$$= -\frac{1}{P} \sum_{\mu=1}^{P} \phi(g_{\text{in}}(\boldsymbol{x}_\mu); \boldsymbol{0})^\top \frac{\partial \ell_\mu}{\partial \boldsymbol{\zeta}}$$

$$= \boldsymbol{0}.$$

For $\phi(\boldsymbol{z}; \boldsymbol{u})$ that is linear in $\boldsymbol{u}$, the partial derivative $\frac{\partial \phi(\boldsymbol{z}; \boldsymbol{u})}{\partial u_n}$ is a function that does not involve $\boldsymbol{u}$. Thus, using Equation (22), we obtain the dynamics of $\left(u_{i,n} - \sum_{j \neq i} \gamma_j u_{j,n}\right)$ for $n = 1, \cdots, N_u$

$$\frac{\mathrm{d}}{\mathrm{d}t}\left(u_{i,n} - \sum_{j \neq i} \gamma_j u_{j,n}\right) = -\frac{1}{P} \sum_{\mu=1}^{P} \left(\boldsymbol{v}_i - \sum_{j \neq i} \gamma_j \boldsymbol{v}_j\right)^\top \frac{\partial \phi(g_{\text{in}}(\boldsymbol{x}_\mu); \boldsymbol{u}_i)^\top}{\partial u_{i,n}} \frac{\partial \ell_\mu}{\partial \boldsymbol{\zeta}} = 0.$$

∎

## F.3 EXPRESSIVITY ON INVARIANT MANIFOLDS

When the weights of a network lie on an invariant manifolds, its input-output map is expressible by the architecture with fewer units than its actual width.

(i) For any $\phi$, when two units have equal weights, $\boldsymbol{\theta}_i = \boldsymbol{\theta}_j$, we can remove the $i$-th unit and multiply $\boldsymbol{v}_j$ by 2, obtaining a network with one less unit that expresses the same input-output map.

(ii) For $\phi$ such that $\forall \boldsymbol{z}, \phi(\boldsymbol{z}; \boldsymbol{u}_{\text{zero}}) = 0$, when a unit has zero weights, $\boldsymbol{v}_i = \boldsymbol{0}, \boldsymbol{u}_i = \boldsymbol{u}_{\text{zero}}$, we can just remove this unit, obtaining a network with one less unit that expresses the same input-output map.

(iii) For $\phi(\boldsymbol{z}; \boldsymbol{u})$ that is degree-1 homogeneous in $\boldsymbol{u}$, when two units have proportional weights, $\boldsymbol{\theta}_i = \gamma \boldsymbol{\theta}_j$, we can remove the $i$-th unit and multiply $\boldsymbol{v}_j$ by $(1 + \gamma^2)$, obtaining a network with one less unit that expresses the same input-output map.

(iv) For $\phi(\boldsymbol{z}; \boldsymbol{u})$ that is linear in $\boldsymbol{u}$, when there is a linear dependence $\boldsymbol{\theta}_i = \sum_{j \neq i} \gamma_j \boldsymbol{\theta}_j$, we can remove the $i$-th unit and modify the second-layer weights in all remaining units as follows

$$\boldsymbol{v}_j^{\text{new}} = \boldsymbol{v}_j + \gamma_j \sum_{j' \neq i} \gamma_{j'} \boldsymbol{v}_{j'}. \tag{28}$$

This new width-$(H-1)$ network expresses the same input-output map as the original width-$H$ network

$$\sum_{j=1, j \neq i}^{H} \phi(g_{\text{in}}(\boldsymbol{x}); \boldsymbol{u}_j) \boldsymbol{v}_j^{\text{new}} = \sum_{j \neq i} \phi(g_{\text{in}}(\boldsymbol{x}); \boldsymbol{u}_j) \left( \boldsymbol{v}_j + \gamma_j \sum_{j' \neq i} \gamma_{j'} \boldsymbol{v}_{j'} \right)$$

$$= \sum_{j \neq i} \phi(g_{\text{in}}(\boldsymbol{x}); \boldsymbol{u}_j) \boldsymbol{v}_j + \phi \left( g_{\text{in}}(\boldsymbol{x}); \sum_{j \neq i} \gamma_j \boldsymbol{u}_j \right) \sum_{j' \neq i} \gamma_{j'} \boldsymbol{v}_{j'}$$

$$= \sum_{j \neq i} \phi(g_{\text{in}}(\boldsymbol{x}); \boldsymbol{u}_j) \boldsymbol{v}_j + \phi(g_{\text{in}}(\boldsymbol{x}); \boldsymbol{u}_i) \boldsymbol{v}_i$$

$$= \sum_{j=1}^{H} \phi(g_{\text{in}}(\boldsymbol{x}); \boldsymbol{u}_j) \boldsymbol{v}_j.$$

## F.4 THE EMBEDDED FIXED POINTS ON INVARIANT MANIFOLDS

The set of embedded fixed points lying on the invariant manifolds given in Theorem 3 is a subset of the embedded fixed points given in Theorem 1. Here we specify the subset of embedded fixed points that lie on invariant manifolds.

(i) When $\gamma_v = 1/2$ in Equation (4), the embedded fixed point has $\boldsymbol{u}_H = \boldsymbol{u}_i = \boldsymbol{u}_i^*, \boldsymbol{v}_H = \boldsymbol{v}_i = \boldsymbol{v}_i^*/2$, which is on the invariant manifold of $\boldsymbol{\theta}_H = \boldsymbol{\theta}_i$ in Theorem 3(i).

(ii) For $\phi$ such that $\forall \boldsymbol{z}, \phi(\boldsymbol{z}; \boldsymbol{u}_{\text{zero}}) = 0$, it is clear that $\boldsymbol{u}_H = \boldsymbol{u}_{\text{zero}}, \boldsymbol{v}_H = \boldsymbol{0}$ is on the invariant manifold in Theorem 3(ii).

(iii) For $\phi(\boldsymbol{z}; \boldsymbol{u})$ that is degree-1 homogeneous in $\boldsymbol{u}$, when $\gamma_v = \gamma_u/(1 + \gamma_u^2)$ in Equation (6), the embeded fixed point has

$$\boldsymbol{u}_H = \gamma_u \boldsymbol{u}_i, \; \boldsymbol{v}_H = \frac{\gamma_v}{1 - \gamma_u \gamma_v} \boldsymbol{v}_i = \gamma_u \boldsymbol{v}_i, \tag{29}$$

which is on the invariant manifold of $\boldsymbol{\theta}_H = \gamma_u \boldsymbol{\theta}_i$ in Theorem 3(iii).

(iv) For $\phi(\boldsymbol{z}; \boldsymbol{u})$ that is linear in $\boldsymbol{u}$, let us first rearrange Equation (7) by substituting $\boldsymbol{v}_i = \boldsymbol{v}_i^* - \gamma_{u_i} \boldsymbol{v}_H$ into $\boldsymbol{v}_H = \sum_{i=1}^{H-1} \gamma_{v_i} \boldsymbol{v}_i^*$ and obtaining an expression of $\boldsymbol{v}_H$ in terms of $\{\boldsymbol{v}_i\}_{i=1}^{H-1}$

$$\boldsymbol{v}_H = \sum_{i=1}^{H-1} \gamma_{v_i} (\boldsymbol{v}_i + \gamma_{u_i} \boldsymbol{v}_H) \quad \Rightarrow \quad \boldsymbol{v}_H = \frac{1}{1 - \sum_{j=1}^{H-1} \gamma_{v_j} \gamma_{u_j}} \sum_{i=1}^{H-1} \gamma_{v_i} \boldsymbol{v}_i. \tag{30}$$

When $\gamma_{v_i} = \gamma_{u_i}/(1 + \sum_{j=1}^{H-1} \gamma_{u_i}^2)$ in Equation (7), the embedded fixed point has

$$\boldsymbol{u}_H = \sum_{i=1}^{H-1} \gamma_{u_i} \boldsymbol{u}_i, \tag{31a}$$

$$\boldsymbol{v}_H = \frac{1}{1 - \sum_{j=1}^{H-1} \gamma_{v_j} \gamma_{u_j}} \sum_{i=1}^{H-1} \gamma_{v_i} \boldsymbol{v}_i$$

$$= \frac{1}{1 - \sum_{j=1}^{H-1} \frac{\gamma_{u_j}^2}{1 + \sum_{k=1}^{H-1} \gamma_{u_k}^2}} \sum_{i=1}^{H-1} \frac{\gamma_{u_i}}{1 + \sum_{j=1}^{H-1} \gamma_{u_j}^2} \boldsymbol{v}_i$$

$$= \sum_{i=1}^{H-1} \gamma_{u_i} \boldsymbol{v}_i, \tag{31b}$$

which is on the invariant manifold of $\boldsymbol{\theta}_H = \sum_{i=1}^{H-1} \gamma_{u_i} \boldsymbol{\theta}_i$ in Theorem 3(iv).

## G  DYNAMICS OF LINEAR NETWORKS

### G.1  GRADIENT FLOW EQUATIONS

We derive the gradient flow equations given in Equation (9).

For $i = 1, \cdots, H$, the gradient flow dynamics on squared loss, $\ell(\boldsymbol{y}, \hat{\boldsymbol{y}}) = \frac{1}{2}\|\boldsymbol{y} - \hat{\boldsymbol{y}}\|_2^2$, is given by

$$
\begin{aligned}
\dot{\boldsymbol{v}}_i &= -\frac{1}{P}\sum_{\mu=1}^{P}\frac{\partial \mathcal{L}}{\partial f(\boldsymbol{x}_\mu)}\frac{\partial f(\boldsymbol{x}_\mu)}{\partial \boldsymbol{v}_i} \\
&= \frac{1}{P}\sum_{\mu=1}^{P}(\boldsymbol{y}_\mu - \boldsymbol{W}\boldsymbol{z}_\mu)\boldsymbol{z}_\mu^\top \boldsymbol{u}_i \\
&= \left(\frac{1}{P}\sum_{\mu=1}^{P}\boldsymbol{y}_\mu \boldsymbol{z}_\mu^\top - \boldsymbol{W}\frac{1}{P}\sum_{\mu=1}^{P}\boldsymbol{z}_\mu \boldsymbol{z}_\mu^\top\right)\boldsymbol{u}_i, \\
\dot{\boldsymbol{u}}_i &= -\frac{1}{P}\sum_{\mu=1}^{P}\frac{\partial \mathcal{L}}{\partial f(\boldsymbol{x}_\mu)}\frac{\partial f(\boldsymbol{x}_\mu)}{\partial \boldsymbol{u}_i} \\
&= \frac{1}{P}\sum_{\mu=1}^{P}\boldsymbol{z}_\mu(\boldsymbol{y}_\mu - \boldsymbol{W}\boldsymbol{z}_\mu)^\top \boldsymbol{v}_i \\
&= \left(\frac{1}{P}\sum_{\mu=1}^{P}\boldsymbol{z}_\mu \boldsymbol{y}_\mu^\top - \frac{1}{P}\sum_{\mu=1}^{P}\boldsymbol{z}_\mu \boldsymbol{z}_\mu^\top \boldsymbol{W}^\top\right)\boldsymbol{v}_i.
\end{aligned}
$$

Recall that the data statistics are defined as

$$
\boldsymbol{\Sigma}_{yz} = \frac{1}{P}\sum_{\mu=1}^{P}\boldsymbol{y}_\mu \boldsymbol{z}_\mu^\top, \quad \boldsymbol{\Sigma}_{zz} = \frac{1}{P}\sum_{\mu=1}^{P}\boldsymbol{z}_\mu \boldsymbol{z}_\mu^\top.
$$

Substituting in the data statistics, we obtain the gradient flow equations in Equation (9), which are

$$
\dot{\boldsymbol{v}}_i = \left(\boldsymbol{\Sigma}_{yz} - \boldsymbol{W}\boldsymbol{\Sigma}_{zz}\right)\boldsymbol{u}_i, \quad \dot{\boldsymbol{u}}_i = \left(\boldsymbol{\Sigma}_{yz} - \boldsymbol{W}\boldsymbol{\Sigma}_{zz}\right)^\top \boldsymbol{v}_i, \quad i = 1, \cdots, H.
$$

### G.2  PROOF OF TIMESCALE SEPARATION

We here prove Theorem 4, that is the timescale separation between directions in a linear dynamical system.

*Proof.* The linear dynamical system in Equation (10) can be written as

$$
\dot{\boldsymbol{\theta}}_i = \boldsymbol{M}\boldsymbol{\theta}_i, \quad \text{where } \boldsymbol{M} = \begin{bmatrix} \boldsymbol{0} & \boldsymbol{\Sigma}_{yx} \\ \boldsymbol{\Sigma}_{yx}^\top & \boldsymbol{0} \end{bmatrix}, \boldsymbol{\theta}_i = \begin{bmatrix} \boldsymbol{v}_i \\ \boldsymbol{u}_i \end{bmatrix}. \tag{32}
$$

The symmetric matrix $\boldsymbol{M}$ has $D$ positive eigenvalues, $D$ negative eigenvalues, and $(N_v + N_u - 2D)$ zero eigenvalues. The nonzero eigenvalues are the singular values of $\boldsymbol{\Sigma}_{yx}$ and their negative

$$
\boldsymbol{M}\begin{bmatrix} \boldsymbol{q}_k \\ \boldsymbol{r}_k \end{bmatrix} = s_k \begin{bmatrix} \boldsymbol{q}_k \\ \boldsymbol{r}_k \end{bmatrix}, \quad \boldsymbol{M}\begin{bmatrix} \boldsymbol{q}_k \\ -\boldsymbol{r}_k \end{bmatrix} = -s_k \begin{bmatrix} \boldsymbol{q}_k \\ -\boldsymbol{r}_k \end{bmatrix}, \quad k = 1, \cdots, D. \tag{33}
$$

The exact time-course solution to Equation (10) is

$$
\boldsymbol{\theta}_i(t) = \begin{bmatrix} \boldsymbol{v}_i \\ \boldsymbol{u}_i \end{bmatrix}(t) = \sum_{k=1}^{D}\left(c_{ki}e^{s_k t}\begin{bmatrix} \boldsymbol{q}_k \\ \boldsymbol{r}_k \end{bmatrix} + b_{ki}e^{-s_k t}\begin{bmatrix} \boldsymbol{q}_k \\ -\boldsymbol{r}_k \end{bmatrix}\right) + \boldsymbol{\xi}_i, \quad i = 1, \cdots, H, \tag{34}
$$

where the constants $c_{ki}, b_{ki}$ are projections of the initial weights onto the eigenvectors with nonzero eigenvalues, and $\boldsymbol{\xi}_i$ is the initial weights projected onto the eigenspace with zero eigenvalue

$$c_{ki} = \frac{1}{2}\left(\boldsymbol{q}_k^\top \boldsymbol{v}_i(0) + \boldsymbol{r}_k^\top \boldsymbol{u}_i(0)\right), \tag{35a}$$

$$b_{ki} = \frac{1}{2}\left(\boldsymbol{q}_k^\top \boldsymbol{v}_i(0) - \boldsymbol{r}_k^\top \boldsymbol{u}_i(0)\right), \tag{35b}$$

$$\boldsymbol{\xi}_i = \boldsymbol{\theta}_i(0) - \sum_{k=1}^{D}\left(c_{ki}\begin{bmatrix}\boldsymbol{q}_k\\\boldsymbol{r}_k\end{bmatrix} + b_{ki}\begin{bmatrix}\boldsymbol{q}_k\\-\boldsymbol{r}_k\end{bmatrix}\right). \tag{35c}$$

Recall that the projection matrix $\boldsymbol{P}$, defined in Equation (11), corresponds to the rank-$r$ subspace spanned by the top $r$ singular vectors. The projection of the weights on this subspace has $\ell_2$ norm

$$\|\boldsymbol{P}\boldsymbol{\theta}_i\| = \left\|\sum_{k=1}^{r} c_{ki}e^{s_k t}\begin{bmatrix}\boldsymbol{q}_k\\\boldsymbol{r}_k\end{bmatrix}\right\| = e^{s_1 t}\sqrt{2\sum_{k=1}^{r} c_{ki}^2} = e^{s_1 t}\|\boldsymbol{P}\boldsymbol{\theta}_i(0)\| \tag{36}$$

The time it takes for $\|\boldsymbol{P}\boldsymbol{\theta}_i\|$ to reach $O(1)$ is

$$T = \frac{1}{s_1}\ln\frac{1}{\|\boldsymbol{P}\boldsymbol{\theta}_i(0)\|}. \tag{37}$$

At time $T$, the projection of the weights on the null-space of $\boldsymbol{P}$ has $\ell_2$ norm

$$\begin{aligned}
\|(\boldsymbol{I} - \boldsymbol{P})\boldsymbol{\theta}_i(T)\| &= \left\|\sum_{k=r+1}^{D} c_{ki}e^{s_k T}\begin{bmatrix}\boldsymbol{q}_k\\\boldsymbol{r}_k\end{bmatrix} + \sum_{k=1}^{D} b_{ki}e^{-s_k T}\begin{bmatrix}\boldsymbol{q}_k\\-\boldsymbol{r}_k\end{bmatrix} + \boldsymbol{\xi}_i\right\| \\
&= \sqrt{2\sum_{k=r+1}^{D} c_{ki}^2 e^{2s_k T} + 2\sum_{k=1}^{D} b_{ki}^2 e^{-2s_k T} + \|\boldsymbol{\xi}_i\|^2} \\
&\leq e^{s_{r+1}T}\sqrt{2\sum_{k=r+1}^{D} c_{ki}^2 + 2\sum_{k=1}^{D} b_{ki}^2 + \|\boldsymbol{\xi}_i\|^2} \\
&= \left(\frac{1}{\|\boldsymbol{P}\boldsymbol{\theta}_i(0)\|}\right)^{\frac{s_{r+1}}{s_1}}\|(\boldsymbol{I} - \boldsymbol{P})\boldsymbol{\theta}_i(0)\| \\
&= \left(\frac{\|(\boldsymbol{I} - \boldsymbol{P})\boldsymbol{\theta}_i(0)\|}{\|\boldsymbol{P}\boldsymbol{\theta}_i(0)\|}\right)^{\frac{s_{r+1}}{s_1}}\|(\boldsymbol{I} - \boldsymbol{P})\boldsymbol{\theta}_i(0)\|^{1-\frac{s_{r+1}}{s_1}} \tag{38}
\end{aligned}$$

Because the each entry of the initial weight $\boldsymbol{\theta}_i(0)$ is independently sampled from $\mathcal{N}(0, \epsilon^2)$, the norm is $\|\boldsymbol{\theta}_i(0)\| = O(\epsilon)$ and the ratio $\frac{\|(\boldsymbol{I}-\boldsymbol{P})\boldsymbol{\theta}_i(0)\|}{\|\boldsymbol{P}\boldsymbol{\theta}_i(0)\|} = O(1)$. Thus, we have

$$\|(\boldsymbol{I} - \boldsymbol{P})\boldsymbol{\theta}_i(T)\| = O(1)\|(\boldsymbol{I} - \boldsymbol{P})\boldsymbol{\theta}_i(0)\|^{1-\frac{s_{r+1}}{s_1}} = O\left(\epsilon^{1-\frac{s_{r+1}}{s_1}}\right). \tag{39}$$

Hence, at time $T$, $\|\boldsymbol{P}\boldsymbol{\theta}_i(T)\| = O(1)$, while $\|(\boldsymbol{I} - \boldsymbol{P})\boldsymbol{\theta}_i(T)\| = O(\epsilon^{1-s_{r+1}/s_1})$ is still small. ∎

### G.3 FIXED POINTS OF LINEAR NETWORKS

In Lemma 6, we specify all the fixed points in the linear network learning dynamics in Equation (9).

**Lemma 6.** *Denote the eigenvectors of the symmetric matrix $\boldsymbol{\Sigma}_{yz}\boldsymbol{\Sigma}_{zz}^{-1}\boldsymbol{\Sigma}_{yz}^\top$ as $\boldsymbol{e}_k \in \mathbb{R}^{N_v}$, $k = 1, \cdots, N_v$, arranged in descending order of their associated eigenvalues. There are at most $D = \min(N_v, N_u)$ nonzero eigenvalues. The sufficient and necessary condition for a set of weights to be a fixed point of the dynamics in Equation (9) is*

$$\sum_{i=1}^{H} \boldsymbol{v}_i\boldsymbol{u}_i^\top = \sum_{k \in \mathcal{A}_r} \boldsymbol{e}_k\boldsymbol{e}_k^\top \boldsymbol{\Sigma}_{yz}\boldsymbol{\Sigma}_{zz}^{-1}, \quad \boldsymbol{v}_i \in \text{span}\{\boldsymbol{e}_k\}_{k \in \mathcal{A}_r}, \quad \boldsymbol{u}_i \in \text{span}\left\{\boldsymbol{\Sigma}_{zz}^{-1}\boldsymbol{\Sigma}_{yz}^\top \boldsymbol{e}_k\right\}_{k \in \mathcal{A}_r}, \tag{40}$$

*where $\mathcal{A}_r$ is a set of indices $\mathcal{A}_r \subseteq \{1, 2, \cdots, D\}$, $|\mathcal{A}_r| = r$.*

*Proof.* The proof can be found in the seminal work by Baldi & Hornik (1989) or follow-up work (Kawaguchi, 2016; Lu & Kawaguchi, 2017; Yun et al., 2018; Laurent & von Brecht, 2018; Achour et al., 2024). ∎

The dynamics near a fixed points defined in Equation (40) is approximately a linear dynamical system, for $i = 1, \cdots, H$,

$$\dot{\boldsymbol{v}}_i = (\boldsymbol{\Sigma}_{yz} - \boldsymbol{W}\boldsymbol{\Sigma}_{zz}) \boldsymbol{u}_i \approx \left( \boldsymbol{\Sigma}_{yz} - \sum_{k \in \mathcal{A}_r} \boldsymbol{e}_k \boldsymbol{e}_k^\top \boldsymbol{\Sigma}_{yz} \boldsymbol{\Sigma}_{zz}^{-1} \boldsymbol{\Sigma}_{zz} \right) \boldsymbol{u}_i = \tilde{\boldsymbol{\Sigma}}_{yz} \boldsymbol{u}_i, \tag{41a}$$

$$\dot{\boldsymbol{u}}_i = (\boldsymbol{\Sigma}_{yz} - \boldsymbol{W}\boldsymbol{\Sigma}_{zz})^\top \boldsymbol{v}_i \approx \left( \boldsymbol{\Sigma}_{yz} - \sum_{k \in \mathcal{A}_r} \boldsymbol{e}_k \boldsymbol{e}_k^\top \boldsymbol{\Sigma}_{yz} \boldsymbol{\Sigma}_{zz}^{-1} \boldsymbol{\Sigma}_{zz} \right)^\top \boldsymbol{v}_i = \tilde{\boldsymbol{\Sigma}}_{yz}^\top \boldsymbol{v}_i. \tag{41b}$$

where $\widetilde{\boldsymbol{\Sigma}}_{yz}$ is $\boldsymbol{\Sigma}_{yz}$ projected onto a rank-$(D - r)$ subspace

$$\tilde{\boldsymbol{\Sigma}}_{yz} = \sum_{k \notin \mathcal{A}_r} \boldsymbol{e}_k \boldsymbol{e}_k^\top \boldsymbol{\Sigma}_{yz}. \tag{42}$$

*Remark* 2. When defining $\mathcal{A}_r$ in Lemma 6, there are $\binom{D}{r}$ possible choices of $r$ indices out of $D$, assuming the eigenvalues are distinct. Each choice produces a different input-output linear map. Thus, there are $\binom{D}{r}$ embedded fixed points of effective width $r$ in linear networks, if we count fixed points by the distinct input-output maps they implement. When a linear network undergoes saddle-to-saddle dynamics and approaches fixed points of effective width $r = 0, 1, \cdots, D$ sequentially, determining which one of the $\binom{D}{r}$ fixed points it approaches at each stage is a non-trivial open problem. Even in diagonal linear networks, specifying the sequence of visited saddles requires non-trivial work (Berthier, 2023; 2026; Pesme & Flammarion, 2023).

# H DYNAMICS OF QUADRATIC NETWORKS

## H.1 GRADIENT FLOW EQUATIONS

The gradient flow dynamics of Equation (13) trained on squared loss, $\ell(y, \hat{y}) = \frac{1}{2}(y - \hat{y})^2$, is given by

$$
\dot{v}_i = -\frac{1}{P} \sum_{\mu=1}^{P} \frac{\partial \mathcal{L}}{\partial f(\boldsymbol{x}_\mu)} \frac{\partial f(\boldsymbol{x}_\mu)}{\partial v_i}
$$

$$
= \frac{1}{P} \sum_{\mu=1}^{P} \left( y_\mu - \sum_{j=1}^{H} v_j \boldsymbol{u}_j^\top \boldsymbol{Z}_\mu \boldsymbol{u}_j \right) \boldsymbol{u}_i^\top \boldsymbol{Z}_\mu \boldsymbol{u}_i
$$

$$
= \boldsymbol{u}_i^\top \left( \frac{1}{P} \sum_{\mu=1}^{P} y_\mu \boldsymbol{Z}_\mu \right) \boldsymbol{u}_i - \sum_{j=1}^{H} v_j \boldsymbol{u}_j^\top \left( \frac{1}{P} \sum_{\mu=1}^{P} \boldsymbol{Z}_\mu \boldsymbol{u}_j \boldsymbol{u}_i^\top \boldsymbol{Z}_\mu \right) \boldsymbol{u}_i,
$$

$$
\dot{\boldsymbol{u}}_i = -\frac{1}{P} \sum_{\mu=1}^{P} \frac{\partial \mathcal{L}}{\partial f(\boldsymbol{x}_\mu)} \frac{\partial f(\boldsymbol{x}_\mu)}{\partial \boldsymbol{u}_i}
$$

$$
= \frac{1}{P} \sum_{\mu=1}^{P} \left( y_\mu - \sum_{j=1}^{H} v_j \boldsymbol{u}_j^\top \boldsymbol{Z}_\mu \boldsymbol{u}_j \right) 2 \boldsymbol{Z}_\mu \boldsymbol{u}_i
$$

$$
= 2\boldsymbol{u}_i^\top \left( \frac{1}{P} \sum_{\mu=1}^{P} y_\mu \boldsymbol{Z}_\mu \right) \boldsymbol{u}_i - 2 \sum_{j=1}^{H} v_j \left( \frac{1}{P} \sum_{\mu=1}^{P} \boldsymbol{Z}_\mu \boldsymbol{u}_i \boldsymbol{u}_j^\top \boldsymbol{Z}_\mu \right) \boldsymbol{u}_j.
$$

Denote the data statistics as

$$
\boldsymbol{\Sigma}_{yZ} = \frac{1}{P} \sum_{\mu=1}^{P} y_\mu \boldsymbol{Z}_\mu, \quad \boldsymbol{\Sigma}_{ZZ} = \frac{1}{P} \sum_{\mu=1}^{P} \text{vec}(\boldsymbol{Z}_\mu) \text{vec}(\boldsymbol{Z}_\mu)^\top. \tag{43}
$$

The dynamics can be written as

$$
\dot{v}_i = \boldsymbol{u}_i^\top \boldsymbol{\Sigma}_{yZ} \boldsymbol{u}_i - \sum_{j=1}^{H} v_j (\boldsymbol{u}_j \otimes \boldsymbol{u}_j)^\top \boldsymbol{\Sigma}_{ZZ} (\boldsymbol{u}_i \otimes \boldsymbol{u}_i), \tag{44a}
$$

$$
\dot{\boldsymbol{u}}_i = 2v_i \boldsymbol{\Sigma}_{yZ} \boldsymbol{u}_i - 2v_i \sum_{j=1}^{H} v_j (\boldsymbol{u}_j \otimes \boldsymbol{u}_j)^\top \boldsymbol{\Sigma}_{ZZ} (\boldsymbol{u}_i \otimes \boldsymbol{I}_D), \tag{44b}
$$

where $\otimes$ denotes the Kronecker product.

## H.2 DERIVATIONS FOR TIMESCALE SEPARATION

We here provide the derivations for Proposition 5, that is the timescale separation between units in quadratic dynamics.

To reduce clutter, we omit the index $i$ in Equation (14) for now; we will put it back when we need it. We study the dynamics

$$
\dot{v} = \boldsymbol{u}^\top \boldsymbol{\Sigma}_{yZ} \boldsymbol{u}, \quad \dot{\boldsymbol{u}} = 2v \boldsymbol{\Sigma}_{yZ} \boldsymbol{u}. \tag{45}
$$

Step 1: reduction to one-dimensional dynamics.

Denote the eigenvalues and eigenvectors of the symmetric matrix $\boldsymbol{\Sigma}_{yZ}$ as

$$
\boldsymbol{\Sigma}_{yZ} \boldsymbol{r}_k = s_k \boldsymbol{r}_k, \quad k = 1, \cdots, D. \tag{46}
$$

We change variables by projecting $\boldsymbol{u}$ onto the (orthonormal) eigenvectors of $\boldsymbol{\Sigma}_{yZ}$,

$$
a_k \equiv \frac{1}{\sqrt{2}} \boldsymbol{r}_k^\top \boldsymbol{u}, \quad k = 1, \cdots, D \tag{47}
$$

where the factor of $\sqrt{2}$ is for convenience only. Since time has arbitrary unit, we can let $t \to t/2$. Then, the dynamics of $\boldsymbol{u}$ and $v$ can be expressed in terms of the new coordinates $a_1, \cdots, a_D$ and $v$,

$$\dot{v} = \sum_{k=1}^{D} s_k a_k^2, \tag{48a}$$

$$\dot{a}_k = v s_k a_k, \quad k = 1, \cdots, D. \tag{48b}$$

This set of equations admits a conservation law,

$$\frac{\mathrm{d}}{\mathrm{d}t} \left( v^2 - \sum_{k=1}^{D} a_k^2 \right) = 2v \sum_{k=1}^{D} s_k a_k^2 - 2v \sum_{k=1}^{D} s_k a_k^2 = 0. \tag{49}$$

Thus, their difference at initialization is conserved throughout training

$$v(t)^2 - \sum_{k=1}^{D} a_k(t)^2 = v(0)^2 - \sum_{k=1}^{D} a_k(0)^2. \tag{50}$$

We also notice a relationship between the $a_k$,

$$\frac{\mathrm{d}a_m}{\mathrm{d}a_k} = \frac{s_m a_m}{s_k a_k} \quad \Rightarrow \quad \frac{1}{s_m} \ln \frac{a_m(t)}{a_m(0)} = \frac{1}{s_k} \ln \frac{a_k(t)}{a_k(0)}. \tag{51}$$

Thus, different $a_k(t)$ can be expressed in terms of each other,

$$a_k(t) = a_k(0) \left( \frac{a_m(t)}{a_m(0)} \right)^{s_k/s_m}. \tag{52}$$

Using Equations (50) and (52), and defining

$$\pi_k(t) \equiv \frac{a_k(t)}{a_k(0)}, \tag{53}$$

we can express $v(t)$ in terms of $\pi_m(t)$,

$$v(t)^2 = v(0)^2 + \sum_{k=1}^{D} a_k(0)^2 \left( \pi_m(t)^{2s_k/s_m} - 1 \right). \tag{54}$$

At this point $m$ is arbitrary; we will set it to a particular value shortly. Substituting Equation (54) into Equation (48b) and using Equation (53), we obtain a one-dimensional and separable differential equation for $\pi_m$

$$\dot{\pi}_m = v s_m \pi_m = \mathrm{sign}\left( v(0) \right) s_m \pi_m \sqrt{v(0)^2 + \sum_{k=1}^{D} a_k(0)^2 \left( \pi_m^{2s_k/s_m} - 1 \right)} \tag{55}$$

with initial condition $\pi_m(0) = 1$. Reducing the $(D+1)$-dimensional dynamics in Equation (45) to the one-dimensional separable differential equation in Equation (55) may be of independent interest.

Step 2: bounding the growth time.

Let us choose $m$ to maximize $\mathrm{sign}(v(0)) s_m$. If $v(0)$ is positive, the chosen $m$ maximizes $s_m$; if it is negative the chosen $m$ minimizes $s_m$ (and thus maximize $|s_m|$, assuming that there are both negative and positive eigenvalues). In either case, $\max_k(s_k/s_m) = 1$. And with this maximization, the prefactor becomes $|s_m|$.

We are interested in the time it takes for $a_m(t)$ to become large; that happens when $t \sim 1/a_m(0)$, which is large for small initial conditions. The time it takes for that to happen, denoted $t_{\mathrm{final}}$, is bounded by the time it takes for $\pi_m(t)$ to go to infinity, giving us

$$t_{\mathrm{final}} < t_\infty = \frac{1}{|s_m|} \int_1^\infty \frac{\mathrm{d}\pi}{\pi \sqrt{v(0)^2 + \sum_{k=1}^{D} a_k(0)^2 (\pi^{2s_k/s_m} - 1)}}. \tag{56}$$

So far we have ignored the dependence on unit, $i$. However, each $i$ has associated with it a different $t_\infty$. Let us use $t_{\infty,i}$ to denote the different times, arranged in increasing order,

$$t_{\infty,1} < t_{\infty,2} < ... < t_{\infty,H} . \tag{57}$$

We will also add a subscript $i$ to all the other variables as well. At time $t_{\infty,1}$, we know that $\pi_{m,1}(t_{\infty,1})$ is large, but what about $\pi_{m,i}(t_{\infty,1})$ for $i > 1$? That is given implicitly by

$$t_{\text{final}} = \frac{1}{|s_{m,i}|} \int_1^{\pi_{m,i}(t_{\text{final}})} \frac{\mathrm{d}\pi}{\pi \sqrt{v_i(0)^2 + \sum_{k=1}^D a_{k,i}(0)^2 (\pi^{2s_k/s_{m,i}} - 1)}}. \tag{58}$$

Note that $s_m$ acquired a subscript $i$, because it is either the maximum or minimum eigenvalue, depending on the sign of $v_i(0)$. Rearranging terms slightly and performing a small amount of algebra, this can be written as

$$\int_{\pi_{m,i}(t_{\text{final}})}^\infty \frac{\mathrm{d}\pi}{\pi^2 \sqrt{1 + \Psi_i(\pi)}} = |s_{m,i} a_{m,i}(0)|(t_{\infty,i} - t_{\text{final}}) \tag{59}$$

where

$$\Psi_i(\pi) \equiv \frac{1}{\pi^2} \left( \frac{v_i(0)^2}{a_{m,i}(0)^2} + \sum_{k \neq m} \frac{a_{k,i}(0)^2}{a_{m,i}(0)^2} (\pi^{2s_k/s_{m,i}} - 1) - 1 \right). \tag{60}$$

We can bound the left hand side,

$$\int_{\pi_{m,i}(t_{\text{final}})}^\infty \frac{\mathrm{d}\pi}{\pi^2 \sqrt{1 + \Psi_i(\pi)}} < \frac{1}{\sqrt{1 + \Psi_{\min,i}}} \int_{\pi_{m,i}(t_{\text{final}})}^\infty \frac{\mathrm{d}\pi}{\pi^2} = \frac{1}{\sqrt{1 + \Psi_{\min,i}}} \frac{1}{\pi_{m,i}(t_{\text{final}})} \tag{61}$$

where $\Psi_{\min,i}$ is the minimum value of $\Psi_i(\pi)$. Inserting this into Equation (59) then gives us

$$\pi_{m,i}(t_{\text{final}}) < \frac{1}{\sqrt{1 + \Psi_{\min,i}}} \frac{1}{|s_{m,i} a_{m,i}(0)|(t_{\infty,i} - t_{\text{final}})}. \tag{62}$$

To determine the size of the right hand side, we first note that because the spread in initial conditions is $O(1)$, the relative spread in the $t_{\infty,i}$ is $O(1)$. Second, the $t_{\infty,i}$ scale inversely with initial conditions (see Equation (56)), whose typical size we denote $\epsilon$. Consequently,

$$|s_{m,i} a_{m,i}(0)|(t_{\infty,i} - t_{\text{final}}) \sim \epsilon \cdot \frac{1}{\epsilon} \sim O(1), \tag{63}$$

from which it follows that $\pi_{m,i}(t_{\text{final}}) \sim O(1)$. And so, given the definition of $\pi_m(t)$ in Equation (53), we have $a_{m,i}(t_{\text{final}}) \sim \epsilon$. Thus, when the variables associated with one of the units become $O(1)$, the variables associated with all the other units are $O(\epsilon)$.

## I IMPLEMENTATION DETAILS

Videos of the learning dynamics in Figure 1 are provided in the supplementary material.

For all models, we sample the initial weights from $\mathcal{N}(0, \epsilon^2)$ and train them with squared loss

$$\mathcal{L} = \frac{1}{P} \sum_{\mu=1}^{P} \ell(\boldsymbol{y}_\mu, f(\boldsymbol{x}_\mu)) = \frac{1}{P} \sum_{\mu=1}^{P} \|\boldsymbol{y}_\mu - f(\boldsymbol{x}_\mu)\|_2^2. \tag{64}$$

**Linear fully-connected network** (Figure 1B).

The network is defined as

$$f(\boldsymbol{x}) = \sum_{i=1}^{H} \boldsymbol{v}_i \boldsymbol{u}_i^\top \boldsymbol{x}, \quad \text{where } \boldsymbol{v}_i, \boldsymbol{u}_i, \boldsymbol{x} \in \mathbb{R}^2, H = 50. \tag{65}$$

The training set $\{\boldsymbol{x}_\mu, \boldsymbol{y}_\mu\}_{\mu=1}^{P}$ is generated as

$$\boldsymbol{y}_\mu = \boldsymbol{W}^* \boldsymbol{x}_\mu, \quad \boldsymbol{x}_\mu \sim \mathcal{N}\left(\begin{bmatrix} 0 \\ 0 \end{bmatrix}, \begin{bmatrix} 1 & 1 \\ 1 & 4 \end{bmatrix}\right).$$

Here $P = 8192, \epsilon = 10^{-6}$, and the learning rate is 0.01.

**Linear convolutional network** (Figure 1C).

The network is defined as

$$f(\boldsymbol{x}) = \sum_{i=1}^{H} [v_{i1} \quad v_{i2}] \begin{bmatrix} u_{i1} & u_{i2} & 0 & 0 \\ 0 & 0 & u_{i1} & u_{i2} \end{bmatrix} \boldsymbol{x}, \quad \text{where } \boldsymbol{x} \in \mathbb{R}^4, H = 50. \tag{66}$$

Here the first layer is a one-dimensional convolutional layer with kernel size 2, stride 2, and padding 0. We set $H = 50$. The pytorch code for setting up this layer is

```
torch.nn.Conv1d(in_channels=1,
                out_channels=50,
                kernel_size=2,
                stride=2,
                padding=0,
                dilation=1,
                groups=1,
                bias=False)
```

The training set $\{\boldsymbol{x}_\mu, y_\mu\}_{\mu=1}^{P}$ is generated as

$$y_\mu = \boldsymbol{w}^{*\top} \boldsymbol{x}_\mu, \quad \boldsymbol{x}_\mu \sim \mathcal{N}\left(\begin{bmatrix} 0 \\ 0 \\ 0 \\ 0 \end{bmatrix}, \begin{bmatrix} 1 & 0 & 0 & 0 \\ 0 & 1 & 0 & 0 \\ 0 & 0 & 2 & 0 \\ 0 & 0 & 0 & 1 \end{bmatrix}\right), \quad \boldsymbol{w}^* = \frac{1}{\sqrt{5}} \begin{bmatrix} 1 \\ 1 \\ -1 \\ 1 \end{bmatrix}.$$

Here $P = 8192, \epsilon = 10^{-6}$, and the learning rate is 0.01.

**ReLU fully-connected network** (Figure 1D).

The network is defined as

$$f(\boldsymbol{x}) = \sum_{i=1}^{H} v_i \mathsf{ReLU}(\boldsymbol{u}_i^\top \boldsymbol{x}), \quad \text{where } v_i \in \mathbb{R}, \boldsymbol{u}_i, \boldsymbol{x} \in \mathbb{R}^2, H = 50. \tag{67}$$

The training set is an orthogonal input dataset used in Boursier et al. (2022, Figure 3) and Zhang et al. (2025a, Figure 4). It contains two data points

$$\boldsymbol{x}_1 = \begin{bmatrix} 1 \\ 0.5 \end{bmatrix}, \quad y_1 = 1,$$

$$\boldsymbol{x}_2 = \begin{bmatrix} -1 \\ 2 \end{bmatrix}, \quad y_2 = -1.$$

Here $P = 2, \epsilon = 10^{-6}$, and the learning rate is 0.01.

**ReLU convolutional network** (Figure 1E).

The network is defined as

$$f(\boldsymbol{x}) = \sum_{i=1}^{H} \begin{bmatrix} v_{i1} & v_{i2} \end{bmatrix} \mathsf{ReLU} \left( \begin{bmatrix} u_{i1} & u_{i2} & 0 & 0 \\ 0 & 0 & u_{i1} & u_{i2} \end{bmatrix} \boldsymbol{x} \right), \quad \text{where } \boldsymbol{x} \in \mathbb{R}^4, H = 50, \quad (68)$$

which is the same as Equation (66) except for the ReLU activation function. The training set $\{\boldsymbol{x}_\mu, y_\mu\}_{\mu=1}^{P}$ is generated as

$$y_\mu = \boldsymbol{w}^{*\top} \boldsymbol{x}_\mu, \quad \boldsymbol{w}^* = \frac{1}{\sqrt{5}} \begin{bmatrix} 1 \\ 1 \\ -1 \\ 1 \end{bmatrix}.$$

It contains four data points

$$\boldsymbol{x}_1 = \begin{bmatrix} 2 \\ 0 \\ 0 \\ 0 \end{bmatrix}, \ \boldsymbol{x}_2 = \begin{bmatrix} 0 \\ 2 \\ 0 \\ 0 \end{bmatrix}, \ \boldsymbol{x}_3 = \begin{bmatrix} 0 \\ 0 \\ 2\sqrt{2} \\ 0 \end{bmatrix}, \ \boldsymbol{x}_4 = \begin{bmatrix} 0 \\ 0 \\ 0 \\ 2 \end{bmatrix}.$$

Here $P = 4, \epsilon = 10^{-6}$, and the learning rate is 0.01.

**Linear self-attention** (Figure 1F).

The model is defined as

$$f(\boldsymbol{X}) = \boldsymbol{X} + \sum_{i=1}^{H} \boldsymbol{V}_i \boldsymbol{X} \boldsymbol{X}^\top \boldsymbol{K}_i^\top \boldsymbol{Q}_i \boldsymbol{X}, \quad (69)$$

where

$$\boldsymbol{V}_i \in \mathbb{R}^{(D+1)\times(D+1)}, \boldsymbol{K}_i, \boldsymbol{Q}_i \in \mathbb{R}^{R\times(D+1)}, \boldsymbol{X} \in \mathbb{R}^{(D+1)\times(N+1)}. \quad (70)$$

Here $D$ is the embedding dimension, $N$ is the context length, and $R$ is the rank of each attention head. We train the linear self-attention model on an in-context linear regression task (Garg et al., 2022; Zhang et al., 2025b). The training set $\{\boldsymbol{X}_\mu, y_{\mu,q}\}_{\mu=1}^{P}$ is generated as

$$\boldsymbol{X}_\mu = \begin{bmatrix} \boldsymbol{x}_{\mu,1} & \boldsymbol{x}_{\mu,2} & \cdots & \boldsymbol{x}_{\mu,N} & \boldsymbol{x}_{\mu,q} \\ \boldsymbol{w}_\mu^\top \boldsymbol{x}_{\mu,1} & \boldsymbol{w}_\mu^\top \boldsymbol{x}_{\mu,2} & \cdots & \boldsymbol{w}_\mu^\top \boldsymbol{x}_{\mu,N} & 0 \end{bmatrix}, \quad y_{\mu,q} = \boldsymbol{w}_\mu^\top \boldsymbol{x}_{\mu,q},$$

and

$$\boldsymbol{x}_{\mu,n}, \boldsymbol{x}_{\mu,q} \sim \mathcal{N}(\boldsymbol{0}, \boldsymbol{I}), \quad \boldsymbol{w}_\mu \sim \mathcal{N}(\boldsymbol{0}, \boldsymbol{I}), \quad n = 1, \cdots, N, \ \mu = 1, \cdots, P.$$

Here $D = 2, N = 32, R = 1, H = 10, P = 8192, \epsilon = 0.005$, and the learning rate is 0.02.

**Quadratic network** (Figure 1G).

The network is defined as

$$f(\boldsymbol{x}) = \sum_{i=1}^{H} v_i \left( \boldsymbol{u}_i^\top \boldsymbol{x} \right)^2, \quad \text{where } v_i \in \mathbb{R}, \boldsymbol{u}_i, \boldsymbol{x} \in \mathbb{R}^2. \quad (71)$$

The training set $\{\boldsymbol{x}_\mu, y_\mu\}_{\mu=1}^{P}$ is generated as

$$y_\mu = \left( \boldsymbol{w}_1^{*\top} \boldsymbol{x}_\mu \right)^2 + \left( \boldsymbol{w}_2^{*\top} \boldsymbol{x}_\mu \right)^2, \quad \boldsymbol{x}_\mu \sim \mathcal{N}(\boldsymbol{0}, \boldsymbol{I}), \quad \boldsymbol{w}_1^* = \begin{bmatrix} 1 \\ 0 \end{bmatrix}, \quad \boldsymbol{w}_1^* = \begin{bmatrix} 0 \\ 1 \end{bmatrix}.$$

Here $H = 10, P = 8192, \epsilon = 0.005$, and the learning rate is 0.04.

In linear self-attention and the quadratic network, we set $H = 10$, not 50 as the other architectures, because a large $H$ makes the plateaus very short for these two architectures. This effect was discussed in Section 6 and validated with simulations in Figure 2A.

