# OpenReview forum: "Saddle-to-Saddle Dynamics Explains A Simplicity Bias Across Neural Network Architectures"
_ICLR.cc/2026/Conference — ICLR 2026 Poster_

### Official Review · Reviewer_8EAS · 2025-10-21

**Soundness:** 3
**Presentation:** 3
**Contribution:** 3
**Rating:** 8
**Confidence:** 3

**Summary:**

This paper provides a dynamical explanation for the simplicity bias observed in neural networks trained with SGD. The authors argue that when a neural network is initialized with small weights (i.e., near the origin of the loss landscape), it evolves closely along an invariant manifold corresponding to a width-$h$ network. It can then transition to an invariant manifold corresponding to a network of width $h+1$, which results in a steep decrease in loss.

- Their central contribution is that, for this initialization, the saddle-to-saddle dynamics corresponds to a one-by-one activation of neurons. The network starts by fitting a single neuron, evolving closely along the corresponding invariant manifold until the optimal one-neuron solution is found. However, this minimum from a one-neuron perspective is a saddle point for the full-width network. The network then transitions to activate another dormant neuron and evolves closely along a two-neuron invariant manifold. This process repeats.

- While the embedding of width-$(h-1)$ networks within width-$h$ networks is well-known, the authors demonstrate this for CNNs and attention layers and connect the phenomenon to the dynamics of SGD. They also provide novel propositions regarding timescale separation.

**Strengths:**

- This work connects saddle-to-saddle dynamics to the stage-wise learning of neurons along invariant manifolds, proving a specific version of simplicity bias. While the embedding of width-$h$ neural networks in the loss landscape of width-$(h+1)$ NNs is well-known, the authors extend these results to CNNs and attention layers.

- They prove a timescale separation for the saddle-to-saddle dynamics, which is driven by the target function for linear neural networks but by the weight magnitude in the quadratic case.

- One of the most interesting points is the new understanding of the lazy vs. rich regime. The dynamics within these regimes (i.e., saddle-to-saddle dynamics along invariant manifolds) can be distinct from strong feature learning, such as learning a low-rank solution. This has the potential to provide a much more detailed account of SGD dynamics.

**Weaknesses:**

- A more formal introduction to dynamical systems, perhaps in a short paragraph or an appendix, would be helpful. For instance, the term "invariant manifold" is used but never formally defined.

- There is a significant body of literature on spectral bias in kernels (e.g. https://arxiv.org/pdf/1912.01198), such as the Neural Tangent Kernel (NTK). The paper is missing a discussion on how its results connect to spectral bias in function space. Although the mechanism in the infinite-width limit is different, it would be valuable to explain these differences and explore how the two perspectives could be combined.

- A clearer study, even if only empirical, on how the structure of the target function influences learning and saddle-to-saddle dynamics would be beneficial. For example, what happens with single-index models or k-sparse parity? How does the data distribution $p(x)$ (e.g., normal, uniform, or spiked) affect the results? It's unclear to what extent the observed dynamics are solely a result of how neural networks are initialized and trained. Since the gradient is taken with respect to the loss function, it would be highly interesting to understand how data-induced invariant manifolds (see questions) might arise and interact with the model-induced ones.

- A clearer and more detailed phase diagram of the lazy and rich regimes would be nice, incorporating the new insight that the distinction is more about saddles and invariant manifolds than just the initialization scale and learing rate.

-  There is a typo on Line 424.

**Questions:**

- What happens to the dynamics if there are loss-induced invariant manifolds? For instance, in a teacher-student setup where the target function (the teacher) has the same symmetries as the student network, how would this affect the dynamics? In general, how does the data set and e.g. the data set size influence the dynamics and generalization.

- There is a long history of research on simplicity bias (especially: https://www.jmlr.org/papers/volume22/20-676/20-676.pdf, https://www.nature.com/articles/s41467-024-54813-x). How does the notion of simplicity bias presented here connect to these prior findings?

- Based on these results, what are potential new definitions for the lazy vs. rich regimes?

- How can this framework be connected to the kernel perspective on spectral bias?

- How do the results change when we remove the parameter symmetries? (https://neurips.cc/virtual/2024/poster/93573)

---

> ### Author Response · Authors · 2025-11-23
>
> Thank you very much for your thoughtful review. We really appreciate your constructive feedback and your positive assessment of our work. We've incorporated your suggestions into the revised PDF, with all changes marked in blue. We'd like to respond to your questions below.
>
> - **Formal definition of invariant manifolds**
>
>   Thank you for this suggestion. We added Appendix F.1 to provide formal definitions of invariant manifolds.
>
> - **Connecting to literature on spectral bias in kernels**
>
>   We've expanded Appendix A.2 to discuss the literature on spectral bias in more detail. In the kernel regime, the network learns each eigenfunction of the neural tangent kernel at a rate of its associated eigenvalue, so eigenfunctions with larger eigenvalues are learned faster. As you correctly noted, this behavior differs from saddle-to-saddle dynamics, as networks in the kernel regime do not visit saddles or exhibit plateaus during learning. The training loss decays throughout learning, with faster decay early in learning and slower decay later. Thus, spectral bias (kernel regime) and saddle-to-saddle dynamics (rich regime) arise in different regimes of learning and lead to different forms of simplicity bias. These clarifications are now incorporated into Appendix A.2.
>
> - **Connecting to literature on Kolmogorov simplicity bias**
>
>   Thank you for this note. In this work, we study the "dynamical" simplicity bias, while indeed, a broader and longstanding body of research has studied the "stationary" simplicity bias. We added a related work subsection in Appendix A.3, where we discuss the broader simplicity bias literature.
>
> - **New definitions for lazy/rich regimes**
>
>   We appreciate that you found new perspectives on rich/lazy learning suggested by our theory interesting. As detailed in Section 6, our results point to a new criterion: the distance from the initialization to invariant manifolds associated with low effective width may determine the strength of feature learning. Here, "low effective width" is width that is lower than or equal to the minimal number of units needed to express the target function. This new understanding enabled us to make linear networks learn the rich solution quickly in Figure 2C. While we agree that a general phase diagram would be a valuable visualization, we currently rely on the theoretical description because the geometry of invariant manifolds is generally complex and architecture-dependent. We're exploring ways to define an appropriate 2D phase plane and will include a visualization in the final revision if feasible.
>
> - **Effect of task structure, model-induced invariant manifolds vs data-induced invariant manifolds**
>
>   Your question on the effect of task structure is closely related to the effect of data statistics we study in Section 6, as the task structure affects the dynamics only through summary data statistics in the linear and quadratic cases.
>
>   Your idea of distinguishing model-induced and data-induced invariant manifolds points to a very exciting direction for future research. We added a paragraph in our discussion Section 7 to include this point. The fixed points and invariant manifolds in our paper arise solely from the network architecture and thus hold for any training data set and loss function. A future direction is to explore whether particular data sets can induce more fixed points and invariant manifolds than the data-agnostic ones.
>
> - **Effect of removing parameter symmetries**
>
>   Thank you for the interesting question. Indeed, methods like those in [Lim et al., 2024](https://arxiv.org/abs/2405.20231) are designed specifically to break permutation symmetry among hidden units. Since our results rely on this symmetry, we do not expect our fixed points and invariant manifolds to persist in networks where the symmetry is deliberately removed. We note that whether our results persist or break in that setting is neither inherently positive nor negative, as the symmetry breaking is used by Lim et al. (2024) as a tool to experimentally probe the role of symmetry, rather than representing the standard training dynamics.
>
> - We've fixed the typo in line 424. Thanks!
>
> We hope our revision is helpful and welcome any further feedback.

---

> > ### Comment · Reviewer_8EAS · 2025-11-25
> > **Answer to rebuttal**
> >
> > I thank the authors for all the helpful additions. I will keep my score.

---

### Official Review · Reviewer_axwH · 2025-10-25

**Soundness:** 3
**Presentation:** 3
**Contribution:** 3
**Rating:** 6
**Confidence:** 4

**Summary:**

The submitted work analyzes saddle to saddle dynamics in neural networks. Specifically, the authors introduce a unifying framework for analyzing various architectures, including FCNs, CNNs and Attention.

The key results are:
1. Fixed points of narrow networks are embedded inside wider networks.
2. Mechanism behind saddle to saddle dynamics
3. Definition of 'simple' in different networks: for instance, low rank in linear networks

These results have interesting implications:
1. While network width does not affect the time scale separation in linear networks, adding more attention heads in shortens the plateaus
2. The power law exponent of the data results in longer plateaus
3. Shorter plateaus result from increases initialization scale

**Strengths:**

The paper has several interesting contributions:
1. Different architectures learn differently: linear networks are more data driven where quadratic models (such as attention) are sensitive to initialization scale.
2. Conditions for Saddle-to-saddle dynamics

Overall, the paper is well written has significant contributions towards our understanding of neural network dynamics and its dependence on architecture and dataset properties.

**Weaknesses:**

* The work is limited to simple setups: feedforward networks, gradient flow and synthetic datasets

**Questions:**

* Do the authors have an intuition about how saddle to saddle dynamics changes on adding residual connections? At least empirically?
* Can the authors clarify how their results on time scale separation differs from works like [1]

[1] Exact solutions to the nonlinear dynamics of learning in deep linear neural networks, 2014

---

> ### Author Response · Authors · 2025-11-23
>
> Thank you very much for your thoughtful review. We're glad to hear that you find our contributions towards understanding neural network dynamics interesting and significant. We've revised our manuscript based on your feedback, with all changes marked in blue. We'd like to respond to your questions below.
>
> - **Dynamics in networks with residual connections**
>
>   Thank you for this interesting question. We added Figure 8 in Appendix B.3 to show the learning dynamics in deep fully-connected linear and ReLU networks with residual connections. We trained three networks: one with no skip connection, one with a skip connection that skips one layer, and one with a skip connection that skips two layers. All three networks exhibit saddle-to-saddle dynamics when trained from small initialization, with the network that skips more layers learning faster. This is because weights in the skipped layers can remain near zero while weights in other layers escape the zero fixed point and learn. When the skipped layers are effectively unused, the network behaves like a shallower network consisting only of the unskipped layers and exhibits saddle-to-saddle dynamics. Furthermore, since shallower linear networks learn faster, networks that skip more layers also learn faster.
>
> - **Comparison with linear network dynamics literature**
>
>   [Saxe et al. (2014)](https://arxiv.org/abs/1312.6120) indeed studied the saddle-to-saddle learning dynamics in deep linear networks. Our setup differs in two key ways.
>
>   First, Saxe et al. (2014) considered a white input covariance matrix, i.e., identity matrix. Under this assumption, the learning dynamics decouples into several independent one-dimensional ODEs, and the timescale separation follows from the different timescales in these independent ODEs. In comparison, we consider the general non-white input case, where the dynamics cannot be reduced to independent ODEs and therefore requires analysis beyond Saxe et al. (2014).
>
>   Second, Saxe et al. (2014) used an ansatz in which the weights are aligned at initialization, whereas we study why the weights naturally become aligned from small isotropic initialization (Theorem 4). Understanding the mechanism of alignment is important, because it informs us about how the data statistics and initialization affect the timescales. This is essential for dissociating the data-induced and initialization-induced plateaus, which were not distinguished in prior literature.
>
>   We've expanded Appendix A.1 to discuss this line of related literature in more detail.
>
> - **Experiment beyond synthetic datasets and gradient flow**
>
>   We added Figure 3 showing the learning dynamics of two-layer linear / ReLU networks trained for binary classification of MNIST digits, in which we use SGD with a batch size of 64. Although the dynamics is noisier than that on synthetic datasets, there is a pronounced intermediate plateau in training. The intermediate plateau is longer when the two digits are harder to distinguish. For example, the plateau in learning to classify digits 3/5 is longer than that of digits 0/1. We also plotted the top singular values of the first-layer weight matrix. Consistent with our theory, the growth of the first and second singular values coincides with the first and second abrupt drops in the training loss, respectively. The remaining singular values are small, implying that the network has effective width 1 and 2 during the intermediate plateau and at convergence.
>
> We hope our revision is helpful and welcome any further feedback.

---

### Official Review · Reviewer_S6X7 · 2025-10-26

**Soundness:** 4
**Presentation:** 2
**Contribution:** 3
**Rating:** 4
**Confidence:** 4

**Summary:**

This work attempts to explain the emergence of saddle-to-saddle dynamics in
the training of neural networks by gradient descent via invariant manifolds. The
invariant manifolds in question correspond to the number of effective units of the
network. In order for saddle-to-saddle dynamics to emerge, the authors argue
that initialization must be close to an invariant manifold on which zero loss is
unattainable and that the escape path from saddles closely follows invariant
manifolds.

Concretely, a result on the embedding of local minima in smaller networks
as saddles within larger networks is proven, extending [7]. Theorem 3 provides
a group of properties invariant under gradient flow. The theoretical results
on saddle-to-saddle dynamics focus on two-layer linear and quadratic (in the
weights) networks. Timescale separation is argued to occur in linear networks
due to the distribution of data and in quadratic networks (analysis restricted
to scalar targets) due to the initialization. Based on this characterization of
the invariant manifolds visited during the training, the authors provide some
conjectures on learning dynamics.

**Strengths:**

The structure of the paper is good and related literature is thoroughly documented. Equations (6) and (7) in Theorem 1 provide new examples of embedded fixed points which may be of independent interest. Theorem 3 shows
that the properties of units being zero, have equal weights, have proportional
weights in the case of $\phi$ homogeneous in $u$, or have linearly dependent weights
in the case of $\phi$ linear in $u$ are preserved under gradient flow. I found these
results to be interesting. In Appendix F.1 it is shown that on such invariant
manifolds the network may be represented with one with fewer units. Therefore saddle-to-saddle dynamics between such manifolds could indicate learning
sequentially more complex networks. The proofs of these results are clear and I
can find no issue with them. Theorem 4 and Proposition 5 are presented clearly
and their proofs (if one accepts the setup) seem to be correct and concise, with
observations on different drivers of saddle-to-saddle dynamics in the two considered settings. Proposition 5 seems to be both novel and interesting. The
motivation for scaling up width in transformers is a good application and experiments align with the theory. More generally, claims are supported with well
presented and relevant experiments. The authors provide conjectures on when
saddle-to-saddle dynamics fail to emerge and broader settings exhibiting permutation symmetry where saddle-to-saddle dynamics may emerge. That some
future research directions are motivated by this work is positive.

**Weaknesses:**

I am concerned that Theorem 4 is not novel. As the authors note in Appendix
A.1, the incremental learning dynamics of linear networks are quite well understood. [12] and [8] seem to cover the same ground as Theorem 4, and closely related work includes [1, 9, 10, 6, 13]. Could the authors be more explicit in
explaining the novelty of this theorem?

It is noted that one of the conditions for saddle-to-saddle dynamics is that
escape paths closely follow invariant manifolds. The authors do not discuss at
length when this may be satisfied or violated. As this is posited as key for
the emergence of such dynamics the paper would be improved with some more
examples and heuristic commentary, even if a general theoretical treatment of
this condition is infeasible.

The procedure of linearizing about fixed points and studying this dynamical
system is not valid once some of the weights have begun to move away from the
critical point. Questions of rigour notwithstanding, the theoretical setting is
quite restrictive in that it only considers networks of depth 2 and with no non-linear activation. This setting is arguably more restrictive than, for example, [2], although I appreciate that a diagonal parametrisation is used here (as an
aside the authors might like to consider including this reference?). The authors
suggest why tanh networks fail to exhibit saddle-to-saddle dynamics, but a
wider discussion of how the results apply to commonly used activations (e.g.
leaky ReLU) would strengthen the paper. Similarly, a discussion of the role
of depth would make the paper more relevant. It seems to me that Appendix
B illustrates that saddle-to-saddle dynamics can still occur in deeper networks
with non-linear activations, but without much comment on differences with the
linear two-layer case (or justification if there is no difference).

**Questions:**

The claim that the Theorems 1 and 3 together imply saddle-to-saddle dynamics
of increasing network complexity seems quite strong. On this point I have the
following four questions:

1) Can the authors justify that there do not exist many other invariant
manifolds under gradient flow? If others do exist, why could the dynamics not
take us between these manifolds?

2) Can the authors more clearly justify the statement ”invariant manifolds
indicate that there exist gradient flow paths connecting pairs of embedded fixed
points defined in Theorem 1”? Unless I’m missing something simple this may
need a reference/proof/discussion.

3) Can the authors justify that the learning between saddles is incremental?
It is not apparent to me that one could not ’skip’ several stages of complexity,
or that (noting the numerical errors and stochastic effects in practice) in
general some perturbations may not push the flow away from following saddle-to-saddle dynamics.

4) Could the authors explain how their theory interacts with times when
one does not escape a simple representation, sometimes at the cost of failing to
minimize the loss function. Papers I have in mind here are [4, 3].

I then have three more points:

5) In 300 ”Subsequent iterations of saddle-to-saddle dynamics operate similarly.” Providing a full account of visited saddles in the case of a diagonal
network [11] shows that the saddles one visits in this setting follow iterates
of AISSM [5], and it is non-trivial to prove. Is this setting simpler owing to
the comments in paragraph 4 of A.1 noting ”Linear networks with scalar input
or scalar output do not have the embedded fixed points described by Theorem
1 except the zero fixed point”? Otherwise it seems like there might still be more to say.

6) Could the authors briefly explain how [14] differs in setup to allow saddle-to-saddle dynamics, as compared with comments in Sect 7?

7) Typo: 311: follow

References:

[1] Sanjeev Arora, Nadav Cohen, Wei Hu, and Yuping Luo. Implicit regularization in deep matrix factorization. NeurIPS 2019.

[2] Enric Boix-Adsera, Etai Littwin, Emmanuel Abbe, Samy Bengio, and
Joshua Susskind. Transformers learn through gradual rank increase, NeurIPS 2023.

[3] Etienne Boursier and Nicolas Flammarion. Early alignment in two-layer
networks training is a two-edged sword. Journal of Machine Learning Research, 26(183):1-75, 2025.

[4] Etienne Boursier and Nicolas Flammarion. Simplicity bias and optimization
threshold in two-layer ReLU networks. ICML 2025.

[5] Martin Burger, Michael Moller, Martin Benning, and Stanley Osher. An
adaptive inverse scale space method for compressed sensing. Mathematics
of Computation, 82(281):269-299, 2013.

[6] Yatin Dandi, Florent Krzakala, Bruno Loureiro, Luca Pesce, and Ludovic
Stephan. How two-layer neural networks learn, one (giant) step at a time. Journal of Machine Learning Research 25:1-165, 2024.

[7] K Fukumizu and S Amari. Local minima and plateaus in hierarchical
structures of multilayer perceptrons. Neural networks, 13(3):317-327, 2000.

[8] Gauthier Gidel, Francis R. Bach, and Simon Lacoste-Julien. Implicit regularization of discrete gradient dynamics in deep linear neural networks. NeurIPS 2019.

[9] Suriya Gunasekar, Jason D Lee, Daniel Soudry, and Nati Srebro. Implicit
bias of gradient descent on linear convolutional networks. NeurIPS 2018.

[10] Michael Kleinman, Alessandro Achille, and Stefano Soatto. Critical
learning periods emerge even in deep linear networks. ICLR 2024.

[11] Scott Pesme and Nicolas Flammarion. Saddle-to-saddle dynamics in diagonal linear networks. NeurIPS 2023.

[12] Andrew M Saxe, James L McClelland, and Surya Ganguli. Exact solutions
to the nonlinear dynamics of learning in deep linear neural networks. ICLR 2014.

[13] Zhenfeng Tu, Santiago Tomas Aranguri Diaz, and Arthur Jacot. Mixed dynamics in linear networks: Unifying the lazy and active regimes. NeurIPS 2024.

[14] Yaoyu Zhang, Zhongwang Zhang, Tao Luo, and Zhiqin J Xu. Embedding principle of loss landscape of deep neural networks. NeurIPS 2021.

---

> ### Author Response · Authors · 2025-11-23
>
> Thank you very much for your in-depth review. We really appreciate your expertise and the insightful feedback you provided to help us improve our paper. We're glad to hear that you find our findings interesting and view the future directions motivated by our work positively. We've incorporated your suggestions into the revised PDF, with all changes marked in blue. Our responses to your questions are provided below.
>
> - **Novelty remark on Theorem 4**
>
>   We've added Remark 2 in Section 5.1 to explicitly clarify that Theorem 4 is not intended to add technical novelty. Its purpose is to accompany Proposition 5 in enabling a clear comparison, allowing us to disentangle the data-induced and the initialization-induced timescale separations. This comparison then yields new predictions about how the network width, data statistics and initialization affect dynamics, which are novel. We also agree with the reviewer that Proposition 5 is technically novel.
>
>   We've also expanded Appendix A.1 to discuss the linear network dynamics literature in more detail. In a seminal work on deep linear network learning dynamics ([Saxe et al., 2014](https://arxiv.org/abs/1312.6120), [12] in your reference list), aligned weights were introduced as an ansatz, with the analysis assuming aligned initial weights rather than deriving alignment from small isotropic initialization. This ansatz is known in the linear network literature as the "spectral initialization" assumption [(Tarmoun et al., 2021, Table 1)](https://proceedings.mlr.press/v139/tarmoun21a.html). As for showing aligned weights from small isotropic initialization, the most related study may be [Atanasov et al. (2021)](https://arxiv.org/abs/2111.00034), which showed alignment in the scalar output case and coined the effect "silent alignment". Our Theorem 4 extends the “silent alignment” to the vector output case and agrees with it when the output is scalar.
>
> - **Discussion on general nonlinear activation**
>
>   Thank you for this suggestion. We added a paragraph at the end of Section 5 to discuss general smooth activation functions and added Figure 4 to provide more examples. When the activation $\phi(x;u_i)$ is smooth but not a polynomial, we can Taylor expand $\phi(x;u_i)$ around $u_i=0$ and examine the lowest-order non-vanishing term. With small initialization, $u_i\approx 0$, the early dynamics is dominated by the lowest-order non-vanishing term.
>
>   To illustrate this, we discuss two examples here. First, for a two-layer tanh network, the lowest-order non-vanishing term of $\phi(x;u)=\text{tanh}(u^\top x)$ is the linear term, so the early dynamics is similar to a linear network. Indeed, as shown in Figure 4B, weights become rank-one in early dynamics. But because rank-one weights do not place a tanh network on invariant manifolds, the subsequent dynamics is not saddle-to-saddle. Second, for a quadratic tanh network, the lowest-order non-vanishing term of $\phi(x;u)=(u^\top x)^2\cdot\text{tanh}(u^\top x)$ is quadratic, so the early dynamics produces a timescale separation between units. Because the weight configuration in which one or two units have nonzero weights and the rest are zero corresponds to invariant manifolds for the quadratic tanh network, it exhibits saddle-to-saddle dynamics, as shown in Figure 4D.
>
>   We also added Figure 5 showing the learning dynamics of leaky ReLU networks with different negative slopes.
>
> - **Discussion on the role of depth**
>
>   We expanded our discussion on deep networks in Section 7 to specify open questions and to present a conjecture for predicting which type of timescale separation (between directions or units) arises within a layer of a deep network. We also have a discussion about depth in our response to a specific question from Reviewer Peza.

---

> ### Author Response · Authors · 2025-11-23
> **Response to questions 1-7**
>
> 1. **Exhaustiveness of invariant manifolds**
>
>    The exhaustiveness of the invariant manifolds defined in Theorem 3 is indeed an interesting open question. We've added a paragraph in the Discussion Section to clarify that the exhaustiveness question remains open. We include a note highlighting that the invariant manifolds in Theorem 3 arise solely from the architecture and thus hold for any training data sets. A further question for future research is whether particular data sets may induce additional invariant manifolds, beyond the data-agnostic ones.
>
> 2. **Why invariant manifolds connect pairs of saddles**
>
>    This is a good point because not every embedded fixed point lies on the invariant manifolds in Theorem 3. The embedded fixed points lying on invariant manifolds are the ones embedded in a particular way. For example, in Equation (4), the embedded fixed point with $\gamma_v=1/2$ (i.e. splitting a unit into two equal halves) lies on the invariant manifolds $\theta_i = \theta_j$ in Theorem 3(i). We've added Appendix F.4 and referred to it in the main body, specifying the subset of embedded fixed points that lie on invariant manifolds.
>
> 3. **When GD may skip several stages of complexity**
>
>    Your intuition is right -- it is definitely possible to skip several stages of complexity. In Figure 2B, the linear network trained on data with equal eigenvalues (power law exponent $\kappa=0$) does not exhibit plateaus except the initial one due to escaping the zero fixed point. This corresponds to the largest singular value having multiplicity $r = D$ in Theorem 4, causing the solution to jump directly from effective width $0$ to $D$, skipping the stages in between. We've added a sentence in the "Effect of data distribution" paragraph to highlight this point.
>
> 4. **Downside of simplicity bias**
>
>    Thank you for raising this question. According to the no free lunch theorem, no single inductive bias is universally beneficial, and simplicity bias is no exception. We've added a related work subsection on "Simplicity bias" in Appendix A.3, where we discuss the literature showing cases in which simplicity bias can hinder generalization or optimization.
>
> 5. **A full account of visited saddles**
>
>    We agree that a full account of the visited saddles is non-trivial, even in the case of diagonal linear networks. In this sense, our claim is weaker because we only state that the effective width increases from $h$ to $(h+1)$ (with non-degenerate data statistics) at each saddle-to-saddle transition, without specifying which effectively width-$(h+1)$ saddle it chooses to visit. For a two-layer linear network with a rank-$D$ target linear map, there are $D \choose h $ fixed points with effective width $h$ in the function space. We've added Remark 3 in Appendix G.3 to explain this point.
>
>    As for the linear network with scalar output case you mentioned, it is indeed free of this problem because there is no nonzero saddle and thus no intermediate plateau. The gradient descent dynamics from small initialization escapes from the zero fixed point and directly goes to a global minimum.
>
> 6. **Comparison with Zhang et al. (2021)**
>
>    Thank you for this suggestion. We've added a paragraph in Appendix A.1 to discuss the related work by [Zhang et al. (2021)](https://arxiv.org/abs/2105.14573). A key difference is that Zhang et al. (2021) empirically observed saddle-to-saddle dynamics in two-layer networks trained with Adam, whereas we study saddle-to-saddle dynamics under gradient descent. While their theoretical analysis is done for gradient flow, it focuses on fixed points and does not study invariant manifolds or dynamics around fixed points.
>
> 7. We've fixed the typo "follow". We've also added [Boix-Adsera et al. (2023)](https://arxiv.org/abs/2306.07042) to our references. Thank you for this note and for providing a comprehensive reference list accompanying your review!
>
> We hope our revision is helpful and welcome any further feedback.

---

> > ### Comment · Reviewer_S6X7 · 2025-11-26
> >
> > Thank you very much for the rebuttal.
> >
> > What I was asking in Question 4 was not whether a simplicity bias can be good or bad, rather how does your work reconcile with the fact that sometimes optimization can reach a critical point of a low-rank manifold which does not minimise the loss, and yet the dynamics do not shift to a new manifold, i.e., they stop earlier than your work suggests?
> >
> > As for Question 6, my point was that your citation of Zhang et al. (2021) in section 1, where you say that they documented saddle-to-saddle dynamics in tanh networks seems to contradict to some extent your claim in section 7, where under "Condition for saddle-to-saddle dynamics" you say that tanh network probably do not have saddle-to-saddle dynamics in general.  Can you somehow resolve those two competing statements in the main paper (where they are made) rather than in the appendix?

---

> > > ### Author Response · Authors · 2025-11-27
> > >
> > > Thank you very much for your engagement and for clarifying the questions.
> > >
> > > For question 4, we agree that optimization can terminate at a local minimum rather than a global one. In Section 3, we discussed that the embedded fixed point associated with the global minimum of a narrow network (with nonzero loss) may appear as either a saddle or local minimum in a wider network. They are guaranteed to be saddles in deep linear networks and, under mild conditions, are saddles in general architectures. Thus, exceptions do exist. We have now made this point explicit in Section 7.
> > >
> > > For question 6, thank you for pointing out this contradiction. Although we have not yet been able to reproduce the tanh network setting from Zhang et al. (2021) due to limited implementation details, we acknowledge that our theory does not strictly exclude the possibility of saddle-to-saddle dynamics occurring in tanh networks for particular data sets.
> > >
> > > To address both questions, we added the following paragraph to Section 7:
> > >
> > > "We do not rule out the possibility of saddle-to-saddle dynamics arising for particular datasets in networks that violate the first condition (Zhang et al., 2021). For example, if the target weights of a tanh network are small enough that activations stay within the approximately linear region near zero, the tanh network would behave like a linear network and exhibit saddle-to-saddle dynamics. For certain networks and datasets, we also do not rule out the possibility that saddle-to-saddle dynamics may terminate at a local minimum rather than at a global minimum (Holzmuller & Steinwart, 2022; Boursier & Flammarion, 2025a;b)."
> > >
> > > We hope this addition is helpful and welcome any further suggestions.

---

> > > > ### Comment · Reviewer_S6X7 · 2025-11-27
> > > >
> > > > Thanks for these responses, and again for your previous ones, as well as for revising the paper.
> > > >
> > > > I am now happy to raise my rating to "6: marginally above the acceptance threshold. But would not mind if paper is rejected".

---

### Official Review · Reviewer_Peza · 2025-10-29

**Soundness:** 2
**Presentation:** 3
**Contribution:** 3
**Rating:** 6
**Confidence:** 3

**Summary:**

The paper argues that saddle-to-saddle dynamics can induce an implicit bias toward simpler solutions across diverse architectures. Leveraging permutation symmetry, the authors show that a saddle point in a narrower architecture remains a saddle point when extended to a wider one, and they identify new conditions under which such saddles can exist in larger networks. Furthermore, the paper demonstrates that, under these conditions, the dynamics evolve within an invariant manifold, suggesting that saddle-to-saddle dynamics correspond to transitions between invariant manifolds via perturbations.

More specifically, the authors unify multiple architectures within a single framework and exploit the fact that permutation symmetry is shared across them. Their analysis of saddle-to-saddle dynamics also reveals a benefit of increasing the hidden dimension HHH in self-attention architectures—a contrast to the fully connected case—which they further validate empirically.

**Strengths:**

The paper presents a unifying perspective on the widely assumed simplicity bias underlying saddle-to-saddle dynamics across various architectures and, importantly, elucidates its connection to permutation symmetry. This formalization and unification constitute the core contribution of the work.

Although the analysis is based on a heuristic scenario, it is consistent with both experimental observations and existing understanding of saddle-to-saddle dynamics. The analysis further reveals an unexpected benefit of increasing $H$ in self-attention architectures.

The paper is clearly written, and the illustrative examples effectively convey the key ideas.

**Weaknesses:**

While I fully agree with the reasoning presented in Section 5 and consider it a sufficient contribution, several steps remain heuristic.
For instance, the transition between invariant manifolds is not thoroughly explored: while the paper studies the existence of invariant manifolds and the dynamics within them, it offers limited explanation for why the dynamics should adhere to these manifolds in the first place. Proposition 5 states that if one variable takes a larger value, the others must be small, yet it does not clarify why one should dominate in the first place. The argument is intuitively consistent with the rich-get-richer dynamics of layerwise structures, but making this connection more explicit—especially in settings where many neurons can fit the target — has been a central focus of works such as Kunin et al. (2025), which the paper also cites.

While I acknowledge that the analysis in Section 5 is restricted to two-layer networks, I believe the current analysis can be potentially misleading, particularly regarding Figures 2A and 2B and the conclusion that the saddle-to-saddle mechanism differ for different architectures. As the authors conjecture in line 354, I believe that the degree or effective depth — the number of times the learnable parameters are multiplied to achieve $f(x)$ — is the main cause of the sensitivity to $H$ and initialization: a 3-layer network with linear dependence on $\phi$ would likely exhibit similar sensitivity to $H$ and initialization as the 2-layer self-attention networks.

The intuitive description of Proposition 5 (line 331) and the conjecture on depth (line 354) seem to suggest that the number of times learnable parameters are multiplied — the effective depth — is the key factor, and that the self-attention and quadratic networks simply obtain an additional effective depth from $\phi \propto u^2$. While I believe the analysis is correct for two-layer networks, the authors could improve clarity on what specifically causes the sensitivity to $H$ and initialization, especially as the authors already discuss the limitations of Section 5 regarding depth. Please see my question 2 for a verifiable experiment.

The empirical section could also be strengthened by providing result for all architectures (convolutional, attention, quadratic) in Figures 4, to make it consistent to Figure 1. This is important as the theory claims to unify dynamics across diverse deep architectures.  Additionally, one or two experiments on real dataset would significantly strengthen the paper.

**Questions:**

## First Question
Do the authors have corresponding results for all deep architectures in Figure 4?

 ## Second Question
From the analysis related to Figures 2A and 2B, authors argue that the difference between linear attention and fully connected networks arise from the quadratic dependence of $\phi(x, u)$ on $u$. However, I believe the observation of Figures 2A and 2B arise from the depth and initialization.

Can the authors repeat the experiments of Fig 2 A and B for 3-layer diagonal-ish linear network

$f(x) =  \sum_j \sum_{i=1}^Ha_{ij}b_{ij}c_{ij}x_j$

using the same initialization as the self-attention network in the paper? Here, $a_{ij},b_{ij}, c_{ij} \in \mathbb{R}$ and $x_j$ are the entries of the input. The model, while similar to a quadratic network, does not have $u^2$ dependence. It only shares that there are three products of parameters ($abc$ instead of $vu^2$) to express $f$.

Greater effective depth—i.e., more multiplicative parameters—typically increases sensitivity to initialization, as suggested by the more Heaviside-like saddles observed with larger parameter products (e.g., your Figure 4). Notably, the self-attention and quadratic networks use a larger initialization $\epsilon=0.005$, compared to $\epsilon=10^{-6}$ for fully connected and convolutional networks; I interpret this as a practical choice to escape a longer initial saddle plateau caused by more products of small initial weights (three for quadratic/attention vs. two for fully connected/convolution).

When depth-induced sensitivity is combined with a larger (yet still “rich-regime”) initialization, increasing $H$ may simply allow the model to sample slightly larger initial effective parameters, thereby shortening the transient around the saddle. If a 3-layer diagonal-like network exhibits similar sensitivity to $H$ and initialization as observed in the self-attention networks, the analysis should emphasize the role of multiplicative parameterization rather than attributing the phenomenon primarily to the activation form only (i.e. a special case). If no such effect is observed, the current interpretation is convincing.

---

> ### Author Response · Authors · 2025-11-23
> **Response to first question**
>
> Thank you very much for your in-depth review. We really appreciate your expertise and taking the time to raise very insightful and actionable feedback. We've incorporated your suggestions into the revised PDF, with all changes marked in blue. We'd like to respond to questions below.
>
> - **Experiments on deep architectures**
>
>   Thanks for this suggestion. We've expanded Figure 6 to incorporate the deep counterparts of all architectures in Figure 1.
>
> - **Experiments on a real dataset**
>
>   We added Figure 3 showing the learning dynamics of two-layer linear / ReLU networks trained for binary classification of MNIST digits, in which we use SGD with a batch size of 64. Although the dynamics is noisier than that on synthetic datasets, there is a pronounced intermediate plateau in training. The intermediate plateau is longer when the two digits are harder to distinguish. For example, the plateau in learning to classify digits 3/5 is longer than that of digits 0/1. We also plotted the top singular values of the first-layer weight matrix. Consistent with our theory, the growth of the first and second singular values coincides with the first and second abrupt drops in the training loss, respectively. The remaining singular values are small, implying that the network has effective width 1 and 2 during the intermediate plateau and at convergence.

---

> ### Author Response · Authors · 2025-11-23
> **Response to second question**
>
> - **Discussion on how to extrapolate our results to deep architectures**
>
>   Thank you for this valuable question and for designing the verifiable experiment! We acknowledge that there was ambiguity in our initial manuscript in how to extrapolate our results to deep architectures. We've thus revised the "Deep Networks" paragraph in Section 7 to better clarify the subtleties involved and highlight open questions. We respond to your Second Question below.
>
>   We find that the different kinds of behaviors cannot be fully explained by the number of times the learnable parameters are multiplied to achieve $f(x)$. Let us consider the following three models, which all have three parameters multiplied to achieve $f(x)$.
>   $$
>   \begin{align*}
>   \text{Model A (deep linear MLP):}\quad&
>   \sum_{i=1}^{H} v_i \sum_{j=1}^{H} u_{ij} w_j^\top x \\\\
>   \text{Model B (linear attention):}\quad&
>   \sum_{i=1}^{H} V_i XX^\top K_i^\top Q_i X \\\\
>   \text{Model C (diagonal network):}\quad&
>   \sum_{i=1}^H \sum_{j} a_{ij} b_{ij} c_{ij} x_j
>   \end{align*}
>   $$
>   If the number of multiplied weights suffices to categorize models, models A,B,C should have the same category of behavior. However, we find that they don't. We observe that model A behaves similarly to two-layer linear networks: it exhibits a timescale separation between directions and the weights acquire increasing ranks during its saddle-to-saddle dynamics, as shown in Figure 6A. In contrast, models B and C behave similarly to quadratic networks: they exhibit a timescale separation between units and individual units grow during their saddle-to-saddle dynamics, as shown in Figure 1F, 4C. The width and power law exponent sweep experiments on models A,B,C are provided in Figures 2 and 7, which are consistent with their different categories of timescale separations.
>
>   Based on these observations, we believe the effect of depth cannot be fully explained by the two-layer analysis and warrants further future research. Even the definition of depth has subtlety: although the most straightforward name for model C is a 3-layer diagonal linear network, it actually fits into our expression of a 2-layer quadratic network
>   $$
>   \sum_{i=1}^H\sum_{j} a_{ij} b_{ij} c_{ij} x_j
>   = \sum_{i=1}^H \mathbf{a}_i^\top \phi(\mathbf x;\mathbf{b}_i,\mathbf{c}_i) ,
>   \quad \text{where }
>   \phi(\mathbf x;\mathbf{b}_i,\mathbf{c}_i) = \mathbf{b}_i \odot \mathbf{c}_i \odot \mathbf {x}
>   $$
>   Here $\odot$ denotes the Hadamard product. Given existing evidence, we conjecture that the key factor may not only be the number of multiplied parameters, but the number of multiplied parameters associated with a unit of the layer under consideration, i.e., parameters indexed by $i$. Although models A,B,C all have three multiplied parameters, the number of multiplied parameters with index $i$ (or index $j$) is 2 in model A, while this number for models B,C is 3, suggesting their different behaviors. We've added this caveat to the discussion section to avoid misleading extrapolations of our results to deep networks.
>
> - **Another motivation for categorizing architectures based on linear /  quadratic activations**
>
>   Another motivation for categorizing architectures by whether $\phi(x;u_i)$ is linear or quadratic in $u_i$ is that it can also help us understand networks with general, non-polynomial activations. When $\phi(x;u_i)$ is not a polynomial, we can Taylor expand $\phi(x;u_i)$ around $u_i=0$ and examine the lowest-order non-vanishing term. With small initialization, $u_i\approx 0$, the early dynamics is dominated by the lowest-order non-vanishing term.
>
>   For example, the lowest-order non-vanishing term of $\phi(x;u)=\text{tanh}(u^\top x)$ is the linear term and that of $\phi(x;u)=(u^\top x)^2\cdot\text{tanh}(u^\top x)$ is the quadratic term. As shown in Figure 4B, the 2-layer tanh network develops rank-one weights in early dynamics similar to a linear network. In contrast, the 2-layer quadratic tanh network has a timescale separation between units similar to a quadratic network and exhibits saddle-to-saddle dynamics; see Figure 4D.
>
>   We added a paragraph at the end of Section 5 to explain this point.
>
> We hope our revision is helpful and welcome any further discussion.

---

> > ### Comment · Reviewer_Peza · 2025-11-26
> >
> > I thank the authors for adding a real-data experiment and for making Figure 6 consistent with Figure 1. Section 5 and the analysis of Figure 2A could still be improved to better convey the intuition behind the role of multiplicative parameters—specifically, how they should be decoupled in terms of H, rather than fully directional as in FCNs, to benefit from increased H. Nonetheless, the current revision is sufficient.
> >
> > Based on this, I am raising my score to 8.

---

> > > ### Author Response · Authors · 2025-11-27
> > >
> > > Thank you very much for your feedback. We're glad the real-data experiment and updates to Figure 6 were helpful. We also appreciate your insightful comments regarding Section 5 and will keep the multiplicative parameters in mind for future research.

---

### Official Review · Reviewer_9Lv2 · 2025-10-30

**Soundness:** 3
**Presentation:** 3
**Contribution:** 2
**Rating:** 6
**Confidence:** 3

**Summary:**

This paper presents a unifying theoretical framework explaining simplicity bias in gradient descent dynamics through what the authors call saddle-to-saddle dynamics. The authors argue that across diverse architectures—fully connected, convolutional, ReLU, quadratic, and self-attention networks—training often progresses by traversing a sequence of saddles, each corresponding to increasingly complex solutions. The work generalizes earlier analyses of linear networks, introducing formal results on embedded fixed points and invariant manifolds to explain why networks successively recruit additional “effective units.” The paper also provides clear implications for how initialization, network width, and data distribution affect the number and duration of learning plateau.

**Strengths:**

1. The paper offers a broad, architecture-agnostic framework that connects dynamical simplicity bias phenomena seen across many model types. The idea of interpreting learning stages as saddle-to-saddle transitions is elegant and theoretically grounded.
2. The analysis of fixed points, invariant manifolds, and timescale separation (linear vs. quadratic cases) is mathematically sound, and the simulations (Figure 1 & 2) convincingly demonstrate the predicted dynamics and validate the theory’s implications for initialization and data statistics.
3. Despite the depth of the mathematics, the exposition is clean, well-structured, and includes intuitive visualizations that aid understanding.

**Weaknesses:**

1. The simulations, while well-chosen, are limited to small synthetic examples. It would strengthen the paper to show whether the predicted stage transitions are visible in real-world or larger-scale training runs.
2. Some proofs depend on idealized gradient flow dynamics and homogeneity assumptions that may not hold under practical stochastic optimization with large step size.

**Questions:**

See Weakness.

---

> ### Author Response · Authors · 2025-11-23
>
> Thank you very much for your thoughtful review. We're glad to hear that you find our proposed framework elegant and and theoretically grounded. We've revised our manuscript based on your feedback, with all changes marked in blue. We'd like to respond to your questions below.
>
> - **Visible stages in a more realistic setup**
>
>   We added Figure 3 showing the learning dynamics of two-layer linear / ReLU networks trained for binary classification of MNIST digits, in which we use SGD with a batch size of 64. Although the dynamics is noisier than that on synthetic datasets, there is a pronounced intermediate plateau in training. The intermediate plateau is longer when the two digits are harder to distinguish. For example, the plateau in learning to classify digits 3/5 is longer than that of digits 0/1. We also plotted the top singular values of the first-layer weight matrix. Consistent with our theory, the growth of the first and second singular values coincides with the first and second abrupt drops in the training loss, respectively. The remaining singular values are small, implying that the network has effective width 1 and 2 during the intermediate plateau and at convergence.
>
>   **Stochastic optimization with large step size**
>
>   In Figure 3, we use SGD with a batch size of 64. This indeed introduces noise into the dynamics, but the stages remain pronounced. We agree that gradient flow is an idealized  setting that captures the behaviors of SGD in the limit of a small step size and a large batch size, where noise vanishes. While analyzing the effects of large step sizes and stochastic mini-batches is an important direction for future work, we believe studying the gradient flow dynamics serves as a necessary prerequisite and remains the primary scope of this paper.
>
> - **Homogeneity assumption**
>
>   Another motivation for categorizing architectures by whether $\phi(x;u_i)$ is linear or quadratic in $u_i$ is that it can also help us understand networks with general
>
>   We added a paragraph at the end of Section 5 to discuss how our analysis can help understand two-layer networks with general smooth activation functions that are not homogeneous polynomials. When $\phi(x;u_i)$ is not a polynomial, we can Taylor expand $\phi(x;u_i)$ around $u_i=0$ and examine what the lowest-order non-vanishing term is. For example, the lowest-order non-vanishing term of $\phi(x;u)=\text{tanh}(u^\top x)$ is the linear term and that of $\phi(x;u)=(u^\top x)^2\cdot\text{tanh}(u^\top x)$ is the quadratic term. As shown in Figure 4, the 2-layer tanh network develops rank-one weights in early dynamics similar to a linear network, while the 2-layer quadratic tanh network has a timescale separation between units similar to a quadratic network and exhibits saddle-to-saddle dynamics. This occurs because with small initialization, $u_i\approx 0$, the early dynamics is dominated by the lowest-order non-vanishing term.
>
>   We hope our revision is helpful. Please let us know if we misunderstood your question or if anything remains unclear.

---

> > ### Comment · Reviewer_9Lv2 · 2025-11-26
> >
> > I thank the authors for all the helpful additions. I will keep my score as it‘s already positive.

---

### Author Response · Authors · 2025-11-23
**Summary of rebuttal revision**

We thank all five reviewers for their insightful and constructive comments. We appreciate that all reviewers recognized our main contribution in providing a unified framework for understanding simplicity bias across different architectures, and gave positive feedback on the exposition, novelty, and significance. We also appreciate the particular aspects that different reviewers found interesting and surprising, such as the unexpected benefit of increasing width in linear self-attention (Reviewer Peza), a new understanding of lazy/rich regimes (Reviewer 8EAS), and the technical novelty in analyzing quadratic network dynamics (Reviewer S6X7). It is encouraging to see that distinct elements of the work resonated with different readers, suggesting that our theory may be useful to a range of different audiences.

Based on reviewers' suggestions, we've revised the PDF and marked the changes in blue. We summarize our main revisions below:

- To elaborate on the role of depth, we expanded Figure 6 to include training dynamics of deep networks with more architectures and added more details in Section 7 (Discussion).
- To provide a more comprehensive literature review, we expanded Appendix A (Additional Related Work) to discuss spectral bias in the kernel regime, stationary simplicity bias, and weight alignment in linear networks.
- To demonstrate saddle-to-saddle dynamics beyond synthetic datasets, we added Figure 3, which shows saddle-to-saddle dynamics in learning binary classification of MNIST digits.

---

### Author Response · Authors · 2025-12-03
**Summary of rebuttal discussion**

We thank the reviewers for their thoughtful reviews and engagement during the rebuttal period. Below is a brief summary of the discussions:

- Reviewer 9Lv2 confirmed they maintained their review with rating 6.
- Reviewer Peza increased their rating from 6 to 8 following our response and revisions.
- Reviewer S6X7 engaged in further constructive discussion and subsequently increased their rating from 4 to 6.
- Reviewer axwH did not have the opportunity to respond before the early termination of discussion; their initial rating was 6.
- Reviewer 8EAS confirmed they maintained their review with rating 8.

These discussions are kept intact on this page. The rating changes are verifiable as the reviewers posted messages specifying their changes when they edited their reviews.

---

### Meta-Review · Area_Chair_PR76 · 2026-01-04

**Summary:**

Reviewers raised concerns from several directions, including the novelty of the theoretical results (in particular, Theorem 4), the practical validity of the theoretical findings, the empirical evaluation, more detailed properties of the manifolds, and the relationship to other established concepts. The authors have successfully addressed these concerns through detailed responses and revisions to the manuscript. Given that the contribution of this paper is solid, I recommend acceptance.

**Reviewer Concerns:**

I believe that most of the concerns have been addressed in the rebuttal, and no significant issues remain outstanding.

**Reviewer Scores:**

**Reviewer 9Lv2** raised concerns about the lack of simulations in realistic settings and the validity of the homogeneity assumption in the analysis of gradient flow dynamics in practical scenarios. In the rebuttal, the authors added experiments on MNIST along with additional discussion. These additions address the concerns, and the reviewer is happy to maintain the original positive score.

**Reviewer Peza** raised concerns regarding deeper architectures. In their response, the authors provided more detailed discussion and additional experiments, which satisfied the reviewer. As a result, the reviewer is happy to increase the score.

**Reviewer S6X7** raised concerns about the novelty of Theorem 4 and several issues related to invariant manifolds, including uniqueness, properties, and transitions between saddles. The authors provided detailed responses and addressed these issues. Therefore, the reviewer is happy to increase the score.

**Reviewer 8EAS** raised concerns about the relationship to other well-established concepts, such as spectral bias and the lazy/rich regimes. These concerns have been addressed in the authors’ response, and the reviewer is happy to maintain the positive score.

---

### Decision · Program_Chairs · 2026-01-26

Accept (Poster)